# Optimal ablation for interpretability

**Maximilian Li**
Harvard University

**Lucas Janson**
Harvard University

## Abstract

Interpretability studies often involve tracing the flow of information through machine learning models to identify specific model components that perform relevant computations for tasks of interest. Prior work quantifies the importance of a model component on a particular task by measuring the impact of performing *ablation* on that component, or simulating model inference with the component disabled. We propose a new method, *optimal* ablation (OA), and show that OA-based component importance has theoretical and empirical advantages over measuring importance via other ablation methods. We also show that OA-based component importance can benefit several downstream interpretability tasks, including circuit discovery, localization of factual recall, and latent prediction.

## 1 Introduction

Interpretability work in machine learning (ML) seeks to develop tools that make models more intelligible to humans in order to better monitor model behavior and predict failure modes. Early work in interpretability sought to identify relationships between model outputs and input features (Ribeiro et al., 2016; Covert et al., 2022), but with only black-box query access to observe inputs and outputs, it can be difficult to evaluate a model's internal logic. Hence, recent interpretability work often seeks to take advantage of access to an ML model's intermediate computations to gain insights about its decision-making, focusing on deciphering internal units like neurons, weights, and activations (Räuker et al., 2022). In addition to finding associations between latent representations and semantic concepts (Bau et al., 2017; Mu and Andreas, 2021; Burns et al., 2022; Li et al., 2023; Gurnee and Tegmark, 2024), interpretability studies aim to investigate how intermediate results are used in later computation and identify specific model components that extract relevant information or perform necessary computation to produce low loss on particular inputs.

A key instrumental goal in interpretability is quantifying the importance of a particular model component for prediction. Studies often measure a component's importance by performing *ablation* on that component and comparing model performance with and without the component ablated. Ablating a component typically entails replacing its value with a counterfactual value during inference, sometimes referred to as "activation patching." However, the details vary greatly and there is a lack of consensus on best practices (Heimersheim and Nanda, 2024). For example, Meng et al. (2022) adds Gaussian noise to ablated components' values, while Geva et al. (2023) replaces these values with zeros, and Ghorbani and Zou (2020) replaces them with their means over the training distribution.

In this paper, we present *optimal ablation* (OA), a new method that sets a component's value to a constant that minimizes the expected loss of the ablated model. In section 2, we introduce OA and show that it is, in a certain sense, a canonical choice of ablation method for measuring component importance. We then show that using OA produces meaningful improvements for several common downstream applications of measuring component importance. In section 3, we apply OA to algorithmic circuit discovery (Conmy et al., 2023), or the identification of a sparse subset of components sufficient for performance on a subset of the training distribution. We demonstrate

38th Conference on Neural Information Processing Systems (NeurIPS 2024).

that OA-based performance is a reasonable metric for evaluating circuits and using OA for circuit discovery produces smaller and lower-loss circuits than previous ablation methods. In deploying OA to this application, we also propose a new search algorithm for identifying sparse circuits that achieve low loss according to any performance metric. In section 4, we use OA to locate relevant components for factual recall (Meng et al., 2022) and show that OA better identifies important components compared to prior work. In section 5, we apply OA to latent prediction (Belrose et al., 2023a), or the elicitation of output predictions using intermediate activations. We propose an OA-based prediction map and show that it has better predictive power and causal faithfulness than previous methods.

## 2 Optimal ablation

### 2.1 Motivation

Let $\mathcal{M}$ represent a model that is trained to minimize $\mathbb{E}_{X,Y} \mathcal{L}(\mathcal{M}(X), Y)$ for a given loss function $\mathcal{L}$ and a distribution of random input-label pairs $(X, Y)$. A common theme in interpretability work is quantifying the importance of some model component $\mathcal{A}$ for inference. For example, $\mathcal{A}$ could represent a single neuron, a direction in activation space, a token embedding, an attention head, or an entire transformer layer; further examples of $\mathcal{A}$ are discussed in Section 3 and Appendix C.2. Let $\mathcal{A}(x)$ represent the value of $\mathcal{A}$ when the model is evaluated on input $x$. To identify domain specialization among model components, studies often measure the importance of $\mathcal{A}$ for model performance on a particular "subtask" $\mathcal{D}$, or an interpretable human-curated distribution of input-label pairs that captures a general aspect of model behavior. We write $(X, Y) \sim \mathcal{D}$ to indicate sampling input-label pairs or $X \sim \mathcal{D}$ to indicate sampling only inputs from the subtask distribution.

Although some works quantify component importance via gradients (Leino et al., 2018; Dhamdhere et al., 2018), such an approach is inherently *local* (even when aggregated over many inputs) and as such can fail to accurately represent the overall importance of $\mathcal{A}$ in highly nonlinear models. Instead, most interpretability studies use *ablation* to quantify the importance of $\mathcal{A}$ by studying the gap in performance between the full model $\mathcal{M}$ and a modified version $\mathcal{M}^{\backslash \mathcal{A}}$ with $\mathcal{A}$ ablated:

$$\Delta(\mathcal{M}, \mathcal{A}) := \mathcal{P}(\mathcal{M}^{\backslash \mathcal{A}}) - \mathcal{P}(\mathcal{M}), \tag{1}$$

where $\mathcal{P}$ is a selected metric for model performance. In the context of measuring importance, we argue that the construction of $\mathcal{M}^{\backslash \mathcal{A}}$ is motivated by the following question:

*What is the best performance on subtask $\mathcal{D}$ the model $\mathcal{M}$ could have achieved without component $\mathcal{A}$?*

To formalize this question, we split its meaning into four elements.

I. "**Performance on subtask $\mathcal{D}$**": The relevant performance metric $\mathcal{P}$ is the expected loss on the subtask with respect to the full model's predictions: $\mathcal{P}(\tilde{\mathcal{M}}) = \mathbb{E}_{X \sim \mathcal{D}} \mathcal{L}(\tilde{\mathcal{M}}(X), \mathcal{M}(X))$ (note that $\mathbb{E}$ aggregates over any randomness in $\tilde{\mathcal{M}}$).[1] We call $\Delta$ defined using this choice of $\mathcal{P}$ the *ablation loss gap*.

II. "**Model $\mathcal{M}$ could have achieved**": Since the goal of measuring component importance is to interpret the model $\mathcal{M}$, the ablated model $\mathcal{M}^{\backslash \mathcal{A}}$ should be constructed solely by changing the value of $\mathcal{A}$, holding fixed all other parts of $\mathcal{M}$. We write $\mathcal{M}_{\mathcal{A}}(x, a)$ to indicate computing $\mathcal{M}$ on input $x$ while setting the value of $\mathcal{A}$ to $a$ instead of $\mathcal{A}(x)$ (see Appendix C.2 for details).

III. "**Without component $\mathcal{A}$**": The ablated model $\mathcal{M}^{\backslash \mathcal{A}}(x)$ should use a value for $\mathcal{A}$ that is devoid of information about the input $x$.

Elements II and III motivate the following definition:

**Definition 2.1** (Total ablation). *A total ablation method satisfies $\mathcal{M}^{\backslash \mathcal{A}}(X) = \mathcal{M}_{\mathcal{A}}(X, A)$ for some random $A$, where $A \perp\!\!\!\perp X$. (Conversely, for a* partial *ablation method, $A$ can depend on $X$.)*

IV. "**Best**" **performance**: To measure the importance of $\mathcal{A}$, we wish to understand how much model performance degrades *as a result* of ablating $\mathcal{A}$. If two constructions of the ablated model $\mathcal{M}^{\backslash \mathcal{A}}$ both perform total ablation on $\mathcal{A}$ but one performs worse than another, the former's underperformance cannot be entirely be attributed to ablating $\mathcal{A}$, since the latter also totally

---

[1]A common alternative choice is measuring performance in terms of proximity to the correct labels rather than the original model predictions, i.e. $\tilde{\mathcal{P}}(\tilde{\mathcal{M}}) = \mathbb{E}_{(X,Y) \sim \mathcal{D}} \mathcal{L}(\tilde{\mathcal{M}}(X), Y)$. See Appendix C.4 for discussion.

ablates $\mathcal{A}$ and yet does not degrade performance to the same extent. Thus, the relevant $\mathcal{M}^{\backslash\mathcal{A}}$ for measuring importance should incur the *minimum* $\Delta$ among total ablation methods.

To make element IV more concrete, for an ablation method satisfying element II and a given $x$, replacing $\mathcal{A}(x)$ with $a$ can degrade the ablated model's performance via both deletion and spoofing:

1. **Deletion**. The original value $\mathcal{A}(x)$ could carry informational content specific to $x$ that serves some mechanistic function in downstream computation and helps the model arrive at its prediction $\mathcal{M}(x)$. Replacing $\mathcal{A}(x)$ with some other value $a$ could *delete* this information about $x$, hindering the model's ability to compute the original $\mathcal{M}(x)$.

2. **Spoofing**. The replacement value $a$ could "spoof" the downstream computation by *inserting* information about the input that either:
   (a) causes the model to treat $x$ like a different input $x'$;[2] or
   (b) causes the model to treat $x$ in a way that it never treated *any* training input, if the new information is *inconsistent* with information about $x$ derived from retained components.

   In the latter case, the confluence of conflicting information could cause later activations to become incoherent, causing performance degradation because these abnormal activations were not observed during training and thus not necessarily regulated to lead to reasonable predictions.

To measure importance, we seek to isolate the contribution of effect 1 to performance degradation. While total ablation methods all capture a maximal deletion effect since $A$ does not depend on $X$, measuring the "best" performance minimizes the additional contribution of potential spoofing effects.

## 2.2 Prior work

Component importance is strongly related to *variable* importance (Sobol, 1993; Homma and Saltelli, 1996; Breiman, 2001; Ishwaran, 2007; Gromping, 2009), a longstanding area of research in statistics and ML. The vast body of work in this area is too extensive to review here, and the recent surge of research interest in interpreting *internal* model components has raised new and unique challenges relating to the values of internal components often being deterministically related. Thus, we focus on recent work applying ablation methods to internal components in this section, and defer broader discussion to Appendix B.

Ablation methods previously applied to internal components can be separated into *subtask-agnostic* methods, which can be applied out-of-the-box to any subtask, and *subtask-specific* methods, which only work on subtasks for which inputs satisfy a designated structure, and even then require human ingenuity to adapt to each new subtask.

Subtask-agnostic ablation methods include *zero ablation* (Baan et al., 2019; Lakretz et al., 2019; Bau et al., 2020; Olsson et al., 2022; Geva et al., 2023; Cunningham et al., 2023; Merullo et al., 2024; Gurnee et al., 2024), which replaces $\mathcal{A}(x)$ with zero, i.e. $\mathcal{M}^{\backslash\mathcal{A}}(x) = \mathcal{M}_{\mathcal{A}}(x, 0)$; *mean ablation* (Ghorbani and Zou, 2020; McDougall et al., 2023; Tigges et al., 2023; Gould et al., 2023; Li et al., 2024; Marks et al., 2024), which replaces $\mathcal{A}(x)$ with its mean, i.e. $\mathcal{M}^{\backslash\mathcal{A}}(x) = \mathcal{M}_{\mathcal{A}}(x, \mathbb{E}_{X'\sim\mathcal{D}}[\mathcal{A}(X')])$; and *(marginal) resample ablation* (Chan et al., 2022; Lieberum et al., 2023; McGrath et al., 2023; Belrose et al., 2023a; Rushing and Nanda, 2024), which replaces $\mathcal{A}(x)$ with $\mathcal{A}(X')$ for an independent copy $X' \sim \mathcal{D}$ of the input, i.e. $\mathcal{M}^{\backslash\mathcal{A}}(x) = \mathcal{M}_{\mathcal{A}}(x, \mathcal{A}(X'))$. While zero, mean, and resample ablation are total ablation methods, adding Gaussian noise to $\mathcal{A}(x)$ (Meng et al., 2022) is a subtask-agnostic partial ablation method. These methods are all applicable to any subtask.

On the other hand, subtask-specific ablation methods rely on particular details of a chosen subtask. Singh et al. (2024) replaces $\mathcal{A}(x)$ with interpretable values, e.g. setting an attention pattern to copy from the previous token, while Goldowsky-Dill et al. (2023) replaces $\mathcal{A}(x)$ with $\mathcal{A}(x^*)$ for an interpretable reference input $x^*$. Hanna et al. (2023) employs *counterfactual ablation* (CF), a partial ablation method that replaces $\mathcal{A}(x)$ with $\mathcal{A}(\pi(x))$, where $\pi$ is a map that sends each input $x$ to a "neutral" (potentially random) input $\pi(x)$ that preserves most aspects of $x$ but removes information relevant to the subtask, i.e. $\mathcal{M}^{\backslash\mathcal{A}}(x) = \mathcal{M}_{\mathcal{A}}(x, \mathcal{A}(\pi(x)))$. Wang et al. (2022) also considers a counterfactual distribution of inputs for *counterfactual mean ablation*, which replaces $\mathcal{A}(x)$ with its mean over the distribution of counterfactuals, i.e. $\mathcal{M}^{\backslash\mathcal{A}}(x) = \mathcal{M}_{\mathcal{A}}(x, \mathbb{E}_{(X'\sim\mathcal{D}),\pi}\mathcal{A}(\pi(X')))$.

---

[2]See Appendix D.1 for a brief example of why "treating $x$ like $x'$" goes beyond deletion.

Subtask-specific methods can be useful, but it is usually unclear how to generalize them beyond the subtask originally selected. CF is the most popular among these methods, leveraged by a range of manual (Vig et al., 2020; Merullo et al., 2023; Stolfo et al., 2023; Tigges et al., 2023; Hendel et al., 2023; Heimersheim and Janiak, 2023; Todd et al., 2024; Marks et al., 2024) and algorithmic (Conmy et al., 2023; Syed et al., 2023) studies and recommended by meta-studies (Zhang and Nanda, 2024; Heimersheim and Nanda, 2024). For text data, the effectiveness of CF relies heavily on token parallelism between $x$ and $\pi(x)$, which typically share exact tokens at all but a few sequence positions. Though studies have thus far focused on toy subtasks for which suitable mappings $\pi$ are relatively easily constructed, it may be difficult or impossible to select well-suited input pairs for certain subtasks (see Appendix F.3 for a few simple examples), especially more general model behaviors. Even for subtasks that admit such a mapping, how $\pi(x)$ is engineered to withhold subtask-relevant information differs from subtask to subtask, and the construction of $\pi$ for each particular subtask is a subjective process that requires human ingenuity. Finally, CF is only a partial ablation method; since $\mathcal{A}(\pi(x))$ depends on $x$, it may give away information about $x$ that is useful for performance on $\mathcal{D}$.

## 2.3 Definition and properties of optimal ablation

We present *optimal ablation* (OA), our proposed approach to simulating component removal.

**Definition 2.2** (Optimal ablation). *To ablate $\mathcal{A}$, we replace $\mathcal{A}(x)$ with an "optimal" constant $a^*$.*

$$\mathcal{M}_{(\mathrm{opt})}^{\backslash \mathcal{A}}(x) := \mathcal{M}_{\mathcal{A}}(x, a^*), \qquad a^* := \arg\min_a \mathbb{E}_{X \sim \mathcal{D}} \, \mathcal{L}(\mathcal{M}_{\mathcal{A}}(X, a), \mathcal{M}(X)) \qquad (2)$$

We define $\Delta_{\mathrm{opt}}$ by plugging $\mathcal{M}_{(\mathrm{opt})}^{\backslash \mathcal{A}}(x)$ into Equation (1). Like zero, mean,[3] and resample ablation, optimal ablation is a total ablation method satisfying Definition 2.1. But among all total ablation methods, optimal ablation is optimal in the sense that it yields the lowest $\Delta$.

**Proposition 2.3.** *Let $\Delta(\mathcal{M}, \mathcal{A})$ be the ablation loss gap for some component $\mathcal{A}$ measured with any total ablation method. Then, $\Delta_{\mathrm{opt}}(\mathcal{M}, \mathcal{A}) \leq \Delta(\mathcal{M}, \mathcal{A})$.*

*Proof.* Consider a total ablation method that defines $\mathcal{M}^{\backslash \mathcal{A}}(X)$ by replacing $\mathcal{A}(X)$ with $A$ (per Definition 2.1), and let $\Delta(\mathcal{M}, \mathcal{A})$ be the measured ablation loss gap. By the tower property,

$$\Delta(\mathcal{M}, \mathcal{A}) = \mathbb{E}_{(X \sim \mathcal{D}), A} \, \mathcal{L}(\mathcal{M}_{\mathcal{A}}(X, A), \mathcal{M}(X)) = \mathbb{E}_A \left[ \mathbb{E} \left[ \mathcal{L}(\mathcal{M}_{\mathcal{A}}(X, A), \mathcal{M}(X)) \big| A \right] \right].$$

Since $A \perp\!\!\!\perp X$, $\mathbb{E} \left[ \mathcal{L}(\mathcal{M}_{\mathcal{A}}(X, A), \mathcal{M}(X)) \, \big| \, A = a \right] = \mathbb{E}_{X \sim \mathcal{D}} \mathcal{L}(\mathcal{M}_{\mathcal{A}}(X, a), \mathcal{M}(X)) =: g(a)$.

$$\forall a, \ \Delta_{\mathrm{opt}}(\mathcal{M}, \mathcal{A}) = \mathbb{E}_{X \sim \mathcal{D}} \, \mathcal{L}(\mathcal{M}_{\mathcal{A}}(X, a^*), \mathcal{M}(X)) \leq \mathbb{E}_{X \sim \mathcal{D}} \, \mathcal{L}(\mathcal{M}_{\mathcal{A}}(X, a), \mathcal{M}(X)) = g(a)$$
$$\implies \Delta_{\mathrm{opt}}(\mathcal{M}, \mathcal{A}) \leq \mathbb{E}_A \, g(A) = \Delta(\mathcal{M}, \mathcal{A}). \qquad \square$$

Optimal ablation thus provides the *unique* answer to our motivating question in Section 2.1, since it produces the best performance among all total ablation methods, including zero, mean, and resample ablation. Intuitively, OA minimizes the contribution of spoofing (effect 2 from Section 2.1) to $\Delta$ by setting ablated components to constants $a^*$ that are *maximally consistent* with information from other components, e.g. by conveying a lack of information about $x$ or by hedging against a wide range of possible $x$ rather than strongly associating with a particular input other than the original $x$. OA does not entirely eliminate spoofing, since it may be the case that every possible value of $\mathcal{A}$ conveys at least weak information to the model. However, the excess ablation gap $\Delta - \Delta_{\mathrm{opt}}$ for $\Delta$ measured with ablation methods that replace $\mathcal{A}(x)$ with a (random) value $A$ is *entirely* caused by spoofing, since replacing $\mathcal{A}(x)$ with the constant $a^*$ achieves lower loss without giving away any more information about $x$. In practice, $\Delta - \Delta_{\mathrm{opt}}$ for prior ablation methods is typically *very* large compared to $\Delta_{\mathrm{opt}}$ for both single components (see Table 1) and circuits (see Section 3.2) on prototypical language subtasks. This disparity indicates that effect 2 dominates the $\Delta$ measurements for previous ablation methods, making them poor estimators for effect 1 compared to OA.

Subtask-specific methods often try to generate consistent interventions by *conditioning* on features of the input to avoid replacing $\mathcal{A}(x)$ with values that could confuse the model. For CF, choosing $\pi(x)$ to share many tokens with $x$ mitigates the contribution of effect 2b to $\Delta_{\mathrm{CF}}$, which is the main reason the technique is so widely employed. Thus, among *previous* measures of component importance, $\Delta_{\mathrm{CF}}$, when it can be well-constructed, may be the best quantification of effect 1. To demonstrate this intuitive relation between OA and CF as techniques that aim to isolate effect 1, we perform a case study in Section 2.4 that shows that among other ablation methods, OA produces the measurements

---

[3]See Appendix D.2 for an interesting connection of OA to mean ablation.

most similar to CF. However, not only is OA more general than subtask-specific methods like CF, but $\Delta_{\mathrm{opt}}$ may still be a better estimator for effect 1 than $\Delta_{\mathrm{CF}}$ even when CF is well-defined. In Section 3, we show that for circuits, $\Delta_{\mathrm{opt}}$ is much lower than $\Delta_{\mathrm{CF}}$ *despite reflecting a weakly stronger deletion effect*, indicating that effect 2 also contributes to $\Delta_{\mathrm{opt}}$ less than it does to $\Delta_{\mathrm{CF}}$, and thus $\Delta_{\mathrm{opt}}$ is a more accurate reflection of components' informational importance.

**Computation of $a^*$.** Though it is impossible to derive $a^*$ in closed form, we find that in practice, mini-batch stochastic gradient descent (SGD) performs well at finding constants $\widehat{a}$ that greatly reduce $\Delta$ compared to heuristic point estimates like zero and the mean. We generally adopt the approach of initializing each $\widehat{a}$ to the subtask mean $\mathbb{E}_{X \sim \mathcal{D}}[\mathcal{A}(X)]$ and performing SGD to minimize $\Delta$.

## 2.4 Comparison of single-component ablation results on IOI

The Indirect Object Identification (IOI) subtask (Wang et al., 2022) consists of prompts like "When Mary and John went to the store, Mary gave the apple to ___," which GPT-2-small (Radford et al., 2019) completes with the correct indirect object noun ("John"). We use IOI as a case study because it is discussed extensively in interpretability work (Merullo et al., 2023; Makelov et al., 2023; Wu et al., 2024; Lan et al., 2024; Zhang and Nanda, 2024). To implement CF, for each prompt $x$, Wang et al. (2022) constructs a random $\pi(x)$ in which the names are replaced with random distinct names.

We evaluate $\Delta$ for attention heads and MLP blocks using zero, mean, resample, counterfactual, counterfactual mean, and optimal ablation. In Table 1, we show that among attention heads and MLP blocks, $\Delta_{\mathrm{opt}}$ accounts for only 11.1% of $\Delta_{\mathrm{zero}}$, 33.0% of $\Delta_{\mathrm{mean}}$, and 17.7% of $\Delta_{\mathrm{resample}}$ for the median component. Furthermore, among these $\Delta$ measurements, $\Delta_{\mathrm{opt}}$ has the highest highest rank correlation (0.907) with $\Delta_{\mathrm{CF}}$. Full results are shown in Appendix E.3.

Table 1: Comparison of ablation loss gap $\Delta$ on IOI

|  | **Zero** | **Mean** | **Resample** | **CF-Mean** | **Optimal** | **CF** |
|---|---|---|---|---|---|---|
| Rank correlation with CF | 0.590 | 0.825 | 0.828 | 0.833 | **0.907** | 1 |
| Median ratio of $\Delta_{\mathrm{opt}}$ to $\Delta$ | 11.1% | 33.0% | 17.7% | 31.7% | 100% | 88.9% |

## 3 Application: circuit discovery

Circuit discovery is the selection of a sparse subnetwork of $\mathcal{M}$ that is sufficient for the recovery of model performance on an algorithmic subtask $\mathcal{D}$. To define what constitutes a "sparse subnetwork," we write $\mathcal{M}$ as a computational graph with vertices $G$ and edges $E$. An edge $e_k := (\mathcal{A}_j, \mathcal{A}_i, z) \in E$ indicates that $\mathcal{A}_j(x)$ is taken as the $z$th input to $\mathcal{A}_i$ in the computation represented by the graph. To ablate edge $e_k$, we replace the $z$th input to $\mathcal{A}_i$, which is equal to $\mathcal{A}_j(x)$ during normal inference, with some value $a$. We compute $\mathcal{M}^{\tilde{E}}(X)$, which represents modified inference with edges $E \setminus \tilde{E}$ ablated, by applying this intervention for each ablated edge (see Appendices C.1 and C.2 for more details). Circuit discovery aims to select a subset of edges $\tilde{E}^* \subseteq E$ such that

$$\tilde{E}^* = \arg\min_{\tilde{E} \subseteq E} \left[ \mathbb{E}_{X \sim \mathcal{D}} \mathcal{L}(\mathcal{M}^{\tilde{E}}(X), \mathcal{M}(X)) + \mathcal{R}(\tilde{E}) \right] = \arg\min_{\tilde{E} \subseteq E} \left[ \Delta(\mathcal{M}, E \setminus \tilde{E}) + \mathcal{R}(\tilde{E}) \right] \quad (3)$$

for a regularization term $\mathcal{R}$ that measures the sparsity level (further discussed in Appendix F.4). Additionally, when implementing OA, though we could use a different constant for each edge, we instead define a single constant $a_j^*$ for each *vertex* $\mathcal{A}_j$, so that if multiple out-edges from $\mathcal{A}_j$ are ablated, the same value is passed to each of its children (further discussed in Appendix F.2).

## 3.1 Methods

We compare $\Delta(\mathcal{M}, E \setminus \tilde{E})$ measured with mean ablation, resample ablation, OA, and CF as metrics for circuit discovery. We consider the manual circuit for each subtask and circuits optimized on each $\Delta$ metric using several search algorithms.

**ACDC** (Conmy et al., 2023) identifies circuits by iteratively considering edge ablations. They start by proposing $\tilde{E} = E$, then iterate over edges $e_k$, ablating $e_k$ and updating $\tilde{E} = \tilde{E} \setminus \{e_k\}$ if the marginal impact on loss, $\Delta(\mathcal{M}, (E \cup \{e_k\}) \setminus \tilde{E}) - \Delta(\mathcal{M}, E \setminus (\{e_k\} \cup \tilde{E}))$, is below a tolerance threshold $\lambda$.

**Edge Attribution Patching** (EAP) (Syed et al., 2023) selects $\tilde{E}$ to contain the edges $e_k$ that have the largest gradient approximation of their single-edge ablation loss gap $\Delta(\mathcal{M}, e_k)$.

**HardConcrete Gradient Sampling** (HCGS) is an adaptation of a pruning technique from Louizos et al. (2018) to circuit discovery. Rather than considering only total ablation of an edge $e_k = (\mathcal{A}_j, \mathcal{A}_i, z)$, we can consider a continuous mask of coefficients $\vec{\alpha}$ and partially ablate $e_k$ by replacing the $z$th input to $\mathcal{A}_i$ with a linear combination of the original value $\mathcal{A}_j(x)$ and ablated value $a_j$, i.e. $\alpha_k \mathcal{A}_j(x) + (1 - \alpha_k)a_j$. Now, $\alpha_k = 0$ designates total ablation (replacing with $a_j$), while $\alpha_k = 1$ designates total retention (keeping $\mathcal{A}_j(x)$). We use $\mathcal{M}^{\vec{\alpha}}(x)$ to represent the model with edges partially ablated according to $\vec{\alpha}$.

Some previous work (Liu et al., 2017; Huang and Wang, 2018) optimizes directly on the mask coefficients $\vec{\alpha}$, but to avoid getting stuck in local minima on $\alpha_k \in (0, 1)$, Louizos et al. (2018) samples $\alpha_k$ from a HardConcrete distribution parameterized by location $\theta_k$ and temperature $\beta_k$ for each edge, and performs SGD with respect to the distributional parameters. In effect, we update the parameters based on gradients evaluated at randomly sampled values of $\vec{\alpha}$ rather than gradients evaluated at any exact $\vec{\alpha}$. Cao et al. (2021) applies this technique to find circuits that consist of a subset of model weights. Conmy et al. (2023) applies this technique to vertices in a computational graph. Unlike previous work, we apply this technique to edges rather than vertices.

**Uniform Gradient Sampling** (UGS) is our proposed method for algorithmic circuit discovery. Similar to HCGS, we consider ablation coefficients $\vec{\alpha}$ and update parameters based on gradients evaluated at sampled values of $\vec{\alpha}$. We keep track of a parameter $\tilde{\theta}_k$ for each edge, where $\theta_k = (1 + \exp(-\tilde{\theta}_k))^{-1}$ indicates an estimated probability of $e_k \in \tilde{E}^*$. Using $w(\theta_k) = \theta_k(1 - \theta_k)$ to determine sampling frequency (further discussed in Appendix F.8), we let $\alpha_k \sim \text{Unif}(0, 1)$ with probability (w.p.) $w(\theta_k)$, $\alpha_k = 1$ w.p. $\theta_k - \frac{1}{2}w(\theta_k)$, and $\alpha_k = 0$ w.p. $1 - \theta_k - \frac{1}{2}w(\theta_k)$. For a batch of $b$ inputs $X^{(1)}, ..., X^{(b)}$, let $\vec{\alpha}^{(j)}$ denote the sampled coefficients corresponding to $X^{(j)}$, and let $N_k = \sum_{j=1}^{b} \mathbb{1}(\alpha_k^{(j)} \in (0, 1))$. We construct a loss function $\mathcal{L}_{(\text{UGS})}$ whose gradient satisfies

$$\nabla_{\theta_k} \mathcal{L}_{(\text{UGS})} = \nabla_{\theta_k} \mathcal{R}(\vec{\theta}) + N_k^{-1} \sum_{j=1}^{b} \mathbb{1}(\alpha_k^{(j)} \in (0, 1)) \cdot \nabla_{\alpha_k^{(j)}} \mathcal{L}(\mathcal{M}^{\vec{\alpha}^{(j)}}(X^{(j)}), \mathcal{M}(X^{(j)})) \quad (4)$$

and perform SGD on the $\tilde{\theta}_k$, where $\mathcal{R}(\vec{\theta})$ is a continuous relaxation of $\mathcal{R}(\tilde{E})$ from Equation (3). In Appendix F.5, we motivate UGS as an estimator for sampling over Bernoulli edge coefficients.

**Optimizing circuits on** $\Delta_{\text{opt}}$    ACDC and EAP are not compatible with optimization on $\Delta_{\text{opt}}$, since the optimal $\vec{a}^*$ values depend on the selected circuit and it is intractable to optimize $\hat{a}$ for every candidate circuit. For our circuit evaluations on $\Delta_{\text{opt}}$, we compare to ACDC- and EAP-generated circuits optimized on $\Delta_{\text{CF}}$. On the other hand, HCGS and UGS allow us to perform SGD to optimize the ablation constants $\hat{a}$ concurrently with the sampling parameters.

## 3.2   Experiments

We study GPT-2-small performance on the IOI (Wang et al., 2022) subtask described in Section 2.4 and the Greater-Than (Hanna et al., 2023) subtask, which involves completing prompts such as "The conflict started in 1812 and ended in 18__" with digits greater than the first year in the context. We select these settings because their exposition in manual studies is particularly thorough and they are used in prior work (Conmy et al., 2023; Syed et al., 2023) to benchmark algorithmic circuit discovery.

We compare the algorithms in Section 3.1 trained to minimize $\Delta$ on the IOI and Greater-Than subtasks when edges $E \setminus \tilde{E}$ are ablated with mean ablation, resample ablation, OA, and CF. For IOI, the mapping $\pi$ for CF is defined in Section 2.4. For Greater-Than, we continue with the practice from Hanna et al. (2023) of selecting counterfactuals $\pi(x)$ by changing the first year in the prompt $x$ to end in "01" so that all numerical completions are equally valid (see Appendix F.3).

UGS achieves Pareto dominance on the $\Delta$-$|\tilde{E}|$ tradeoff over the other methods on both subtasks for each ablation method, identifying circuits that achieve lower $\Delta$ at any given $|\tilde{E}|$ and vice versa. Results for IOI circuits optimized on $\Delta_{\text{CF}}$ are shown in Figure 1 (left). On IOI, UGS finds a circuit with 385 edges that achieves $\Delta_{\text{CF}} = 0.220$. This circuit has 52% fewer edges than the smallest ACDC-identified circuit with comparable $\Delta_{\text{CF}}$ and 48% lower $\Delta_{\text{CF}}$ than the best-performing ACDC-identified circuit with a comparable edge count. Similar improvements to the Pareto frontier,

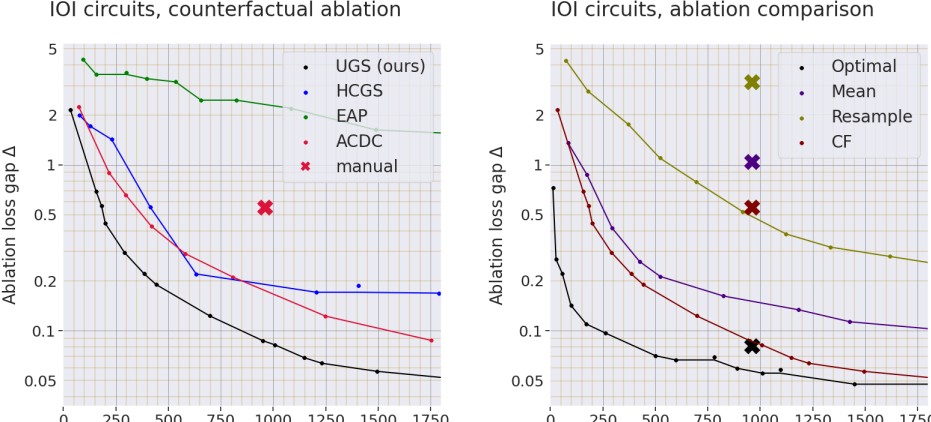

Figure 1: Left: Circuit discovery Pareto frontier for the IOI subtask with counterfactual ablation. Right: Comparison of ablation methods for circuit discovery on IOI (X indicates manual circuit evaluated on each ablation method). $\Delta$ is measured in KL-divergence.

shown in Appendix F.10, occur for mean, resample, and optimal ablation. UGS also creates Pareto improvements for Greater-Than circuits for each ablation method; see Appendix F.11.

Applying OA to circuit discovery reveals that certain sparse circuits can account for model performance on these subtasks to a much greater extent than previously known. We visualize the $\Delta$ for each ablation method achieved by UGS-identified circuits in Figure 1 (right). Using OA to ablate excluded components, we find circuits that recover much lower $\Delta$ at any given circuit size than any circuit for which excluded components are ablated with any other ablation method. For example, for IOI, at a circuit size of 1,000 edges, ablating excluded components with OA enables the existence of circuits with 32% lower $\Delta$ compared to CF, 62% lower $\Delta$ compared to mean ablation, and 88% lower $\Delta$ compared to resample ablation, and the improvement is even larger at smaller circuit sizes. For Greater-Than (results shown in Appendix F.11), OA again admits circuits with by far the lowest $\Delta$ among the four ablation methods. Thus, OA paints a more accurate and compelling picture of how much small subsets of the model can explain behavior on these subtasks.

Unlike other ablation methods, OA indicates that the manual circuits are approximately optimal for their size. Holding $|\tilde{E}|$ fixed, the Pareto-optimal $\Delta_{\mathrm{opt}}$ is 29% below the $\Delta_{\mathrm{opt}}$ of the manual circuit on IOI and 42% below the $\Delta_{\mathrm{opt}}$ of the manual circuit on Greater-Than. However, for the other ablation methods, optimized circuits with fewer edges than the manual circuit achieve 84-85% lower $\Delta$ than the manual circuit on IOI, and 70-84% lower $\Delta$ on Greater-Than. Since the manual circuits are selected using a thorough mechanistic understanding of the model for each subtask and thus arguably capture the important components, this finding furthers the notion that $\Delta$ measured with previous methods could be artificially high due to spoofing by ablated components, and therefore $\Delta_{\mathrm{opt}}$ is a superior evaluation metric for circuits.

These results show that $\Delta_{\mathrm{opt}}$ is useful for evaluating and discovering circuits and provide evidence that OA better quantifies the removal of important mechanisms than previous ablation methods.

## 4   Application: factual recall

Transformers can store and retrieve a large corpus of factual associations. One goal in interpretability is *localizing factual recall*, or identifying components that store specific facts. To this end, Meng et al. (2022) proposes *causal tracing*, which involves removing important information about the prompt $x$ and evaluating which components can recover the original $\mathcal{M}(x)$. To isolate components responsible for an association between a subject (e.g. "Eiffel Tower") and an attribute ("located in Paris"), they select a prompt $x$ ("The Eiffel Tower is located in the city of ___") that elicits from $\mathcal{M}$ a correctly memorized response $y$ ("Paris"). They produce a corrupted input $\xi_{\mathrm{GN}}(x)$ by adding a Gaussian noise (GN) term $Z \sim \mathcal{N}(0, 9\Sigma)$, to all token embeddings that encode the subject, where $\Sigma$ is a diagonal

matrix and $\Sigma_{ii}$ represents the variance of the $i$th neuron among token embeddings sampled from the training distribution. Letting $[\mathcal{M}(x)]_y$ represent the probability assigned by $\mathcal{M}(x)$ to label $y$. Since $\xi_{\mathrm{GN}}$ partially ablates information about the subject, $[\mathcal{M}(\xi_{\mathrm{GN}}(x))]_y$ is typically much smaller than $[\mathcal{M}(x)]_y$. For each component $\mathcal{A}$, they estimate its contribution to the recall of $y$ with the following "average indirect effect" (AIE) representing the proportion of probability on the correct $y$ recovered by replacing $\mathcal{A}(\xi(X))$ with $\mathcal{A}(X)$, averaged over $(X, Y) \sim \mathcal{D}$, where $\xi = \xi_{\mathrm{GN}}$: [4]

$$\mathrm{AIE}(\mathcal{A}) := \min\left(0, 1 - \frac{\mathbb{E}_{(X,Y)\sim\mathcal{D}} \max(0, [\mathcal{M}(X)]_Y - [\mathcal{M}_{\mathcal{A}}(\xi(X), A(X))]_Y)}{\mathbb{E}_{(X,Y)\sim\mathcal{D}}[\mathcal{M}(X)]_Y - \mathbb{E}_{(X,Y)\sim\mathcal{D}}[\mathcal{M}(\xi(X))]_Y}\right) \tag{5}$$

where we declare $\mathrm{AIE}(\mathcal{A}) = 0$ if the denominator is non-positive and ablating subject tokens actually helps identify the correct label (however, this is never the case).

**Method**   We perform causal tracing by removing the subject with optimal ablation (OA-tracing, or OAT) rather than with Gaussian noise (GNT). We define $\xi(x) = \xi_{\mathrm{OA}}(x, a_{\mathcal{A}})$ by replacing subject token embeddings with a constant $a_{\mathcal{A}}$ trained to minimize the numerator in Equation (5), which represents $\Delta$ with a carefully chosen loss function (see Appendix G.2).

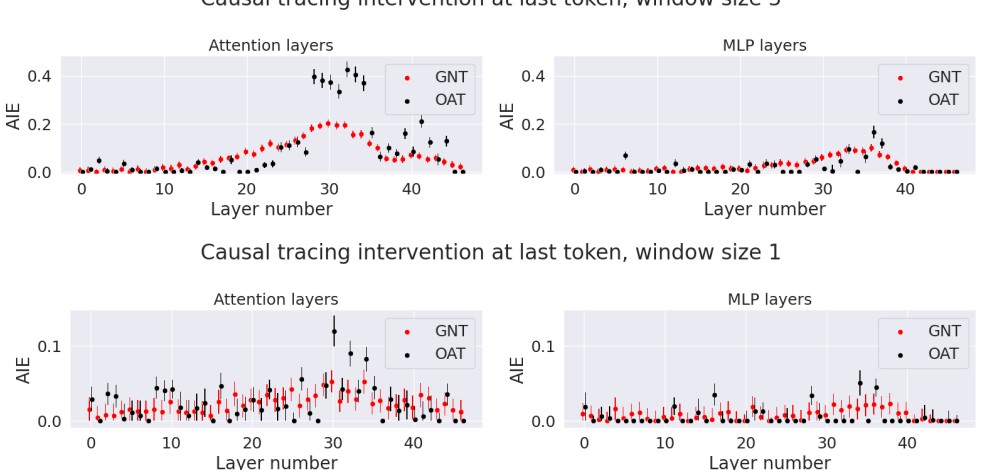

Figure 2: Comparison of AIE with GNT and OAT. In the top figure, layer $\ell$ on the x-axis represents replacing a sliding window of 5 layers with $\ell$ as the median. Error bars indicate the sample estimate plus/minus two standard errors (details given in Appendix G.4).

**Experiments**   We compare GNT and OAT for GPT-2-XL on a dataset of subject-attribute prompts from Meng et al. (2022) for which the model completes the correct answer via sampling with temperature 0. To increase the sample size, we augment the data with similarly constructed prompts from follow-up work on factual recall (Hernandez et al., 2022). We train OAT on 60% of the dataset and evaluate both methods on the other 40%. On the test set, $\mathbb{E}[\mathcal{M}(X)]_Y = 30.6\%$, $\mathbb{E}[\mathcal{M}(\xi_{\mathrm{GN}}(X))]_Y = 12.3\%$, and $\mathbb{E}[\mathcal{M}(\xi_{\mathrm{OA}}(X))]_Y = 8.7\%$. We let $\mathcal{A}(X)$ represent an attention or MLP layer output at a certain token position(s): namely, all subject token positions, only the last subject token position, and only the last token position in the entire sequence. Rather than considering only one layer at a time, Meng et al. (2022) lets $\mathcal{A}$ represent the outputs of a sliding window of several consecutive attention layers or MLP layers. Thus, in addition to replacing the output of a single layer (window size 1), we show results for replacing windows of sizes 5 and 9.

OAT offers a more precise localization of relevant components compared to GNT. While GNT indicates a small positive AIE for most components, OAT shows a few components have large contributions while most have little to no effect. For example, Figure 2 (top left) shows that the AIE for a window of 5 attention layers at the last token is as high as 42.6% for the window consisting of layers 30-34, while the AIE peaks at only 20.2% for GNT. On the other hand, for windows centered around layers 15-23, the average AIE for OAT is only 1.7%, indicating little effect for these potentially

---

[4] Unlike Meng et al. (2022), we clip $[\mathcal{M}(X)]_Y - [\mathcal{M}_{\mathcal{A}}(\xi(X), A(X))]_Y$ to be non-negative, so we do not give $\mathcal{A}$ additional credit for increasing the probability mass of the true label past that given by the full model. We also report AIE in proportion probability recovered compared to the full model rather than percentage points.

unimportant layers, compared to 7.0% for GNT. For sliding windows of 9 attention layers at subject token positions, GNT shows marginally positive AIE measurements across layers 0-30, but OAT specifically shows highly positive AIE for layers 0-5 and 25-30 (see Figure 14). Moreover, whereas Meng et al. (2022) focuses on sliding window replacement because GNT effects from single-layer replacements are very small, OAT can sometimes identify information gain from just one layer. For instance, at the last token position, OAT records AIEs above 8% for each of attention layers 30, 32, and 34 by themselves (see Figure 2, bottom left), much greater than the AIE of the other layers. This greater level of granularity opens up the possibility of selectively investigating combinations of layers as opposed to relying on the prior that adjacent layers work together.

## 5   Application: latent prediction

One practice in interpretability is eliciting predictions from latent representations. Let $\mathcal{M}$ have layers $0, ..., N$ and let $\ell_i(X)$ be the residual stream activation at the last token position (LTP) after layer $i$. *Logit attribution* (Geva et al., 2022; Wang et al., 2022; Dar et al., 2023; Katz and Belinkov, 2023; Dao et al., 2023; Merullo et al., 2024; Halawi et al., 2024) is the practice of applying a transformer model's unembedding map to an activation to obtain a semantic interpretation of that activation. When applied to the LTP activation after layer $i$, this practice is equivalent to zero ablating layers $i+1$ to $N$. However, the semantic meanings of LTP activations after layer $N$ can be different from those of LTP activations in earlier layers. As an alternative, *tuned lens* (Belrose et al., 2023a; Din et al., 2023) is a linear map $f_i(\ell_i) = W_i\ell_i + b_i$ that "translates" from $\ell_i(X)$ to a predicted $\widehat{\ell}_N(X)$. $\mathcal{M}_{\mathrm{TL}}(X)$ is defined by replacing $\ell_N(X)$ with $\widehat{\ell}_N(X) := f_i(\ell_i(X))$ during inference, and training $W_i$ and $b_i$ to minimize $\mathcal{L}_{\mathrm{TL}} := \mathbb{E}_X\mathcal{L}(\mathcal{M}_{\mathrm{TL}}(X), \mathcal{M}(X))$. Tuned lens demonstrates *when* information is transferred to LTP: if replacing $\ell_N(X)$ with $\widehat{\ell}_N(X)$ achieves low loss, then $\ell_i(X)$ contains sufficient context for computing $\mathcal{M}(X)$, so key information is transferred prior to layer $i$.

**Method**   We propose Optimal Constant Attention (OCA) lens. We define $\mathcal{M}_{\mathrm{OCA}}(X)$ by using OA to ablate *attention* layers $i+1$ to $N$: for each of these layers $k$, we replace its output at LTP with a constant $\widehat{a}_k$. We train $\widehat{a} = (\widehat{a}_{i+1}, ..., \widehat{a}_N)$ to minimize $\mathbb{E}_X\mathcal{L}_{\mathrm{OCA}} := \mathbb{E}_X\mathcal{L}(\mathcal{M}_{\mathrm{OCA}}(X), \mathcal{M}(X))$.

Similar to tuned lens, OCA lens reveals whether the LTP activation after layer $i$ contains sufficient context to compute $\mathcal{M}(X)$ by eliminating information transfer from previous token positions to LTP after layer $i$. While tuned lens is a linear map, OCA lens is a function that leverages the model's existing architecture (specifically, its MLP layers) to translate between LTP activations at different layers. OCA lens has far fewer learnable parameters than tuned lens: $O(Nd_{\mathrm{model}}) < O(d_{\mathrm{model}}^2)$.

**Experiments**   We compare $\mathcal{L}_{\mathrm{OCA}}$ to $\mathcal{L}_{\mathrm{TL}}$ for various model sizes. As additional baselines, we also consider the ablation of later attention layers with mean or resample ablation rather than OA. Results are shown in Figure 3 (left) for GPT-2-XL and Figure 15 for other model sizes. OCA lens achieves significantly lower loss than tuned lens, indicating better extraction of predictive power from LTP activations. For example, the predictive loss of OCA lens drops to below 0.01 around layer 35 of GPT-2-XL, but does not reach this point even at the last layer for tuned lens.

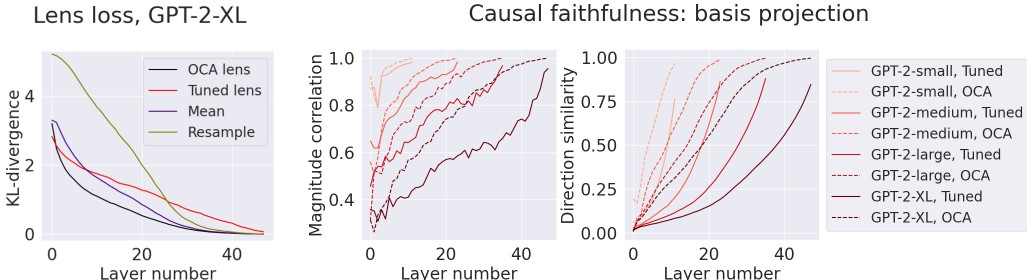

Figure 3: Left: Prediction loss comparison between tuned lens and ablation-based alternatives. Middle, right: Causal faithfulness metrics for tuned and OCA lens under basis-aligned projections.

Additionally, Belrose et al. (2023a) explains that one desiderata for latent prediction is *causal faithfulness*, i.e. $f_i$ should use $\ell_i(X)$ in the same way as $\mathcal{M}$. We can investigate causal faithfulness by intervening on $\ell_i(X)$ and evaluating the extent to which $\mathcal{M}_{\mathrm{TL}}(X)$ and $\mathcal{M}(X)$ move in parallel. If

$\mathcal{M}_{\text{TL}}(X)$ changes significantly but $\mathcal{M}(X)$ does not, for example, then $f_i$ could be extrapolating from spurious correlations, e.g. by inferring from directions that predict information transfer that occurs in later layers. Consider a random intervention $\xi$ on $\ell_i(X)$ and let $\mathcal{M}_{\text{TL}}(X; \xi)$ represent replacing $\ell_i(X)$ with $\xi(\ell_i(X))$ before applying $f_i$. Similarly, let $\mathcal{M}(X; \xi)$ represent replacing $\ell_i(X)$ with $\xi(\ell_i(X))$ during inference. Belrose et al. (2023a) separates causal faithfulness into two measurable properties (both range from -1 to 1 and higher values reflect greater faithfulness):

1. **Magnitude correlation**: $\text{corr}(\mathbb{E}[\mathcal{L}(\mathcal{M}_{\text{TL}}(X; \xi), \mathcal{M}_{\text{TL}}(X)) \mid \xi], \mathbb{E}[\mathcal{L}(\mathcal{M}(X; \xi), \mathcal{M}(X)) \mid \xi])$.
2. **Direction similarity**: $\mathbb{E}[\langle \mathcal{M}_{\text{TL}}(X; \xi) \ominus \mathcal{M}_{\text{TL}}(X), \ \mathcal{M}(X; \xi) \ominus \mathcal{M}(X)\rangle]$, where $\ominus$ denotes subtraction in logit space and $\langle\rangle$ denotes the Aitchinson similarity between distributions.

We assess these properties for $\mathcal{M}_{\text{TL}}$ and $\mathcal{M}_{\text{OCA}}$ for a variety of interventions $\xi$. In Figure 3 (middle, right), we plot these properties for a modified version of the "causal basis projection" $\xi$ from Belrose et al. (2023a). While they train a basis iteratively, this approach is expensive and unstable, and we instead extract an approximate basis for $\mathcal{M}_{\text{TL}}$ by performing singular value decomposition (SVD) on $W_i \Sigma^{1/2}$, where $\Sigma$ is the covariance matrix of $\ell_i(X)$, and applying $\Sigma^{1/2}$ to the right singular vectors. For $\mathcal{M}_{\text{OCA}}$, we extract this basis by training a linear map to approximate $f_i$ and using the weights as the $W_i$. For both lenses, we compute $\xi(a) = \mu + p(a - \mu)$, where $\mu = \mathbb{E}[\ell_i(X)]$ and $p$ represents projecting to the orthogonal complement of $\text{span}(\vec{v})$ for a uniformly sampled basis vector $\vec{v}$. We plot the magnitude correlation and direction similarity for $\mathcal{M}_{\text{TL}}$ and $\mathcal{M}_{\text{OCA}}$ with respect to $\mathcal{M}$ in Figure 3. We find that OCA lens measures significantly better on both causal faithfulness metrics across all layers, and we achieve similar results for other choices of interventions $\xi$ (see Appendix H.3).

One downstream application of extracting latent predictions from intermediate-layer LTP activations is that they can sometimes be more accurate on text classification tasks than the model's output predictions, especially if the context contains false demonstrations, i.e. examples of incorrect task completions (Halawi et al., 2024). The proposed theory is that the model first computes the correct answer at LTP in early layers, then later layers move contextual information to LTP that lead it to make adjustments that benefit next-token prediction, such as reporting an incorrect answer for consistency with false demonstrations. We compare the *elicitation accuracy boost*, or the best elicitation accuracy across layers minus the accuracy of the model output, for OCA lens and tuned lens for GPT-2-XL with 2,000 classification samples from each of 15 text classification datasets from Halawi et al. (2024), using their calibrated accuracy metric. We find that OCA lens increases this accuracy boost for prompts with true demonstrations on 12 of the 15 datasets and for prompts with false demonstrations on 11 of the 15 (see Figure 21). In particular, for Wikipedia topic classification (DBPedia), OCA lens increases the elicitation accuracy boost from 2.9% to 18.0% with true demonstrations and from 19.2% to 28.8% with false demonstrations (see Figure 4, middle). Full results are reported in Appendix H.4.

Elicitation accuracy on selected datasets with 10 demos, GPT-2-XL

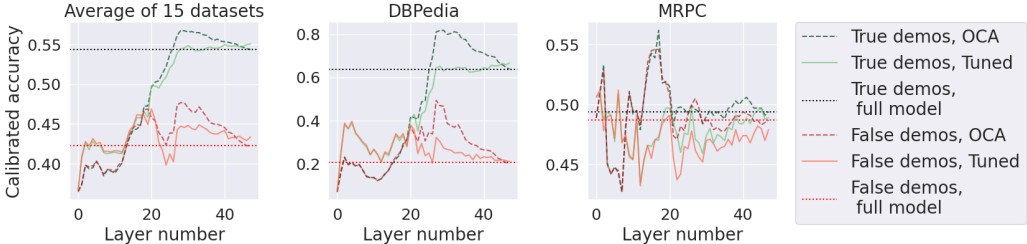

Figure 4: Comparison of calibrated elicitation accuracy on selected datasets.

# 6 Future work

The applications of component importance presented in our work are not exhaustive. A variety of interpretability work either directly applies ablation-based importance or can be framed to use it as a potential tool. OA creates new opportunities to incorporate ablation into studies for which it may be impossible to obtain good results with previous ablation methods. For example, we can train probes derived from using OA with different loss functions (Li et al., 2023), or use an approach similar to OCA lens to decode activations other than the LTP residual stream activation. See Appendix D.3 for an extension of OA to evaluate the extent to which a component performs classification.

## Acknowledgements

LJ was partially supported by DMS-2045981 and DMS-2134157.

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

# A    Limitations

As is the case for previous work, we do not provide an entirely precise definition of the "importance" of a component. The importance of a component can be generally described as an *aggregation of causal effects* in a way that summarizes the component's contribution to model performance. Among the many ways to aggregate causal effects, there may not be a mathematically rigorous way to show that one measure of importance produces the correct or canonical aggregation. However, component importance is useful for a wide variety of applications in interpretability, so aside from showing that our approach to component importance better captures relevant considerations in a conceptual sense, we focus on the utility that it provides to some of these applications.

As noted in Section 2.3, optimal ablation does not entirely eliminate the contribution of information spoofing to $\Delta$. For example, if $\mathcal{A}$ typically conveys strong and exact information about the input, then there may not exist any value of $a$ that hedges between a range of inputs.

Though OA produces circuits that achieve lower loss at a given level of edge sparsity, it may elicit mechanisms that were not previously used for some subtask, especially if there are multiple computational paths that could lead to the same conclusion. However, if the subtask of interest is sufficiently complex, it seems unlikely that a model would have many "dormant" mechanisms that can be repurposed to perform the subtask, because this redundancy wastes computational complexity.

For factual recall, it remains to be seen whether localization is helpful for applications that are further downstream such as producing surgical model edits (Hase et al., 2023; Shah et al., 2024).

# B    Additional related work

As mentioned in Section 2.2, component importance is strongly related to *variable* importance, which quantifies the importance of a model input $X_i$ (also known as a feature or covariate in the variable importance literature).

**Variable importance**    Much of variable importance work concerns "oracle" prediction, which roughly considers how much $X_i$ contributes to the performance of the *best* possible predictor for $Y$ given the set of covariates $(X_1, ..., X_n)$, and frames the importance of $X_i$ as a property of the joint distribution of $(X_1, ..., X_n, Y)$, rather than a property of any particular model used for prediction. Most work in this area analyzes some parametric class of estimators, like linear models (Grömping, 2007; Nathans et al., 2012) or Bayesian networks (Li and Mahadevan, 2017). Later work generalizes parametric variable importance to arbitrary model classes, e.g. by training an ensemble of models that only have access to subsets of the covariates (Strumbelj et al., 2009; Fisher et al., 2019). Recent work has also studied *non-parametric* variable importance, in which we attempt to lower-bound the best performance of any arbitrary estimator (Williamson and Feng, 2020; Zhang and Janson, 2020).

On the other hand, our motivation is to interpret the behavior of one specific model $\mathcal{M}$ (Fisher et al., 2019; Hooker et al., 2019), not to analyze the theoretical relationship between model inputs and outputs. Rather than estimating how well *any* function of $X_i$ can predict $Y$, we wish to estimate how much the *particular* function $\mathcal{M}$ uses an input feature $X_i$ to predict $Y$. Previous work on this "algorithmic" variant of variable importance has taken two main approaches.

**Local function approximations**    One way to quantify how much $X_i$ contributes to model performance is to aggregate *local* function approximations, which approximate the model around a particular input. Common tools for local approximation include the gradient of $\mathcal{M}$ at a given $x$ (Rabitz, 1989; Baehrens et al., 2009; Simonyan et al., 2014; Leino et al., 2018; Nanda, 2023), or a linear function that well-approximates $\mathcal{M}(x + \varepsilon)$ for a chosen noise term $\varepsilon$ (Ribeiro et al., 2016; Smilkov et al., 2017). Since these tools often yield straightforward estimates of the local importance of $X_i$ for the input $x$, one approach to quantifying the *global* importance of $X_i$ is aggregate the importance estimates given by these local approximations, for example by using a first-degree Taylor approximation around a reference input $x_0$ (Bach et al., 2015; Montavon et al., 2017), or integrating over gradients along a straight-line path from $x_0$ to any studied input $x$ (Sundararajan et al., 2017; Dhamdhere et al., 2018). This approach to measuring variable importance works just as well for internal components as it does for inputs. However, local function approximations can fail to capture the overall loss landscape, especially in the common setting where $\mathcal{M}$ has unbounded gradients, and

can often be manipulated to produce arbitrary feature importance values (Slack et al., 2020; Hase et al., 2021).

**Ablation-based measures**  The second main approach considers the *ablation* of feature $X_i$. In this approach, the feature $X_i$ is ablated by replacing it with a different random variable $\overline{X}_i$ that captures less information about the original feature value. We then compare the model performance when $X_i$ is replaced with $\overline{X}_i$ to the original model performance, per the definition of $\Delta$ in Section 2.1 (where the ablated component $\mathcal{A}$ is feature $X_i$ of the model *input*).

Many of the current methods used for ablating internal model components as described in Section 2.2 were first introduced in feature importance work. Zero ablation (Dabkowski and Gal, 2017; Li et al., 2017; Petsiuk et al., 2018; Schwab and Karlen, 2019), mean ablation (Zeiler and Fergus, 2013; Zhou et al., 2015), and Gaussian noise injection (Fong and Vedaldi, 2017; Fong et al., 2019; Guan et al., 2019; Schulz et al., 2020) are all used to remove input features, such as the pixels of an image or tokens of a text input, to assess their importance. Resample ablation is also common in feature importance work; an early variant samples $\overline{X}_i$ from a uniform distribution (Sobol, 1993; Homma and Saltelli, 1996; Strumbelj and Kononenko, 2010), while later work generally performs resample ablation on features by resampling them from their marginal distribution (Breiman, 2001; Robnik-Sikonja and Kononenko, 2008; Datta et al., 2016; Lundberg and Lee, 2017; Janzing et al., 2019; Covert et al., 2020; Kim et al., 2020).

Measuring feature importance via these ablation methods suffers from a well-documented "out-of-distribution" problem (Ishwaran, 2007; Fong and Vedaldi, 2017; Hooker et al., 2019; Hase et al., 2021; Mase et al., 2024): since setting $X_i$ to zero or its mean, resampling $X_i$ from its marginal distribution, or adding Gaussian noise to $X_i$ could result in an input that was never observed during training, the measured feature importance values could potentially be determined by model behavior on impossible and/or nonsensical inputs. One way to mitigate this out-of-distribution problem is replacing feature $X_i$ by a random variable $\overline{X}_i$ sampled from its *conditional* distribution (Strobl et al., 2008; Lundberg et al., 2020), i.e. $\overline{X}_i \sim X_i | X_{-i}$, $(\overline{X}_i \perp\!\!\!\perp X_i) | X_{-i}$, where $X_{-i}$ denotes the other features $X_1, ..., X_{i-1}, X_{i+1}, X_n$. Since the conditional distribution is often intractable to sample from, previous work employs a range of approximation techniques. For example, Zintgraf et al. (2017) samples an ablated pixel from its conditional distribution given a $\ell \times \ell$ patch of its proximate pixels instead of conditioning on the entire image, and Chang et al. (2019) uses a generative model to simulate the conditional distribution.

However, in a setting where the relevant features $X_i$ represent internal model components, rather than inputs to the model, it often does not make sense to discuss an "out-of-distribution" problem because the $(X_1, ..., X_n) = (\mathcal{A}_1(X), ..., \mathcal{A}_n(X))$ are usually near-deterministically related to each other. For example, for any neuron, it is typically the case that its value can be almost deterministically recovered from the values of other neurons in the same layer. Thus, nearly *any* intervention on an internal component $\mathcal{A}_i$ brings the model "out-of-distribution," in the sense that the model observes the vector $(\mathcal{A}_1(X), ..., \mathcal{A}_n(X))$ where $\mathcal{A}_i(X)$ is replaced with $a$ with near-zero probability density.

Our dichotomy of deletion and spoofing in Section 2.1 is more precise than the typical discussion of the out-of-distribution problem in its description of distortions to importance values that we wish to avoid. On one hand, our analysis is more lenient than the blanket requirement that the vector of all internal component values $(\mathcal{A}_1(x), ..., \mathcal{A}_n(x))$ is in-distribution, in the sense that not all interventions that bring $(\mathcal{A}_1(x), ..., \mathcal{A}_n(x))$ out of distribution constitute spoofing; for example, replacing $\mathcal{A}_i(x)$ with $a$ does not have a spoofing-related contribution to $\Delta$ if $\mathcal{A}_i(x)$ and $a$ are equivalent for the sake of downstream computation. On the other hand, our analysis is more stringent in the sense that effect 2a from Section 2.1 is recognized as a form of spoofing that can occur even when interventions are in-distribution (see Appendix D.1 for more details).

**Using dropout to eliminate spoofing**  One way to eliminate spoofing when intervening on $\mathcal{A}(X)$ is to train $\mathcal{M}$ to accept neutral constant values that indicate that component $\mathcal{A}$ has stopped functioning and then replace $\mathcal{A}$ with these built-in neutral values to assess the importance of $\mathcal{A}$. Variations of this technique are common in feature importance (Strumbelj et al., 2009; Chen et al., 2018; Yoon et al., 2018; Hooker et al., 2019). For internal components, we could train neural networks with dropout (Srivastava et al., 2014; Wei et al., 2020), and then use zero ablation to assess the importance of $\mathcal{A}$. Since the downstream computation is trained to recognize $\mathcal{A}(X) = 0$ as an indication that $\mathcal{A}$ carries

no information, as opposed to strong information associated with an input other than the original $X$, $\Delta_{\text{zero}}(\mathcal{M}, \mathcal{A})$ becomes an accurate assessment of deletion (effect 1 from Section 2.1).

However, re-training with neutral values does not necessarily assist in analyzing a particular model $\mathcal{M}$, since re-training $\mathcal{M}$ will in general change $\mathcal{M}$ itself. Furthermore, training with dropout incentivizes $\mathcal{M}$ to lower $\Delta_{\text{zero}}(\mathcal{M}, \mathcal{A})$ for any component $\mathcal{A}$, since part of the loss function involves minimizing loss with a random subset of ablated components. As a result, we expect to observe more redundant computation shared between many model components, since a random subset of them could be ablated during training. This redundancy inherently tends to make $\mathcal{M}$ less modular and harder to analyze – for example, we should expect a broad variety of components to perform relevant computation for any input, even if an accurate prediction could be computed with just a few components, so it becomes difficult to localize model behaviors. Since interpretability involves decomposing model computation into smaller pieces and identifying specialization among model components, models trained with dropout may be less interpretable. To summarize, while the $\Delta_{\text{zero}}$ measurements are more "accurate" when $\mathcal{M}$ is trained with dropout, they may become less "useful" for interpretability. On the other hand, OA makes $\Delta$ measurements a more accurate reflection of deletion effects without training $\mathcal{M}$ on to distort the magnitude of these effects.

**Aggregation mechanisms for ablation methods**   On top of a selected ablation method, some work uses Shapley values to aggregate performance gap measurements for sets of features (Strumbelj et al., 2009; Strumbelj and Kononenko, 2010; Datta et al., 2016; Lundberg and Lee, 2017; Janzing et al., 2019; Lundberg et al., 2020; Covert et al., 2020). This line of work measures the importance of $X_i$ by estimating a weighted average of the performance gap $\Delta(\mathcal{M}, S)$ for all subsets $S \subset \{X_1, ..., X_n\}$ rather than considering only $\Delta(\mathcal{M}, X_i)$. This aggregation mechanism is applied after choosing an ablation method via which to measure each $\Delta$, and is just as compatible with OA as with any other ablation method.

**Sparse pruning and masking**   Finally, in the literature of sparse pruning and masking, an operation that is procedurally similar to optimal ablation is sometimes performed in prior work by adding a bias term to removed features or activations after setting weights to zero.

In some structured pruning work, it is typical to introduce scaled batch normalization layers $\tilde{\mathcal{A}}(X) = \gamma \mathcal{A}(X) + \beta$ for $\gamma \in [0, 1]$ to the output of each computational block $\mathcal{A}$, and regularize the $\gamma$ toward zero to select weights to prune (Liu et al., 2017; Ye et al., 2018; Zhuang et al., 2020). When $\gamma$ reaches 0, the output of $\mathcal{A}$ is set to the constant $\beta$, which is trained to minimize the loss of the pruned model. However, the motivation of this reparameterization is not to measure component importance, and optimal ablation can be applied to more general model components (e.g. computational edges in any graph representation).

Similar to pruning, sparse masking work searches for a mask over input tokens such that for any input, most inputs are zeroed out while model performance is retained (Li et al., 2017; De Cao et al., 2021; Chen et al., 2021; Schlichtkrull et al., 2022). In particular, De Cao et al. (2021) replaces masked tokens in an input $X = X_1, ..., X_n$ with a learned bias $b(X)$. While this operation may seem similar to optimal ablation, a fundamental difference is that the bias $b(X)$ is different for each input sequence $X$ and is trained to equalize embeddings at different token positions in a single $X$ rather than assuming the same constant value for all values of $X$. Thus, for each $X$, $b(X)$ contains specific information about the masked tokens in $X$, so unlike OA, this technique does not perform total ablation on the masked tokens. A follow-up work Schlichtkrull et al. (2022) trains a common $b$ for a dataset of inputs $X$. However, Schlichtkrull et al. (2022) uses an auxiliary linear model $\phi_i(\mathcal{A}_1(X), ..., \mathcal{A}_n(X))$ to predict whether a component $\mathcal{A}_i(X)$ should be masked. Since $\phi_i$ explicitly depends on the values of the masked components $\mathcal{A}_i(X)$, the model output remains dependent on information contained in $\mathcal{A}_i(X)$, and total ablation is not achieved. Moreover, the auxiliary model $\phi$ provides the model with additional computation to distill information about the input, rather than strictly reducing the computational complexity of the original model as OA does. The use of an auxiliary model is a requisite feature of their method and cannot be decoupled from the masking technique: without using $\phi(X)$ to predict the masked components, computing masks requires a separate optimization procedure for each input, which makes it computationally intractable to optimize a single $b$ over an entire distribution of inputs.

# C  Additional preliminaries

## C.1  Models as computational graphs

We can write an ML model $\mathcal{M}$ as a connected directed acyclic graph. The graph's source vertices represent the model's (typically vector-valued) input, its sink vertices represent the model's output, and intermediate vertices represent units of computation. For the sake of simplicity, assume $\mathcal{M}$ has a single input and a single output. Each intermediate vertex $\mathcal{A}_i$ represents a computational block that takes in the values of previous vertices evaluated on $x$, and itself produces an output $\mathcal{A}_i(x)$, that is taken as input to later vertices. We indicate that there exists a directed edge from vertex $\mathcal{A}_j$ to vertex $\mathcal{A}_i$ if $\mathcal{A}_j(x)$ is taken as input to $\mathcal{A}_i$.

Let $\mathcal{M}$ be represented by computational graph $(G, E)$ where $G$ is the overall set of vertices and $E$ be the set of edges. Let $\mathcal{A}_{0:n}$ be a tuple representing $G$ in topologically sorted order ($\mathcal{A}_0$ represents the model input, while $\mathcal{A}_n$ represents the model output). For a particular vertex $\mathcal{A}_i$, let $\vec{G}^i = (G_1^i, ..., G_k^i)$ be the tuple of vertices (duplicates allowed) whose outputs $\mathcal{A}_i$ takes as immediate inputs. As we will see, we will sometimes require multiple edges between a pair of vertices. Rather than the standard edge notation $e = (\mathcal{A}_j, \mathcal{A}_i) \in E$ for simple graphs, we adopt the notation $e = (\mathcal{A}_j, \mathcal{A}_i, z) \in E$ to indicate that $G_z^i = \mathcal{A}_j$, i.e. $\mathcal{A}_j(x)$ is taken as the $z$th input to $\mathcal{A}_i$.

Model inference is performed by evaluating the vertices in topologically sorted order. We perform inference on an input $x$ by setting $\mathcal{A}_0(x) = x$ and then iteratively evaluating $\mathcal{A}_i(x) = \mathcal{A}_i(\vec{G}^i)$ for $i \in \{1, ..., n\}$. By the time we evaluate some vertex $\mathcal{A}_i$, we have already computed the values $G_z^i(x)$ for each of its inputs because they precede $\mathcal{A}_i$ in the topological sort. Finally, we determine that $\mathcal{M}(x) = \mathcal{A}_n(x)$. We will alternate between the notation $\mathcal{A}_i(\vec{G}^i)$ to explicitly write $\mathcal{A}_i$ as a function of its immediate inputs and the notation $\mathcal{A}_i(x)$ to indicate that the output of $\mathcal{A}_i$ is a function of $x$. We also sometimes use $\mathcal{A}_i(x)$ as a standalone quantity apart from evaluating $\mathcal{M}(x)$ and observe that this quantity is a function of $x$ computed by evaluating $\mathcal{A}_j(x)$ in order for $j \in \{1, ..., i\}$.

The graph notation for any ML model is not unique. For any model, there are many equivalent graphs that faithfully represent its computation. In particular, a computational graph can represent a model at varying levels of detail. At one extreme, intermediate vertices can designate individual additions, multiplications, and nonlinearities. Such a graph would have at least as many vertices as model parameters. Fortunately, most model architectures have self-contained computational blocks, which allows them to be represented by graphs that convey a significantly higher level of abstraction. For example, in convolutional networks, intermediate vertices can represent convolutional filters and pooling layers, while in transformer models, the natural high-level computational units are attention heads and multi-layer perceptron (MLP) modules.

## C.2  Activation patching

Activation patching is the practice of evaluating $\mathcal{M}(x)$ while performing the intervention of setting some component $\mathcal{A}_i$ to a counterfactual value $a$ instead of $\mathcal{A}_i(x)$ during inference. We use the notation $\mathcal{M}_{\mathcal{A}_i}(x, a)$ extensively in the paper to indicate this practice, and here we give a more precise definition in terms of $\mathcal{M}$ as a computational graph:

**Definition C.1** (Vertex activation patching). *To compute $\mathcal{M}_{\mathcal{A}_i}(x, a)$, compute $\mathcal{A}_0(x), ..., \mathcal{A}_{i-1}(x)$ in normal fashion and set $\mathcal{A}_i(x) = a$. Then compute each vertex $\mathcal{A}_{i+1}(x), ..., \mathcal{A}_n(x)$ in order, computing each vertex $\mathcal{A}_j$ as a function of its immediate inputs, i.e. $\mathcal{A}_j(x) = \mathcal{A}_j(G_1^j(x), ..., G_k^j(x))$. Finally, return $\mathcal{M}_{\mathcal{A}_i}(x, a) = \mathcal{A}_n(x)$.*

During this modified forward pass, a vertex $\mathcal{A}_j$ that takes $\mathcal{A}_i$ as its $z$th immediate input, i.e. for which $(\mathcal{A}_i, \mathcal{A}_j, z) \in E$, instead takes $a$ as their $z$th input instead of the normal value of $\mathcal{A}_i(x)$. Later, if some other vertex takes $\mathcal{A}_j$ as input, it will take this modified version of $\mathcal{A}_j(x)$ as input, and so on, so the intervention on $\mathcal{A}_i$ may have an effect that carries through the graph computation and eventually makes $\mathcal{M}_{\mathcal{A}_i}(x, a)$ different from $\mathcal{M}(x)$.

In Section 3, we discuss extending this practice to edges $e$:

**Definition C.2** (Edge activation patching). *To compute $\mathcal{M}_e(x, a)$, where $e = (\mathcal{A}_j, \mathcal{A}_i, z)$, compute $\mathcal{A}_0(x), ..., \mathcal{A}_{i-1}(x)$ in normal fashion. Set $\mathcal{A}_i(x) = (G_1^i(x), ..., G_{z-1}^i(x), a, G_{z+1}^i(x), ..., G_k^i(x))$,*

*i.e. setting its zth input to $a$ instead of $\mathcal{A}_j(x)$. Then compute each vertex $\mathcal{A}_{i+1}(x), ..., \mathcal{A}_n(x)$ in order as a function of its immediate inputs. Finally, return $\mathcal{M}_e(x, a) = \mathcal{A}_n(x)$.*

As mentioned in the main text, using activation patching on a particular edge $e = (\mathcal{A}_j, \mathcal{A}_i, z)$ is more surgical than using activation patching on its parent vertex $\mathcal{A}_j$. Performing activation patching on $\mathcal{A}_j$ would replace $\mathcal{A}_j(x)$ with $a$ as an input to *all* of its child vertices, but performing activation patching on only $e$ modifies $\mathcal{A}_j(x)$ only as an input to $\mathcal{A}_i$. Notice that performing activation patching on $\mathcal{A}_j(x)$ is equivalent to performing activation patching on $e = (\mathcal{A}_j, \mathcal{A}_i, z)$ for *all* edges $e$ that emanate from $\mathcal{A}_j$ in the graph.

### C.3 Transformer architecture

The transformer architecture (Vaswani et al., 2017) may be familiar to most readers. However, since our experiments involve interventions during model inference with varying levels of granularity, we include a summary of the transformer computation, which we later reference to crystallize specifically how we edit the computation.

Transformers $\mathcal{M}$ take in token sequences $x_{1:s}$ of length $s$, which are then prepended with a constant padding token $x_0$. Let $x = (x_j)_{i=0}^s$. The model simultaneously computes, for each token position $i$, a predicted probability distribution $(\widehat{\mathbb{P}}(x_{j+1} \mid x_{0:j}))_{j=0}^s$ for the $(j+1)$th token given the first $j$ tokens. We use $\mathcal{M}(x)$ to refer to the predicted probability distribution over the $(s+1)$th token. We sometimes abuse notation to write $\mathbb{P}(\mathcal{M}(x) = y)$ to indicate $\widehat{\mathbb{P}}(y \mid x)$, i.e. the probability placed on prediction $y$ by the distribution $\mathcal{M}(x)$.

Let $\tilde{X}$ be a random input sequence and $S$ be a token position sampled randomly from $\{1, ..., s\}$. $\mathcal{M}$ is trained to minimize $-\mathbb{E}_{X,S} \log \mathbb{P}(\mathcal{M}(\tilde{X}_{0:S-1}) = \tilde{X}_S)$. However, we generally refer to input-label pairs $(X, Y) = (\tilde{X}_{0:S-1}, \tilde{X}_S)$, so that the loss function is instead written

$$\mathbb{E}_{X,Y} \mathcal{L}(\mathcal{M}(X), Y) = \mathbb{E}_{X,Y} \left[ -\log \mathbb{P}(\mathcal{M}(X) = Y) \right]$$

To evaluate $\mathcal{M}$, each token $x_j$ is mapped to a embedding $\mathrm{Resid}_j^{(0)}(x) = t(x_j) + p(j)$ of dimension $d_{\mathrm{model}}$, where $t(x_j)$ is a token embedding of token $x_j$ and $p(j)$ is a position embedding representing position $j$ in the sequence. Over the course of inference, $\mathcal{M}$ keeps track of a "residual stream" representation $\mathrm{Resid}_j^{(i)}$ at each token position $j$ that is a vector of dimension $d_{\mathrm{model}}$, which it updates by iterating over its $n_{\mathrm{layers}}$ layers, adding each layer's contribution to the previous representation:

$$\mathrm{MResid}_j^{(i)}(x) = \mathrm{Resid}_j^{(i-1)}(x) + \sum_{k=1}^{n_{\mathrm{heads}}} \mathrm{Attn}_j^{(i,k)}(x) \tag{6}$$

$$\mathrm{Resid}_j^{(i)}(x) = \mathrm{MResid}_j^{(i)}(x) + \mathrm{MLP}_j^{(i)}(x). \tag{7}$$

Attention heads $\mathrm{Attn}^{(i,k)}$ transfer information between token positions. Let LN (layer-norm) be the function that takes a matrix $Z$ of size $m \times n$ and outputs a matrix of the same size such that each row of $\mathrm{LN}(Z)$ is equal to the corresponding row of $Z$ normalized by its $L_2$ norm: $(\mathrm{LN}(Z))_j = \frac{Z_j}{||Z_j||_2}$. Let $R = \mathrm{LN}(\mathrm{Resid}^{(i-1)}(x))$. Attention heads are computed as follows:

$$\mathrm{AttnScore}^{(i,k)}(x) = \mathrm{softmax}\left( \triangle \cdot \underbrace{\left( RW_Q^{(i,k)} + b_Q^{(i,k)} \right)}_{Q_{(i,k)}} \underbrace{\left( RW_K^{(i,k)} + b_K^{(i,k)} \right)^T}_{K_{(i,k)}^T} \right)$$

$$\mathrm{Attn}^{(i,k)}(x) = \mathrm{AttnScore}^{(i,k)}(x) \underbrace{\left( RW_V^{(i,k)} + b_V^{(i,k)} \right)}_{V_{(i,k)}} W_O^{(i,k)} + b_O^{(i,k)}. \tag{8}$$

The $W^{(i,k)}$s and $b^{(i,k)}$s are weights. $W_Q^{(i,k)}$, $W_K^{(i,k)}$, and $W_V^{(i,k)}$ have size $d_{\mathrm{model}} \times d_{\mathrm{head}}$, and $W_O^{(i,k)}$ has size $d_{\mathrm{head}} \times d_{\mathrm{model}}$, $b_Q^{(i,k)}$, $b_K^{(i,k)}$, and $b_V^{(i,k)}$ have dimension $d_{\mathrm{head}}$ while $b^{(i,k)}$ has dimension $d_{\mathrm{model}}$. Biases are added to each row. $\triangle$ is a lower triangular matrix of 1s, $\cdot$ represents the elementwise product and the softmax is performed row-wise. Multiplying by $\triangle$ ensures that $\mathrm{Attn}_j^{(i,k)}(x)$ only

depends on $\text{Resid}_{0:j}^{(i)}$ and thus information can only be propagated forward, so the prediction of token $j + 1$ can only depend on tokens 0 through $j$.

MLP layers are computed token-wise; the same map is applied to $\text{Resid}_j^{(i-1)}(x)$ at each token position $j$. Let $R = \text{LN}(\text{MResid}^{(i)}(x))$, and let $R_j$ be the $j$th row of $R$. MLPs are computed as follows:
$$\text{MLP}_j^{(i)}(x) = \text{ReLU}\left(R_j W_{\text{in}}^{(i)} + b_{\text{in}}^{(i)}\right) W_{\text{out}}^{(i)} + b_{\text{out}}.$$

The $W^{(i)}$ and $b^{(i)}$ are weights. $W_{\text{in}}^{(i)}$ has shape $d_{\text{model}} \times d_{\text{mlp}}$ and $W_{\text{out}}^{(i)}$ has shape $d_{\text{mlp}} \times d_{\text{model}}$. $b_{\text{in}}^{(i)}$ has dimension $d_{\text{mlp}}$ and $b_{\text{out}}^{(i)}$ has dimension $d_{\text{model}}$.

Finally, the output probability distribution is determined by applying a final transformation to the residual stream representation after the last layer.

$$\text{Out}(x) := \text{softmax}\left(\text{LN}\left(\text{Resid}^{(n_{\text{layers}})}(x)\right) W_{\text{unembed}}\right)$$

$W_{\text{unembed}}$ is a learnable weight matrix of size $d_{\text{model}} \times d_{\text{vocab}}$ and the softmax is performed row-wise. $\text{Out}(x)$ is a matrix of size $(s + 1) \times d_{\text{vocab}}$, where $d_{\text{vocab}}$ is the number of tokens in the vocabulary and each row $\text{Out}_j(x)$ indicates a discrete probability distribution over $d_{\text{vocab}}$ values that predicts the $(j + 1)$th token given the first $j$ tokens. $\mathcal{M}(x)$ is the prediction for the $(s + 1)$th-token continuation of $x$ given the entire sequence $x$, i.e. $\mathcal{M}(x) = \text{Out}_s(x)$.

## C.4 KL-divergence loss function

The performance metric $\mathcal{P}(\tilde{\mathcal{M}}) = \mathbb{E}_{X \sim \mathcal{D}} \, \mathcal{L}(\tilde{\mathcal{M}}(X), \mathcal{M}(X))$ selected in the paper frames performance in terms of proximity to the original model predictions, and thus the corresponding ablation loss gap $\Delta(\mathcal{M}, \mathcal{A})$ measures the importance of component $\mathcal{A}$ for the model to arrive at predictions that are close to its original predictions. A common alternative, $\tilde{\mathcal{P}}(\tilde{\mathcal{M}}) = \mathbb{E}_{(X,Y) \sim \mathcal{D}} \mathcal{L}(\tilde{\mathcal{M}}(X), Y)$, frames performance in terms of proximity to the true *labels*, so the corresponding ablation loss gap $\tilde{\Delta}(\mathcal{M}, \mathcal{A})$ measures the importance of component $\mathcal{A}$ for the model to perform a subtask at a comparable level to the original model. As an example, consider a model $\mathcal{M}$ that computes an approximately-optimal solution $\tilde{\mathcal{M}}(X)$ and then adds noise in a way that changes its predictions but does not improve or worsen $\mathbb{E}_{X,Y} \mathcal{L}(\mathcal{M}(X), Y)$. Presenting $\tilde{\mathcal{M}}$ alone is a satisfactory interpretation of the behavior of $\mathcal{M}$ under $\tilde{\mathcal{P}}$ but not under $\mathcal{P}$.

A major advantage of $\mathcal{P}$ is that it is much more sample-efficient to evaluate for language tasks, especially if the label distribution has high entropy. Let $(X, Y)$ denote a random input-label pair. Recall that a language model is trained to minimize

$$\mathbb{E}_{X,Y} \mathcal{L}(\mathcal{M}(X), Y) = \mathbb{E}_{X,Y} \left[-\log \mathbb{P}(\mathcal{M}(X) = Y)\right] = c + \mathbb{E}_X D_{KL}(\rho(X) \,||\, \mathcal{M}(X))$$

where $c$ is a constant and $\rho(X)$ represents the true probability distribution of $Y|X$. For each $X$, we are unable to observe $\rho(X)$ – in fact, we are usually only able to obtain a single sample from $Y \sim \rho(X)$. On the other hand, $\mathcal{M}(X)$ may be a sufficient estimate for $\rho(X)$, and provides many more bits of information about $\rho$ than the single sample $Y \sim \rho$. Even if our desired performance metric were $\tilde{\mathcal{P}}$, rather than estimating $\mathbb{E}_X[\mathbb{E}_Y[\mathcal{L}(\tilde{\mathcal{M}}(X), Y) \mid X]]$ from individual samples $(X, Y)$, it may often be more sample-efficient to approximate $\mathbb{E}_Y[\mathcal{L}(\tilde{\mathcal{M}}(X), Y) \mid X]$ analytically for a particular $X$ by assuming that the full model well-approximates that true distribution $\rho$, i.e. by assuming that $\mathcal{M}(X) \approx \rho(X)$ (in the sense that $D_{KL}(\rho(X) \,||\, \mathcal{M}(X)) \approx 0$), which implies

$$\mathbb{E}_X D_{KL}(\rho(X) \,||\, \tilde{\mathcal{M}}(X)) \approx \mathbb{E}_X D_{KL}(\mathcal{M}(X) \,||\, \tilde{\mathcal{M}}(X)). \tag{9}$$

and so we still evaluate $\tilde{\mathcal{M}}$ using $\mathcal{P}$ as the performance metric.

In practice, Equation (9) may be an unreasonable assumption and the two criteria may yield very different interpretability results. We cannot estimate $\mathbb{E}_{X,Y} \mathcal{L}(\mathcal{M}(X), Y)$ because it is impossible to obtain an estimate of the ground truth entropy of next-token prediction – for long sequences, we typically never observe the same sequence more than once. However, we can deduce a lower bound $\mathbb{E}_{X,Y} \mathcal{L}(\mathcal{M}(X), Y) \geq 1$ for a model like GPT-2 because larger models reduce cross-entropy by more than this amount compared to GPT-2. Note that a better approximation of $\mathbb{E}_{X,Y} \mathcal{L}(\tilde{\mathcal{M}}(X), Y)$ is to obtain a probability distribution from a larger language model $\mathcal{M}^*$, and future work may wish to explore this direction.

However, there are several other reasons to prefer using $\mathcal{P}$ over $\tilde{\mathcal{P}}$ with labels from a larger model. The use of KL-divergence to the original model $\mathcal{M}$ is consistent with previous work. In the real-world scenario of performing interpretability on the largest frontier model, we will not have access to a better $\mathcal{M}^*$. Most importantly, one concern with circuit discovery for subtasks $(X, Y) \sim \mathcal{D}$ is that it may be possible to adversarially select $\tilde{\mathcal{M}}$ such that

$$\mathbb{E}_{(X,Y)\sim\mathcal{D}}\mathcal{L}(\tilde{\mathcal{M}}(X), Y) \leq \mathbb{E}_{(X,Y)\sim\mathcal{D}}\mathcal{L}(\mathcal{M}(X), Y). \tag{10}$$

which can occur if $\mathcal{M}$ sacrifices some performance on $(X, Y) \sim \mathcal{D}$ for better performance on other regions of the input distribution. Selecting only the components of $\mathcal{M}$ that maximize performance on $\mathcal{D}$ may ignore important mechanisms that must be included in its predictions on $\mathcal{D}$ as a result of this tradeoff. On the other hand, evaluating circuits with $\tilde{\mathcal{P}}$ llows mitigating mechanisms to be included in the selected circuit, since we must select $\tilde{\mathcal{M}}$ in a way that imitates the behavior of $\mathcal{M}$ itself on the subtask $\mathcal{D}$. Using this metric, a subnetwork can never achieve lower loss than $\mathcal{M}$, since $\mathcal{L}(\tilde{\mathcal{M}}(X), \mathcal{M}(X)) \geq \mathcal{L}(\mathcal{M}(X), \mathcal{M}(X))$.

## D  Commentary

### D.1  Understanding the difference between deletion and treating $x$ like $x'$

Colloquially, "deletion" means the model has lost the information it would use to distinguish between inputs $x$ and $x'$. One might expect that if the model were able to rationally handle this lack of information, it would produce an output that hedges between labels corresponding to inputs $x$ and $x'$. On the other hand, subclass 2a of "spoofing" means the model was given information in component $\mathcal{A}$ that is compatible with $x'$ and not $x$, leading the model to output something close to what it would have produced on input $x'$.

To illustrate the difference between deletion and insertion, consider the following example. Assume a classifier $\mathcal{M}$ has two possible labels and two possible inputs, $x$ and $x'$, and the model entirely depends on component $\mathcal{A}$ to determine the correct label. Let $\mathcal{M}$ output a probability vector, and suppose $\mathcal{L}$ is KL-divergence. Let $\mathcal{M}(x) = (1, 0)$ and $\mathcal{M}(x') = (0, 1)$. If we remove the information given by $\mathcal{A}$, we should expect the model to output $(0.5, 0.5)$, giving $\mathcal{L} = \log 0.5$, but if we instead intervene by inserting $\mathcal{A}(x')$ into an inference pass on $x$ or vice versa, then the model places probability 1 on the incorrect label, and the loss is infinite from assessing that the input is $x'$ when the true input is $x$.

### D.2  OA as an extension of mean ablation for nonlinear functions

Let $\mathcal{A}(X)$ be a vector-valued model component. As noted in Lundberg and Lee (2017), one motivation for mean ablation is that $\mathbb{E}[\mathcal{A}(X)]$ is, under certain assumptions, a reasonable point estimate for $\mathcal{A}(X)$. For instance, if the relevant loss function is the squared distance between our point estimate $a$ and the realized value of $\mathcal{A}(X)$, then $\mathbb{E}[\mathcal{A}(X)] = \arg\min_a \mathbb{E}_X ||a - \mathcal{A}(X)||_2^2$. Indeed, the mean is also the best point estimate of $\mathcal{A}(X)$ if the relevant loss is squared distance between $\mathcal{M}_{\mathcal{A}}(X, a)$ and $\mathcal{M}(X) = \mathcal{M}_{\mathcal{A}}(X, \mathcal{A}(X))$ and the model $\mathcal{M}$ is *linear* in $\mathcal{A}(X)$:

$$\mathbb{E}[\mathcal{A}(X)] = \arg\min_a \mathbb{E}_X ||\mathcal{M}_{\mathcal{A}}(X, a) - \mathcal{M}_{\mathcal{A}}(X, \mathcal{A}(X))||_2^2 \tag{11}$$

if $\mathcal{M}(X, a) = M(X)a + b(X)$ for a random matrix $M(X) \perp\!\!\!\perp \mathcal{A}(X)$ and random bias $b(X)$.

Thus, for model components $\mathcal{A}(X)$ for which the downstream computation is roughly linear, $\mathbb{E}[\mathcal{A}(X)]$ could potentially be a reasonable point estimate, hence justifying mean ablation. This presumption of linearity also shows up in other interpretability work, including Hernandez et al. (2024), which uses a linear map to approximate the decoding of subject-attribute relations, and Belrose et al. (2023b), which considers erasing concepts $Z$ from a model's latent space in a "minimal" sense by transforming activations $\mathcal{A}(X)$ with a map $g_Z$ that makes $g_Z(\mathcal{A}(X))$ uncorrelated with $Z$ and minimizes expected squared distance to the original activations, $\mathbb{E}[g_Z(\mathcal{A}(X)), \mathcal{A}(X)]$.

However, in most settings, $\mathcal{M}(X, a)$ is highly nonlinear in $a$, and the mean $\mathbb{E}[\mathcal{A}(X)]$ could be an arbitrarily poor point estimate for $\mathcal{A}(X)$. Optimal ablation generalizes the idea of selecting the "best point estimate" for $\mathcal{A}(X)$ as measured by replacing $\mathcal{A}(X)$ with $a$ and evaluating model loss. In particular, optimal ablation constants $a^*$ generalize the property given in Equation (11) to arbitrary

models $\mathcal{M}$ and loss functions $\mathcal{L}$:

$$a^* = \arg\min_a \mathcal{L}(\mathcal{M}_{\mathcal{A}}(X, a), \mathcal{M}_{\mathcal{A}}(X, \mathcal{A}(X))). \tag{12}$$

## D.3 Generalizing OA to constrained-form estimates of $\mathcal{A}(X)$

Measuring $\Delta_{\text{opt}}(\mathcal{M}, \mathcal{A})$ on a subtask $(X, Y) \sim \mathcal{D}$ is, in a sense, a testing procedure for the hypothesis that "$\mathcal{A}$ does not provide relevant information for model performance on subtask $\mathcal{D}$." Verifying that $\Delta_{\text{opt}} \approx 0$ validates this hypothesis, since a point estimate of $\mathcal{A}(X)$ performs as well as the realized value of $\mathcal{A}(X)$ for the purpose of model inference.

Optimal ablation can be generalized to test interpretability hypotheses beyond assertions that a computed quantity $\mathcal{A}(X)$ is unimportant. In particular, we can test hypotheses about the specific properties of $\mathcal{A}(X)$ that are important.

Suppose $\mathcal{A}(X)$ is vector-valued, and consider the hypothesis "the only relevant information in $\mathcal{A}(X)$ is stored in subspace $W$." We can test this hypothesis by replacing $\mathcal{A}(X)$ with $P_W \mathcal{A}(X) + a^*$, where $a^*$ is an optimal constant that lies in $W^\perp$ and $P_W$ is the projection matrix to subspace $W$. While this example is simple and illustrates some of the flexibility of OA, it does not add to the space of what OA can express, in the sense that we could have simply considered $P_W \mathcal{A}(X)$ and $P_{W^\perp} \mathcal{A}(X)$ as separate vertices in the graph and used OA (or any other ablation method) on only the latter.

However, a real gain of expression from OA materializes from being able to generalize the idea of null point estimates to estimates with constrained form. For example, consider the *subspace* hypothesis "every $\mathcal{A}(X)$ can be adequately represented in subspace $W$." We can test this hypothesis by training an optimal activation $a^*(X)$ for each $X$ that lies in subspace $W$. Though activation training is expensive, we can train a function $a^*(X)$ that maps $X$ to values in $W$, and then estimate the error of this function by performing activation training on a few samples of $X$.

Similarly, we can generalize $a^*$ to include multiple point estimates to test the claim that $\mathcal{A}(X)$ is the outcome of an internal *classification* problem, i.e. the relevant information provided by $\mathcal{A}(X)$ is the classification of $X$ among a few input classes. We can train optimal point estimates $(a_1^*, ..., a_k^*)$ such that

$$(a_1^*, ..., a_k^*) = \arg\min_{(a_1, ..., a_k)} \mathbb{E}_X \min_{j \in \{1, ..., k\}} \mathcal{L}(\mathcal{M}(X, a_j), Y)$$

calling the outer minimized quantity $\Delta_{\text{k-opt}}$. If $\Delta_{\text{k-opt}} \approx 0$, then every $\mathcal{A}(X)$ can be represented by one of a small number of prototype quantities.

# E  Single-component loss on IOI

## E.1  Transformer graph representation

We use a graph representation in which each vertex corresponds to an attention head ($\text{Attn}^{(i,k)}(x)$), an MLP block ($\text{MLP}^{(i)}(x)$), the model input ($\text{Resid}^{(0)}(x)$), or the model output ($\text{Out}(x)$). We also allow vertices representing the $\text{Resid}^{(i)}(x)$ and $\text{MResid}^{(i)}(x)$ computations. Appendix C.3 defines the computation of each vertex.

However, we slightly modify the definition of attention head vertices $\text{Attn}^{(i,k)}$ to save memory and so that ablation constants $a^*$ for OA lie in the column space of attention head outputs. Recall from Equation (8) that attention heads produce output in a $d_{\text{head}}$-dimensional vector space, which is then mapped linearly to $d_{\text{model}}$-dimensional space by a weight matrix $W_O^{(i,k)}$:

$$\text{Attn}^{(i,k)}(x) = \text{AttnScore}^{(i,k)}(x)\left(RW_V^{(i,k)} + b_V^{(i,k)}\right)W_O^{(i,k)} + b_O^{(i,k)}$$

Thus, while $\text{Attn}^{(i,k)}(x)$ is $d_{\text{model}}$-dimensional, its distribution lies within a $d_{\text{head}}$ subspace of the residual stream. If we used vertices $\text{Attn}^{(i,k)}$, then our $d_{\text{model}}$-dimensional $a^*$ for attention head $(i, k)$ could sometimes contribute to subspaces that the attention head $(i, k)$ can never write to. Instead, for an attention head vertex, we represent its output computation in $d_{\text{head}}$-dimensional space:

$$\text{ZAttn}^{(i,k)}(x) = \text{AttnScore}^{(i,k)}(x)\left(RW_V^{(i,k)} + b_V^{(i,k)}\right)$$

and consider replacing $\mathrm{ZAttn}^{(i,k)}(x)$ rather than replacing $\mathrm{Attn}^{(i,k)}(x)$ to ablate an attention head. This slight modification reduces our parameter count by a factor of $d_{\mathrm{model}}/d_{\mathrm{head}}$ when applying OA but does not affect the results for the other ablation types.

We measure the single-component ablation loss gap $\Delta$ for the $\mathrm{ZAttn}^{(i,k)}$ and $\mathrm{MLP}^{(i)}$ vertices, $(n_{\mathrm{heads}} + 1) \cdot n_{\mathrm{layers}} = 156$ vertices in total for GPT-2.

## E.2 Ablation details

We consider zero ablation, mean ablation, optimal ablation, counterfactual mean ablation, resample ablation, and counterfactual ablation.

For a token position $j$, let $[\mathcal{A}(X)]_j$ denote the representation of $\mathcal{A}(X)$ at token position $j$.

**Zero ablation**: To zero ablate $\mathcal{A}$, we replace $[\mathcal{A}(X)]_j$ with 0 at each sequence position $j \geq 1$. We do not replace $[\mathcal{A}(X)]_0$ because it is a constant that does not depend on $X$ (any result at token position 0 must only be a function of $\mathrm{Resid}_0^{(0)}$, which represents a padding token that is the same for every sequence). Transformers may read from this beginning-of-string (BOS) token position in attention heads if no token in the sequence indicates a particularly strong signal, and since this token position does not distinguish between any inputs $X$ and is more appropriately viewed as a structural part of the architecture, we choose not to modify it.

**Mean ablation**: For each vertex $\mathcal{A}$, we compute $\mathbb{E}_{(X,Y)\sim\mathcal{D}}[\mathcal{A}(X)]$ over 20,000 samples, conditional on token position. We let $\mu_j = \mathbb{E}_{(X,Y)\sim\mathcal{D}}[[\mathcal{A}(X)]_j]$. To mean ablate $\mathcal{A}$, we replace $[\mathcal{A}(X)]_j$ with $\mu_j$ at each sequence position $j$.

In the Greater-Than dataset, all prompts $X$ are the same length, but in the IOI dataset, some prompts $X$ are longer than others, reducing our sample size for later sequence positions. In particular, if $X^*$ is the longest prompt in the dataset with $\ell$ tokens, then $\mu_\ell = X_\ell^*$, so the mean value actually carries identifying information about the prompt. Since we want the mean value $\mu_j$ to be uninformative about the original prompt $X$, we instead consider $m$ to the minimum length of any prompt, compute a modified mean $\mu_m = \mathbb{E}_{(X,Y)\sim\mathcal{D},\, S\sim\mathrm{Unif}\{1,...,\ell\}}[[\mathcal{A}(X)]_S \mid S \geq m]$ that considers all values of $\mathcal{A}(X)$ at token positions after token position $m$, and replace $[\mathcal{A}(X)]_j$ with $\mu_m$ if $j \geq m$.

**Optimal ablation**: Similar to mean ablation, we optimally ablate $\mathcal{A}$ by replacing $[\mathcal{A}(X)]_j$ with a constant $\widehat{a}_j$ for each sequence position $j < m$ and replacing $[\mathcal{A}(X)]_j$ with a constant $\widehat{a}_m$ for each $j \geq m$, where $m$ is the minimum length of any prompt. We initialize $(\widehat{a}_0, \widehat{a}_1, ..., \widehat{a}_m) = (\mu_0, \mu_1, ..., \mu_m)$ as defined for mean ablation and then optimize $(\widehat{a}_1, ..., \widehat{a}_m)$ for each ablated component $\mathcal{A}$ to minimize $\Delta(\mathcal{M}, \mathcal{A})$. Note that similarly to zero ablation, we fix $\widehat{a}_0 = \mu_0 = [\mathcal{A}(X)]_0$ and do not optimize its value as an ablation constant because $[\mathcal{A}(X)]_0$ does not depend on $X$ and thus naturally conveys no information about the input.

**Counterfactual mean ablation**: Our implementation is the same as for mean ablation except that we compute means over $(X, Y) \sim \mathcal{D}'$ for the counterfactual distribution $\mathcal{D}'$, and $m$ is taken as the minimum prompt length in the counterfactual distribution.

**Resample ablation**: To perform modified inference on an input $X$, we first sample an independent copy $X' \perp\!\!\!\perp X$. Let $X$ and $X'$ have lengths $s$ and $s'$ respectively. If $s \leq s'$, for an ablated component $\mathcal{A}$, we replace $[\mathcal{A}(X)]_j$ with $[\mathcal{A}(X')]_j$ at each token position $j \in \{1, ..., s\}$ (in other words, we only resample from the first $s$ tokens of $X'$). If $s > s'$, then we left-pad $X'$ with an additional $s - s'$ tokens to form a modified token sequence $\tilde{X}'$ that is the same length as $X$. We then replace ablated component values $\mathcal{A}(X)$ with $\mathcal{A}(\tilde{X}')$ with respect to each sequence position. Before arriving upon this implementation, we tried other choices, like resampling from the last $s$ tokens of $X'$ in the case that $s \leq s'$, or right-padding $X'$ in the case that $s > s'$.

**Counterfactual ablation**: We choose a function $\pi$ (details discussed in the main text and further analyzed in Appendix F.3) that maps inputs $X$ to neutral counterfactual inputs $X'$. Typically, $X$ and $X'$ are the same length and have many tokens in common. For ablated components, we replace $[\mathcal{A}(X)]_j$ with $[\mathcal{A}(\pi(X))]_j$ at each token position.

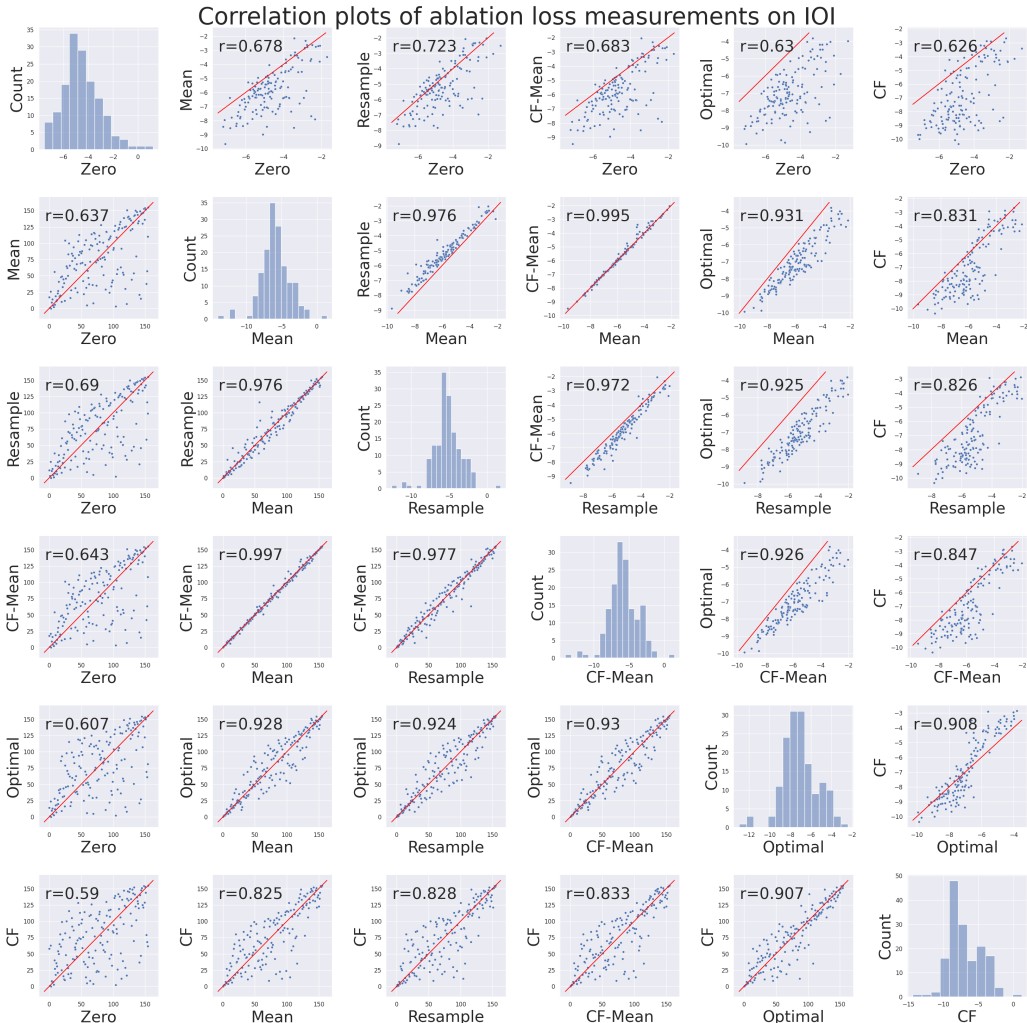

Figure 5: Correlation of single-component ablation loss measurements on IOI. Lower triangle shows rank correlation and upper triangle shows log-log correlation across metrics.

### E.3 Full results

Figure 5 plots the pairwise correlations of single-component ablation loss evaluated on the IOI dataset with a variety of ablation methods. Table 2 is an extended version of Table 1 in the main paper that provides a summary of these results.

Table 2: Comparison of ablation loss gap $\Delta$ on IOI, extended

|  | Zero | Mean | Resample | CF-Mean | Optimal | CF |
|---|---|---|---|---|---|---|
| Log-log correlation with CF | 0.626 | 0.831 | 0.826 | 0.847 | **0.908** | 1 |
| Rank correlation with CF | 0.590 | 0.825 | 0.828 | 0.833 | **0.907** | 1 |
| Mean $\Delta$ | 0.0584 | 0.0405 | 0.0559 | 0.0412 | **0.0035** | 0.0296 |
| Median ratio of $\Delta_{\text{opt}}$ to $\Delta$ | 11.1% | 33.0% | 17.7% | 31.7% | 100% | 88.9% |

## F Circuit discovery

### F.1 Transformer graph representation

We use a *residual rewrite* graph representation favored by Wang et al. (2022), Goldowsky-Dill et al. (2023), and Conmy et al. (2023). Similarly to Appendix E.1, we define vertices that correspond to an attention head, an MLP block, the model input ($\text{Resid}^{(0)}(x)$), or the model output ($\text{Out}(x)$), but we eliminate the $\text{Resid}^{(i)}(x)$ and $\text{MResid}^{(i)}(x)$ vertices. We have $n_{\text{layers}}(n_{\text{heads}} + 1) + 2$ vertices in total (156 for GPT-2). Notice from Appendix C.3 that

$$\text{Resid}^{(\ell)}(x) = \text{Resid}^{(0)}(x) + \sum_{i=1}^{\ell} \left( \text{MLP}^{(i)}(x) + \sum_{k=1}^{n_{\text{heads}}} \text{Attn}^{(i,k)}(x) \right)$$

so rather than assuming that attention heads, MLP blocks, and the model output take $\text{Resid}^{(i)}(x)$ as input, we can assume that they take the output of each previous block as a separate input to the computation. In particular, we can write

$$\text{MLP}^{(i)}(x) = \text{MLP}^{(i)}\Big( \text{Resid}^{(0)}(x), \text{MLP}^{(1)}(x), ..., \text{MLP}^{(i-1)}(x),$$
$$\text{Attn}^{(1,1)}(x), ..., \text{Attn}^{(i,n_{\text{heads}})}(x) \Big) \tag{13}$$

in which the $\text{MLP}^{(i)}$ vertex has $i(n_{\text{heads}} + 1) + 1$ incoming edges from previous vertices. Similarly, we can write

$$\text{Out}(x) = \text{Out}\Big( \text{Resid}^{(0)}(x), \text{MLP}^{(1)}(x), ..., \text{MLP}^{(n_{\text{layers}})}(x),$$
$$\text{Attn}^{(1,1)}(x), ..., \text{Attn}^{(n_{\text{layers}}, n_{\text{heads}})}(x) \Big) \tag{14}$$

so the $\text{Out}$ vertex has $n_{\text{layers}}(n_{\text{heads}} + 1) + 1$ incoming edges, one from each previous vertex in the graph. Finally, notice that attention heads $\text{Attn}^{(i,k)}$ take $\text{Resid}^{(i-1)}$ as input in *three* different locations, once in each of the query, key, and value subcircuits, so we can write attention heads as taking three *copies* of each previous vertex's output, which can be ablated individually.

$$\text{Attn}^{(i,k)}(x) = \text{Attn}^{(i,k)}\Big( \text{Resid}^{(0),Q}(x), \text{MLP}^{(1),Q}(x), ..., \text{MLP}^{(i-1),Q}(x),$$
$$\text{Attn}^{(1,1),Q}(x), ..., \text{Attn}^{(i-1,n_{\text{heads}}),Q}(x),$$
$$\text{Resid}^{(0),K}(x), \text{MLP}^{(1),K}(x), ..., \text{MLP}^{(i-1),K}(x),$$
$$\text{Attn}^{(1,1),K}(x), ..., \text{Attn}^{(i-1,n_{\text{heads}}),K}(x),$$
$$\text{Resid}^{(0),V}(x), \text{MLP}^{(1),V}(x), ..., \text{MLP}^{(i-1),V}(x),$$
$$\text{Attn}^{(1,1),V}(x), ..., \text{Attn}^{(i-1,n_{\text{heads}}),V}(x) \Big) \tag{15}$$

This notation indicates that attention heads admit multiple incoming edges for each previous vertex, which is somewhat non-standard. Alternatively, rather than allowing multiple edges between pairs of vertices, Conmy et al. (2023) creates a separate vertex for each of the query, key, and value subcircuits and considers each attention head output to take the outputs of these three circuits as input. However, edges between the subcircuits and attention head outputs are essentially placeholder edges that cannot be independently removed, since removing them is informationally equivalent to ablating the entire attention head. Thus, our graph representation is more natural and provides a more realistic edge count when considering removing model components.

Furthermore, we continue with the adjustment from Appendix E.1 of using $\text{ZAttn}^{(i,k)}$ as computational vertices rather than $\text{Attn}^{(i,k)}$ to conserve memory and reduce the parameter count of OA. We consider the linear map $\phi^{(i,k)}(Z) = ZW_O^{(i,k)} + b_O^{(i,k)}$ (so that $\text{Attn}^{(i,k)}(x) = \phi^{(i,k)}(\text{ZAttn}^{(i,k)}(x))$) and express all downstream vertices as taking $\text{ZAttn}^{(i,k)}(x)$ as input rather than $\text{Attn}^{(i,k)}(x)$ and performing their computation by pre-composing with $\phi^{(i,k)}$. For example, for an MLP vertex $\text{MLP}^{(i)}$, if $m^{(i)}$ represents how $\text{MLP}^{(i)}(x)$ is computed using $\text{Attn}^{(i,k)}(x)$ values as inputs, then its

computation taking $\mathrm{ZAttn}^{(i,k)}(x)$ as inputs is equal to

$$\mathrm{MLP}^{(i)}(x) = m^{(i)}\big(\mathrm{Resid}^{(0)}(x), \mathrm{MLP}^{(1)}(x), ..., \mathrm{MLP}^{(i-1)}(x),$$
$$\phi^{(1,1)}(\mathrm{ZAttn}^{(1,1)}(x)), ..., \phi^{(i,n_{\mathrm{heads}})}(\mathrm{Attn}^{(i,n_{\mathrm{heads}})}(x))\big)$$

We replace all $\mathrm{Attn}^{(i,k)}$ vertices in the graph structure with the corresponding $\mathrm{ZAttn}^{(i,k)}$ vertex.

In total, we have $(n_{\mathrm{heads}} \cdot n_{\mathrm{layers}})$ $\mathrm{ZAttn}^{(i,k)}$ vertices, $n_{\mathrm{layers}}$ $\mathrm{MLP}^{(i)}$ vertices, an input vertex ($\mathrm{Resid}^{(0)}$) and an output vertex (Out). Letting $V$ represent the set of vertices that includes the input and $\mathrm{MLP}^{(i)}$ vertices. There are $3 \cdot \frac{1}{2} n_{\mathrm{layers}} \cdot (n_{\mathrm{layers}} - 1) \cdot n_{\mathrm{heads}}^2$ edges between two attention heads, $2 \cdot (n_{\mathrm{layers}}+1) \cdot n_{\mathrm{layers}} \cdot n_{\mathrm{heads}}$ edges between an attention head and a vertex in $V$, $\frac{1}{2} \cdot n_{\mathrm{layers}} \cdot (n_{\mathrm{layers}}+1)$ edges between two vertices in $V$, and $(n_{\mathrm{layers}}) \cdot (n_{\mathrm{heads}} + 1) + 1$ edges from any vertex to the output.

For GPT-2, there are 28,512 edges between two attention heads, 3,744 edges between an attention head and a vertex in $V$, 78 edges between two vertices in $V$, and 157 edges from any vertex to the output for a total of 32,491 edges.

## F.2 Ablation details

For mean, resample, and counterfactual ablation, our implementation is the same as in Appendix E.2.

For optimal ablation, we adjust the implementation to remove dependence on token positions and further reduce the parameter count. For each ablated component $\mathcal{A}$, rather than training a different $a_j^*$ to replace $[\mathcal{A}(X)]_j$ for each token position $j$, we train a single optimal constant $a^*$ that is the same shape as any particular $[\mathcal{A}(X)]_j$. We initialize $a^*$ to $\mathbb{E}[[\mathcal{A}(X)]_j \mid j > 9]$, the subtask mean excluding early token positions, since early positional embeddings may have idiosyncratic effects. To ablate $\mathcal{A}$ during inference on $X$, for all token positions $j > 0$, we replace $[\mathcal{A}(X)]_j$ with $a^*$. As in Appendix E.2, we do not replace $[\mathcal{A}(X)]_0$ because this value is a constant that does not depend on $X$.

We take this conservative approach to demonstrate that OA can be implemented in a position-agnostic manner, yet still outperforming position-specific implementations of other ablation methods by a large margin. Since many other ablation methods, like resample and counterfactual ablation, are inherently position-specific, compatibility with a position-agnostic implementation is a key advantage of OA over these ablation methods, and we discuss this advantage further in Appendix F.3.

As noted in Section 3, we train a single constant $a_j^*$ for each vertex $\mathcal{A}_j$. An alternative implementation is training a separate constant for each ablated edge. In theory, this approach is consistent with the spirit of OA: though ablated edges transmit different values to downstream components that would normally receive the same value, none of the transmitted constants transmit any information about the input to downstream components. However, this approach would greatly increase the number of learnable parameters, and arguably may actually increase the computational capacity of the model. Additionally, training a single constant for each vertex has the appealing property that ablating all of the out-edges from a vertex is equivalent to ablating that vertex.

## F.3 Normative comparison of ablation types

Thus far, circuit discovery on language models has focused on synthetic subtasks for which a mapping $\pi$ from studied inputs $x$ to counterfactual inputs $\pi(x)$ is easily constructed. Recall that a crucial criterion for selecting $\pi$ is to preserve as many tokens as possible between $x$ and $\pi(x)$. For Greater-Than, an example counterfactual pair is

```
Token:        S            YY1 YY2            YY1*
x:      The [conflict] began in [18][89] and ended in [18]
π(x): The [conflict] began in [18][01] and ended in [18]
```

where the brackets [] are added to emphasize the two-token representation of the year. Similarly, for IOI, an example counterfactual pair is

```
Token:          S1            IO                            S2
x:      Friends [Alice]  and [Bob]   found a bone at the store. [Alice]  gave the bone to
π(x): Friends [Charlie] and [David] found a bone at the store. [Charlie] gave the bone to
```

Since $x$ and $\pi(x)$ share the same token at all but a few token positions (only the S1, S2, and IO token positions for IOI and only the YY2 token position for Greater-Than), we can isolate the effect of changing the specific token that conveys important information. CF thus allows us to study subtasks involving input-label pairs where subtask-relevant information is given by only one or several tokens.

However, replacing $\mathcal{A}(x)$ with $\mathcal{A}(x')$ typically incurs much higher loss if $x$ and $x'$ differ at many different token positions, even if most tokens are unimportant in relation to the behavior we wish to study. The model representation at a token position $j$ is likely to contain information specific to the tokens at position $j$ and at surrounding positions in the input $X$, so replacing $[\mathcal{A}(x)]_j$ with $[\mathcal{A}(x')]_j$ is likely to inject inconsistent information if $X_j$ and $X'_j$ (or pairs of tokens at corresponding proximate token positions) are different tokens, causing $\Delta$ to be high as a result of spoofing (per Section 2.3).

An illustration is the discrepancy in resample ablation loss between the Greater-Than and IOI subtasks. The Greater-Than dataset only contains a single prompt template, so any sampled $X$ and $X'$ only differ in tokens that encode the subject and year. Here is an example of a sampled $(X, X')$ pair:

```
Token:        S              YY1 YY2               YY1*
X:  The [conflict] began in [18][89] and ended in [18]
X': The [deal]     began in [15][47] and ended in [15]
```

On the other hand, the IOI dataset consists of multiple prompt templates that differ in sentence structure, so $X$ and $X'$ may differ at nearly all token positions, not just the S1, IO, and S2 positions as shown above. For example:

```
X:  Friends Alice and Bob  found  a  bone at  the store. Alice    gave the    bone to
X': <>      <>    <> Then, Charlie and David had a  long  argument, and Charlie said to
```

where <> represents a padding token added to make the sequences the same length. As a result, resample ablation loss is relatively low for Greater-Than (see Figure 10) but relatively high for IOI (see Figure 8), indicating that token parallelism is an important requirement for CF to work well.

While the synthetic IOI and Greater-Than datasets are specifically engineered so that we can modify a prompt $x$ at only a few token positions to obtain a neutral prompt $\pi(x)$, more general language behaviors may not be suited for this type of counterfactual analysis. Here are a few examples of language subtasks for which it not be possible to pair up $x$ and $\pi(x)$ with parallel tokens:

- A case study of the effect of modifiers, e.g. adjectives and adverbs, compared to a sentence with no modifier. Consider the following (degenerate) counterfactual pair, inspired by Marks and Tegmark (2023):

  ```
  x:  Paris is a   city in   the country of
  x': Paris is not a    city in  the     country of
  ```

  Since the presence of a modifier creates an extra token, replacing $\mathcal{A}(x)$ with $\mathcal{A}(x')$ patching between sequences with and without the modifier would result in the embedding at most token positions in $X$ being replaced with embeddings at the token position of a token reflecting the previous input token in $X$ ("city" with "a," "in" with "city," and so on).

- A case study comparing sentence order in situations where order matters, like giving directions. Patching in activations from a counterfactual prompt in which the order of two sentences is permuted involves introducing a new token at many token positions.

  ```
  x:  Make a      left turn,  then walk forward one  block. Your position is now
  x': Walk foward one  block, then make a       left turn.  Your position is now
  ```

- A case study relating to how language models handle mis-tokenization, like processing prompts in which a word is misspelled or the model is required to spell out a word.

  ```
  x:  The correct spelling of   the  word umpire  is
  x': The correct [sp]    [le] [ling] of   [te]    [h] word [up] [mire] is
  ```

Additionally, as the field of interpretability moves forward, we believe that it must progress toward "total" interpretation of models' internal mechanisms. This level of interpretation requires reasoning about subtasks that are much more general than those that have been studied and will require

performing intervention-based analysis across a broad distribution of inputs. For example, we may want to make the claim that certain components of a model $\mathcal{M}$ are unimportant for performing mathematical calculations; or that some components are not involved in ensuring grammatical correctness; or do not assist in making theory-of-mind assessments; etc. Additionally, we likely wish to assess component functions "in the wild" with filtered sampling from the model's training distribution as opposed to engineering synthetic datasets. These circumstances mean that the data will be much less suited for token parallelism between counterfactual prompts, so the adoption of a sequence-position agnostic ablation method is likely critical. This quality of OA makes it a much better candidate than CF as a suitable ablation method for scaling interpretability.

### F.4 Sparsity metric

As stated in Equation (3), we wish to select a circuit $\tilde{E}$ that achieves low loss $\mathbb{E}_X \mathcal{L}_X(\mathcal{M}^{\tilde{E}}_{(\text{opt})}(X))$ and which is a sparse subset of the model. Let $\mathcal{E}_{\mathcal{A}}$ represent the set of edges connected to vertex $\mathcal{A}$ in graph $G$, i.e. $\mathcal{E}_{\mathcal{A}} = \{(\mathcal{A}_j, \mathcal{A}_i, z) \in E \mid \mathcal{A}_j = \mathcal{A} \vee \mathcal{A}_i = \mathcal{A}\}$.

The selected circuit $\tilde{E}$ should ideally satisfy two types of sparsity:

1. *Edge* sparsity: $|\tilde{E}| << |E|$. The circuit should contain a small number of edges compared to the total number of edges in the model.
2. *Vertex* sparsity: $|\{\mathcal{A} \mid |\mathcal{E}_{\mathcal{A}} \cap \tilde{E}| > 0\}| << |G|$. The circuit should pass through a small number of vertices compared to the total number of vertices in the model.

There is a lack of guidance in prior work about whether smaller structures with more densely packed connections are more interpretable than larger structures with more thinly distributed connections. Indeed, one could argue that the larger structure is in fact easier to understand, since we do not need to dissect as many relationships to consider the function of any particular vertex within the circuit.

While circuit discovery aims to localize model behaviors on specific subtasks, we contend that a central challenge in interpretability going forward could be stacking together many circuit analyses to form a sum-of-the-parts analysis of the model's overall structure. Considering circuit discovery as a tool for decomposing model computation into interpretable subtasks, holding the total number of edges equal, we may prefer each circuit to have a smaller number of vertices to reduce the complexity of interactions *between* circuits rather than *within* circuits.

As such, we set $\mathcal{R}(\tilde{E})$ to select for circuits with high levels of both edge and vertex sparsity:

$$\mathcal{R}(\tilde{E}) = \lambda|\tilde{E}| + \gamma\lambda \sum_{\mathcal{A} \in G} \frac{1}{2}|\mathcal{E}_{\mathcal{A}}| \tanh\left(2\frac{|\mathcal{E}_{\mathcal{A}} \cap \tilde{E}|}{|\mathcal{E}_{\mathcal{A}}|}\right) \tag{16}$$

where $\lambda, \gamma$ are constants. Similarly, the continuous relaxation $\mathcal{R}(\theta_k)$ is for HCGS and UGS is derived by replacing $|\tilde{E}|$ with $\sum_{k=1}^{|E|} \theta_k$ and replacing $|\mathcal{E}_{\mathcal{A}} \cap \tilde{E}|$ with $\varepsilon_{\mathcal{A}} := \sum_{e_k \in \mathcal{E}_{\mathcal{A}}} \theta_k$.

Note that $\nabla_{\theta_k} \mathcal{R}(\vec{\theta}) = \lambda + \gamma\lambda \sum_{\mathcal{A} \in G} \text{sech}^2(2\frac{\varepsilon_{\mathcal{A}}}{|\mathcal{E}_{\mathcal{A}}|})$.

The first term, $\lambda$, is generally used to control the tradeoff between edge sparsity and circuit loss; a general interpretation is that we should include an edge $e \in \tilde{E}$ if its marginal contribution, $\Delta(\mathcal{M}, (E \cup \{e_k\}) \setminus \tilde{E}) - \Delta(\mathcal{M}, E \setminus (\{e_k\} \cup \tilde{E}))$, is greater than $\lambda$, expressing the same tradeoff as the discrete threshold $\lambda$ in ACDC. However, since ACDC is a less fine-grained optimization algorithm than UGS, the $\lambda$ required to achieve the same circuit size $|\tilde{E}|$ tends to be larger for ACDC.

The second term expresses vertex sparsity, and its effect is to increase the regularization effect for edges that are attached to vertices that have few other edges included in the circuit. Its effect is small when $\varepsilon_{\mathcal{A}} \approx |\mathcal{E}_{\mathcal{A}}|$, since $\text{sech}^2(2) \approx 0$, so we do not apply additional regularization to edges attached to vertices that have high overall likelihood to be included in the selected circuit. However, its effect is significant when $\varepsilon_{\mathcal{A}}/|\mathcal{E}_{\mathcal{A}}| \approx 0$ to prune the remaining edges from a vertex whose edge probabilities as represented by $\vec{\theta}$ are low on average. We use $\gamma$ to express the maximum influence of vertex regularization as compared to the effect of edge regularization (since $\max_x \text{sech}^2(x) = 1$), and generally select $\gamma = 0.5$, so the second term adds at most 50% more regularization.

## F.5 Uniform Gradient Sampling: motivation

In circuit discovery, the number of possible circuits $\tilde{E} \subset E$ is exponential in $|E|$ and the circuit losses $\Delta(\mathcal{M}, E \setminus \tilde{E})$ for subsets $\tilde{E}$ are not required to be related. $\Delta$ is not even necessarily monotonic in $\tilde{E}$ for any ablation method considered, i.e. $\tilde{E} \subset \tilde{E}'$ does not imply that $\Delta(\mathcal{M}, E \setminus \tilde{E}) \geq \Delta(\mathcal{M}, E \setminus \tilde{E}')$.

In reality, we can hope that the optimal ground-truth circuit $\tilde{E}^*$ is clear-cut and $\Delta$ is relatively well-behaved. If so, we could try to relax the discrete optimization problem and find a solution with gradient descent. As mentioned in Section 3, one continuous relaxation considers partial ablation for each edge $e_k = (\mathcal{A}_j, \mathcal{A}_i, z)$, where we replace $\mathcal{A}_j(x)$ as the $z$th input to $\mathcal{A}_i(x)$ with $\alpha_k \mathcal{A}_j(x) + (1 - \alpha_k)\hat{a}_j$ and use $L_1$ or $L_2$ regularization on the $\alpha_k$. However, this approach is likely to get stuck in local minima in which edge coefficients converge to the optimal magnitude instead of edges being completely ablated or retained. Instead, we consider a vector $\vec{\theta}$ of independent sampling probabilities for the inclusion of each edge, and optimize these probabilities so that each converges to 0 or 1. Our loss function is $f(\vec{\theta}) := \mathbb{E}_{X,(\tilde{E}\sim\vec{\theta})}[\mathcal{L}(\mathcal{M}^{\tilde{E}}(X), \mathcal{M}(X)) + \mathcal{R}(\tilde{E})]$. Denote $\mathcal{L}_{\mathcal{R}}(X, \tilde{E}) = \mathcal{L}(\mathcal{M}^{\tilde{E}}(X), \mathcal{M}(X)) + \mathcal{R}(\tilde{E})$, so that $f(\vec{\theta}) = \mathbb{E}_{X,(\tilde{E}\sim\vec{\theta})}[\mathcal{L}_{\mathcal{R}}(X, \tilde{E})]$. The gradient with respect to the sampling probability $\theta_k$ for edge $e_k$ is a *marginal* ablation loss gap:

$$\frac{\partial f(\vec{\theta})}{\partial \theta_k} = \mathbb{E}_{X,(\tilde{E}\sim\vec{\theta})}\left[\mathcal{L}_{\mathcal{R}}(X, \tilde{E} \cup \{e_k\}) - \mathcal{L}_{\mathcal{R}}(X, \tilde{E} \setminus \{e_k\})\right] =: \mathbb{E}_{X,(\tilde{E}\sim\vec{\theta})}\Delta_{\mathcal{R}}(\mathcal{M}, e_k,) \quad (17)$$

The problem is that $|E|$ is large and it is not tractable to estimate this quantity individually for all $k$. Our goal is to find a good sample estimator for this quantity simultaneously for all $k$.

One way to perform this simultaneous estimation is importance sampling, where we write

$$f(\vec{\theta}) = \mathbb{E}_{X,(\tilde{E}\sim\vec{p})}\left[\frac{\mathbb{P}_{E'\sim\vec{\theta}}(\tilde{E} = E')}{\mathbb{P}_{E'\sim\vec{p}}(\tilde{E} = E')} \cdot \mathcal{L}_{\mathcal{R}}(X, \tilde{E})\right]$$

so when $\vec{p} = \vec{\theta}$, $\frac{\partial f(\vec{\theta})}{\partial \theta_k} = \mathbb{E}_{X,(\tilde{E}\sim\vec{\theta})}\left[\frac{\mathbb{1}(e_k \in \tilde{E})}{\theta_k}\mathcal{L}_{\mathcal{R}}(X, \tilde{E}) - \frac{\mathbb{1}(e_k \notin \tilde{E})}{1-\theta_k}\mathcal{L}_{\mathcal{R}}(X, \tilde{E})\right]$. However, this method leads to poor estimates. Most of the variance in gradient updates to $\theta_k$ comes from sampling different subsets of edges among the $|E| - 1$ edges *other* than $e_k$, not the effect of fixing $e_k \in \tilde{E}$ or $e_k \notin \tilde{E}$ for a particular edge.

Instead, UGS is an approximation of the marginal ablation loss gap for each edge obtained by taking gradients with respect to sampled partial ablation coefficients $\vec{\alpha}$. We consider the extension of $\mathcal{M}^{\tilde{E}}$ to convex relaxations $\mathcal{M}^{\vec{\alpha}}$, where $\alpha_k$ represents the partial ablation coefficient for edge $e_k$ as alluded to above. Similarly, we consider $\mathcal{L}_{\mathcal{R}}(X, \vec{\alpha})$ in place of $\mathcal{L}_{\mathcal{R}}(X, \tilde{E})$. Let $\vec{\alpha}(\tilde{E}, S)$ where $S \subset \{1, ..., |E|\}$ such that $\alpha_k(\vec{U}, \tilde{E}, S) = \mathbb{1}(k \in S)U_k + \mathbb{1}(k \notin S)\mathbb{1}(e_k \in \tilde{E})$.

For any edge $e_k$, the marginal ablation loss gap inside the expectation in Equation (17) is equal to the expected gradient with respect to $\alpha_k \sim \mathrm{Unif}(0, 1)$ when other edges are sampled according to $\tilde{E}$:

$$\frac{\partial f(\vec{\theta})}{\partial \theta_k} = \mathbb{E}_{X,(\tilde{E}\sim\vec{\theta}),(\vec{U}\sim\mathrm{Unif}(0,1)\text{ iid})}\left[\frac{\partial}{\partial U_k}\mathcal{L}_{\mathcal{R}}(X, \vec{\alpha}(\vec{U}, \tilde{E}, \{k\}))\right]. \quad (18)$$

In other words, we can estimate the effect of totally ablating $e_k$ for a given $(X, \tilde{E})$ by sampling a partial ablation coefficient $\alpha_k \sim \mathrm{Unif}(0, 1)$ and taking the loss gradient with respect to $\alpha_k$. However, we run into the same problem of needing to estimate the effect individually for each edge $k$.

UGS assumes that we can estimate this loss gradient for many edges *simultaneously* without much bias. For any particular edge $e_k$, the interference effects caused by sampling other edges $e_k \sim \mathrm{Unif}(0, 1)$ for all $k \in S$ instead of setting them according to $\tilde{E}$ could be small, if $S$ is small enough. UGS assumes that if $S$ is sampled from a distribution $\mathcal{D}_S$

$$\frac{\partial f(\vec{\theta})}{\partial \theta_k} \approx \mathbb{E}_{X,(\tilde{E}\sim\vec{\theta}),(\vec{U}\sim\mathrm{Unif}(0,1)\text{ iid}),(S\sim\mathcal{D}_S)}\left[\frac{\partial}{\partial U_k}\mathcal{L}_{\mathcal{R}}(X, \vec{\alpha}(\vec{U}, \tilde{E}, S)) \,\middle|\, k \in S\right]. \quad (19)$$

## F.6 Uniform Gradient Sampling: construction

We use $\theta_k$ to represent the sampling probabilities for each edge, and perform gradient descent on the parameters by constructing a loss function whose gradient is a sample estimator of Equation (19). As noted in the main text, rather than using $\theta_k \in (0, 1)$ as our parameters, we use $\tilde{\theta}_k \in (-\infty, \infty)$ as our parameters and compute $\theta_k = \sigma(\tilde{\theta}_k)$. We initialize $\tilde{\theta}_k = 1$ for all edges $e_k$. We avoid random initialization because it achieves worse results by causing the resulting circuits to be suboptimally constrained to be close to our random prior.

We sample $S \subset \{1, ..., k\}$ by independently sampling each $\mathbb{1}(k \in S) \sim \text{Bern}(w(\theta_k))$, where a window function $w$ determines how often we sample $\alpha_k \sim \text{Unif}(0, 1)$. Additionally, we require that

$$\mathbb{P}(e_k \in \tilde{E} \mid k \in S) = \mathbb{P}(e_k \notin \tilde{E} \mid k \in S) = \frac{1}{2} \tag{20}$$

so that sampling $\alpha_k \sim \text{Unif}(0, 1)$ takes away probability mass equally from $e_k \in \tilde{E}$ and $e_k \notin \tilde{E}$. We perform this adjustment because for the purpose of estimating gradients $\mathbb{W}$ with respect to edges *other* than $e_k$, Equation (19) implicitly assumes that

$$\mathbb{E}[\mathbb{W} \mid k \in S] \approx p \cdot \mathbb{E}[\mathbb{W} \mid \alpha_k = 0] + (1 - p) \cdot \mathbb{E}[\mathbb{W} \mid \alpha_k = 1], \qquad p = \mathbb{P}(e_k \in \tilde{E} \mid k \in S) \tag{21}$$

and without any priors about the functional form of $\mathbb{W}$, $p = \frac{1}{2}$ is a reasonable choice.

We construct a loss function whose gradient is given by Equation (4), a sample estimator of Equation (19) with this construction of $S$. We construct the batch $X^{(1)}, ..., X^{(b)}$ by choosing $b/n_s$ unique samples of input $X$ and repeating each input $n_s$ times in the batch. We generally use $n_s = 12$ and $b = 60$. We choose $w(\theta_k) = c \cdot \theta_k(1 - \theta_k)$. Note that $c \leq 2$ in order for $\frac{1}{2}w(\theta_k) \leq \min(\theta_k, 1 - \theta_k)$ to hold, as is required by Equation (20), and we choose $c = 1$. We discuss this choice in F.8.

The full algorithm pseudocode is given after describing a few additional details in Appendix F.7.

## F.7 Additional circuit discovery details

For HCGS and UGS, we use learning rates between 0.01 and 0.15 for the sampling parameters.

**Pruning dangling edges** For HCGS and UGS, after sampling $\vec{\alpha}(\vec{U}, \tilde{E}, S)$ for each input, we remove "dangling" edges $e_k$. A dangling vertex is a vertex $\mathcal{A}$ for which there does not exist a path from the model input to $\mathcal{A}$ along edges $e_j$ for which $\alpha_j > 0$, or for which there does not exist such a path from $\mathcal{A}$ to the model output. A dangling edge is an edge that is connected to a dangling vertex. For a dangling edge $e_k$, we replace $\alpha_k = 0$ (i.e. the equivalent of removing $e_k$ from $\tilde{E}$ and $S$).

**Discretization** For HCGS and UGS, after training the $\theta_k$, we select a final circuit by selecting all edges for which $\theta_k > \tau$ for a threshold $\tau$. Generally, for UGS, all but a handful of $\theta_k$ converge to highly negative or positive values, the choice of $\tau$ does not have much impact, and we choose $\tau = 0.5$. However, for HCGS, many edges have $\theta_k$ parameters around zero even after training for 10,000 batches. We again select $\tau = 0.5$ since we observe that including edges with $\theta_k \in (-1, 0.5)$ does not generally affect performance.

**Optimizing constants for OA** For HCGS and UGS, we train ablation constants $\hat{a}$ concurrently with training the sampling parameters $\theta_k$. We use a learning rate of 0.002 for $\vec{a}$, lower than the learning rate used for the sampling parameters.

Note that we only provide gradient updates to $\hat{a}_j$ for a vertex $\mathcal{A}_j$ along edges $e_k$ for which $\alpha_k = 0$. Updating $\hat{a}_j$ when $\alpha_k \neq 0$ can lead $\hat{a}$ to update toward a value that is optimal when taking a linear combination of $\hat{a}$ and $\mathcal{A}_j(X)$, rather than a value that is optimal as a constant. See Figure 6.

Even though we obtain approximate constants $\hat{a}$ through the training process for HCGS and UGS, in order to level the playing field when comparing to ACDC and EAP, we do not use the constants found during training during circuit evaluation. Instead, for each circuit discovery algorithm, we evaluate circuits with OA by initializing constants to subtask means and then training for the same number of batches (10,000) with the same settings with a learning rate of 0.002.

Algorithm 1 shows the full algorithm of UGS, including our exact loss function, for optimization with OA. For optimization with other ablation methods, we set ablated values $\hat{a}$ according to the

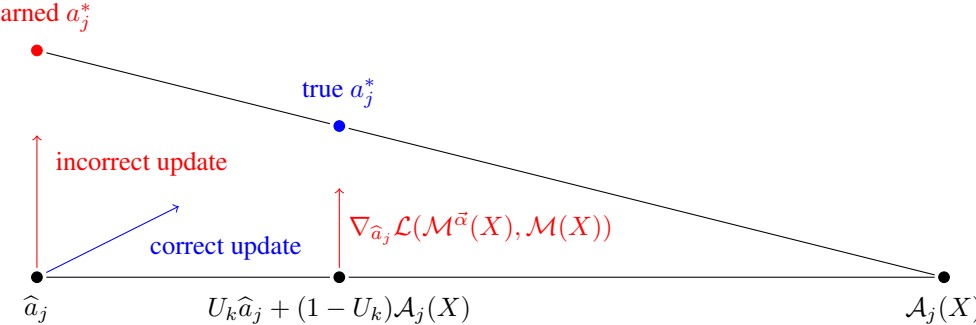

Figure 6: Gradient updates on $\widehat{a}$ can be biased when $\alpha_k \neq 0$.

ablation method and do not perform gradient updates on $\widehat{a}$. Note that we use the notation $\mathcal{M}_{\vec{\alpha}}(X, \widehat{a})$ with OA to indicate running the circuit evaluation with ablation coefficients $\vec{\alpha}$ and replacing ablated components with $\widehat{a}$, in line with our notation in Appendix C.2.

### F.8 Choosing a window size for UGS

We motivate the choice of our window function $w(\theta_k) = \theta_k(1 - \theta_k)$.

Let $f_k^* := \frac{\partial f(\vec{\theta})}{\partial \tilde{\theta}_k} = \frac{\partial f(\vec{\theta})}{\partial \theta_k} \cdot \frac{\partial \theta_k}{\partial \tilde{\theta}_k}$, and let $f_k$ be our sample estimate from approximation 19. Let $K \sim \text{Unif}(\{1, ..., |E|\})$. We may want to minimize the squared distance between our sample estimates and the true gradient values,

$$\varepsilon := \mathbb{E}(f_K^* - f_K)^2 = \mathbb{E}_K \text{Var}(f_K \mid K) + \mathbb{E}_K (\mathbb{E}[f_K \mid K] - f_K^*)^2. \tag{22}$$

Let our sampling distribution $S \sim \mathcal{D}$ be defined by independent Bernoulli random variables $\mathbb{1}(e_k \in S) \sim \text{Bern}(w_k)$. Assume that we collect $b$ samples of $(\vec{U}, \tilde{E}, S)$ in a batch and use the samples $i$ for which $k \in S_i$ to estimate the $f_k$. Let $T_k := \sum_{i=1}^n \mathbb{1}(k \in S_i)$, and let $f_k := \frac{1}{T_k} \sum_{i=1}^n f_k^{(i)} \mathbb{1}(k \in S_i)$. Let $f_k^{()}$ represent the first sample for which $k \in S_i$.

Assuming that $\alpha_k \sim \text{Unif}(0,1)$ w.p. $w_k$, $\alpha_k = 0$ w.p. $\theta_k - \frac{1}{2}w_k$, and $\alpha_k = 1$ w.p. $1 - \theta_k - \frac{1}{2}w_k$ as given by Equation (20), we have the loss derivative $|E|\frac{\partial \varepsilon}{\partial w_k} = \frac{\partial \text{Var}(f_k)}{\partial w_k} + \sum_{\ell \neq k} \left( \frac{\partial \text{Var}(f_\ell)}{\partial w_k} + 2(\mathbb{E}[f_\ell] - f_\ell^*)\frac{\partial \mathbb{E}[f_\ell]}{\partial w_k} \right)$. If we assume that the second term is roughly equal to a constant $c$ for all edges, then

$$|E|\frac{\partial \varepsilon}{\partial w_k} = \text{Var}(f_k^{()}) \left( \frac{\partial \theta_k}{\partial \tilde{\theta}_k} \right)^2 \frac{\partial}{\partial w_k} \mathbb{E}\left[ \frac{1}{T_k} \mid T_k > 0 \right] \tag{23}$$

since $T_k = 0$ for an edge implies that it simply does not get a gradient update. Note that $\frac{\partial \theta_k}{\partial \tilde{\theta}_k} = \theta_k(1 - \theta_k)$, and $\frac{\partial}{\partial w_k} \mathbb{E}\left[ \frac{1}{T_k} \right] \approx -bw_k^{-2}$ for a constant $b > 0$. Solving for $\frac{\partial \varepsilon}{\partial w_k} = 0$, $\varepsilon$ is minimized when $w_k \propto \theta_k(1 - \theta_k)\sqrt{\text{Var}(f_k^{()})}$ which motivates the definition of the window function $w(\theta_k) = \theta_k(1 - \theta_k)$ in our main experiments. We try including the additional factor of $\sqrt{\text{Var}(f_k^{()})}$, but our results do not improve.

### F.9 Comparison of UGS and HCGS

HCGS and UGS both involve sampling gradients from a distribution over $\vec{\alpha} \in (0, 1)^{|E|}$ and taking gradient steps on parameters $\tilde{\theta}_k$ that represent our confidence in $e_k \in \tilde{E}^*$ using an average of gradients with respect to the edge coefficients $\alpha_k$. The original explanation provided by Louizos et al. (2018) for the convergence of HCGS to a satisfactory subset of weights that minimizes a loss function similar to Equation (3) involves $L_0$ regularization. To the contrary, we believe that a more

---

**Algorithm 1** Uniform gradient sampling

---

**Input**: set of edges $E$, initial parameter array $\theta$, initial constant array $\widehat{a}$
**Output**: a set of edges $\tilde{E} \subset E$ that represents the circuit
**Require**: metric $\mathcal{L}$, learning rates $\delta_\theta$, $\delta_a$, final threshold $\tau$, batch size $b$, sample count per input $n_s$, window function $w$

> **loop**
>> $X \leftarrow [\,]$
>> $\alpha \leftarrow [\,]$
>> $\text{UnifCount} \leftarrow [0 \textbf{ for } k \in [\text{length}(\theta)]]$
>> **for** $j \in [b/n_s]$ **do**
>>> $\alpha[j] \leftarrow [\,]$
>>> $X[j] \leftarrow \text{sample\_input}()$
>>> **for** $i \in n_s$ **do**
>>>> $\alpha[j][i] \leftarrow [\,]$
>>>> **for** $k \in [\text{length}(\theta)]$ **do**
>>>>> $U \leftarrow \text{Unif}(0, 1)$
>>>>> $W \leftarrow w(\theta[k]).\text{detach\_gradient}()$
>>>>> $p \leftarrow W \cdot \theta[k] + (1 - W) \cdot \theta[k].\text{detach\_gradient}()$
>>>>> $\alpha[j][i][k] \leftarrow \left(\frac{p-U}{W} + 0.5\right).\text{clamp}(0, 1)$
>>>>> $\text{UnifCount}[k] \leftarrow \text{UnifCount}[k] + \mathbb{1}(\alpha[j][i][k] \in (0,1))$
>>>> $\alpha[j][i] \leftarrow \text{prune\_dangling\_edges}(\alpha[j][i])$
>> $L \leftarrow [\,]$
>> **for** $j \in [b/n_s]$ **do**
>>> **for** $i \in n_s$ **do**
>>>> **for** $k \in [\text{length}(\theta)]$ **do**
>>>>> $g \leftarrow b/\text{UnifCount}[k]$
>>>>> $\alpha[j][i][k] \leftarrow g \cdot \alpha[j][i][k] + (1 - g) \cdot \alpha[j][i][k].\text{detach\_gradient}()$
>>>>> $\widehat{b} \leftarrow (\alpha[j][i] == 0) \cdot \widehat{a} + (\alpha[j][i] > 0) \cdot \widehat{a}.\text{detach\_gradient}()$
>>>>>> where $+$ and $\cdot$ are applied componentwise; this step is only used with OA.
>>>>> $L.\text{append}(\mathcal{L}(\mathcal{M}_{\alpha[j][i]}(X[j], \widehat{b}), \mathcal{M}(X[judged])))$
>> $f \leftarrow (\sum L)/b$
>> $\theta \leftarrow \theta - \delta_\theta \cdot \nabla_\theta f$
>> $\widehat{a} \leftarrow \widehat{a} - \delta_a \cdot \nabla_{\widehat{a}} f$; this step is only used with OA.
>> Note that in practice, we use Adam to determine step sizes.
> $\tilde{E} \leftarrow \emptyset$
> **for** $k \in [\text{length}(p)]$ **do**
>> **if** $\theta[k] > \tau$ **then** $\tilde{E}.\text{add}(E[k])$
> **return** $\tilde{E}$

---

compelling explanation for the performance of HCGS is that its behavior of sampling gradients on a region of partial ablations, $\vec{\alpha} \in (0,1)^{|E|}$, serves as a vague approximation of Equation (18).

Sampling $\alpha_k \in \text{Unif}(0, 1)$ to obtain gradient information, rather than a scaled conditional Concrete distribution, makes the simultaneous gradient-sampling estimator unbiased in the single-dimensional case, and thus is the choice that makes Equation (19) most resemble Equation (18).

### F.10  Additional IOI circuit discovery results

This section displays additional circuit discovery results on the IOI subtask. In Figure 7, we show the tradeoff between $\Delta$ (y-axis) and $|\tilde{E}|$ (x-axis) for optimal ablation to compare different circuit discovery methods. In Figure 8 (left), we show this tradeoff for mean ablation, and in Figure 8 (right), we show this tradeoff for resample ablation.

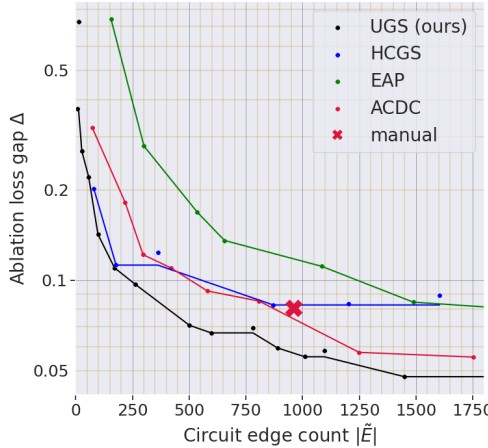

Figure 7: Circuit discovery Pareto frontier for the IOI subtask with optimal ablation.

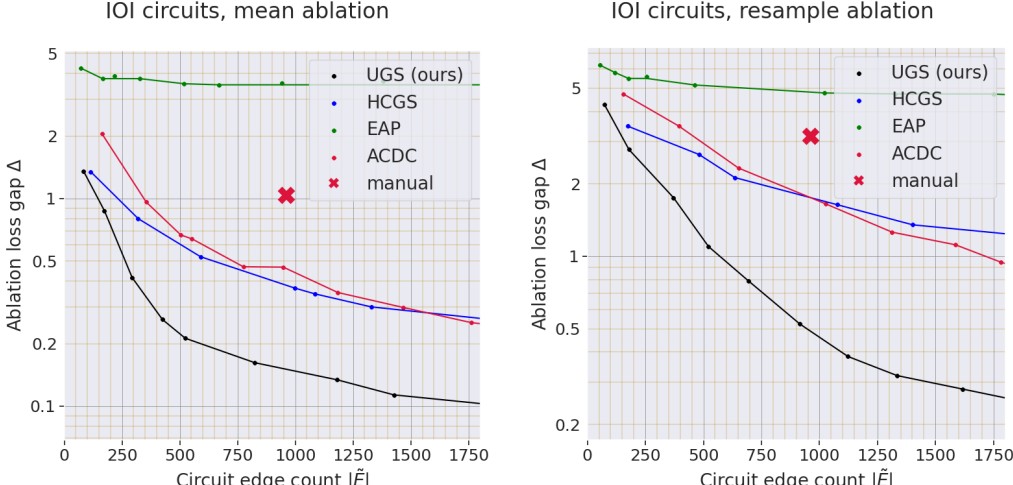

Figure 8: Circuit discovery Pareto frontier for IOI with mean ablation (left) and resample ablation (right).

### F.11 Circuit discovery results for Greater-Than

This section displays circuit discovery results on the Greater-Than subtask. In Figure 9 (left), we show the tradeoff between $\Delta$ (y-axis) and $|\tilde{E}|$ (x-axis) for optimal ablation. In Figure 9 (right), we show this tradeoff for counterfactual ablation. In Figure 10 (left), we show this tradeoff for mean ablation. In Figure 10 (right), we show this tradeoff for resample ablation. Finally, in Figure 11, we show the $\Delta$ achieved by circuits optimized using UGS on $\Delta$ with different ablation methods, analogous to Figure 1 (right) in the main text for the IOI subtask.

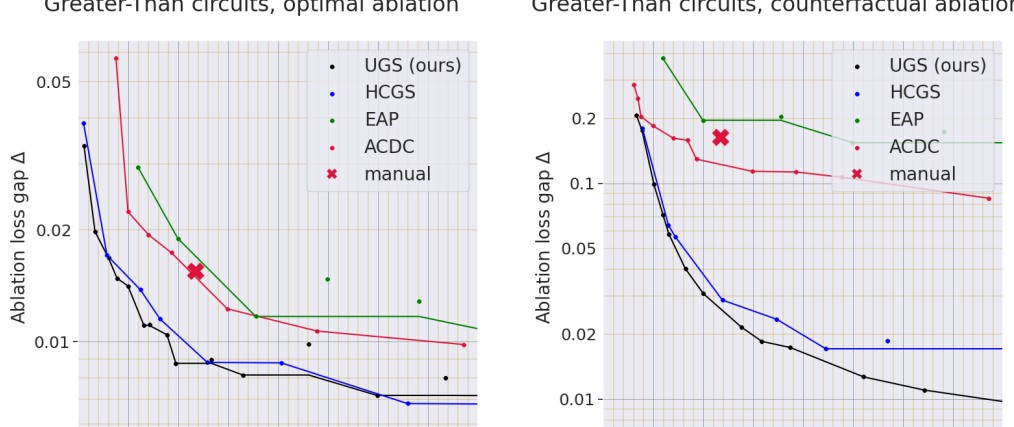

Figure 9: Circuit discovery Pareto frontier for the Greater-Than subtask with optimal ablation (left) and counterfactual ablation (right).

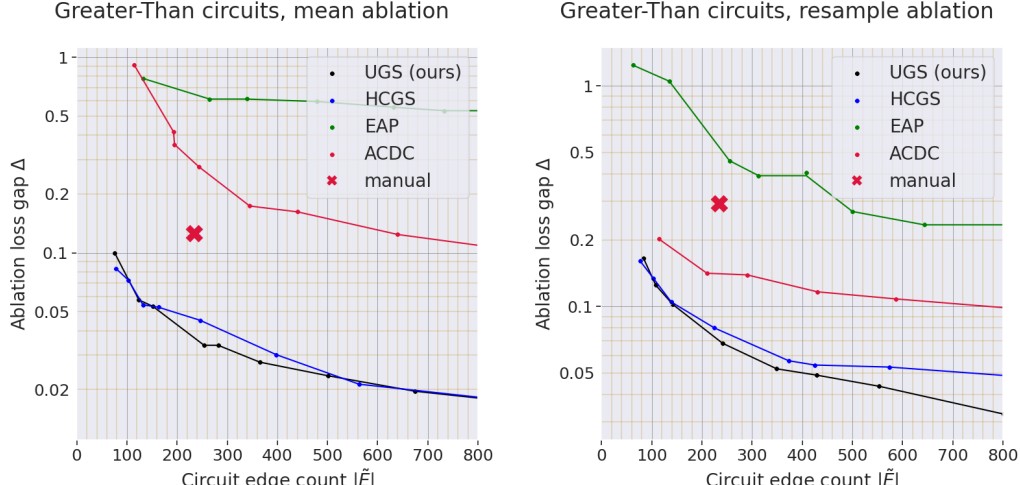

Figure 10: Circuit discovery Pareto frontier for Greater-Than with mean ablation (left) and resample ablation (right).

## F.12 Random circuits

One question is whether it may be possible to extract circuits with OA that do not necessarily explain model behavior on the training distribution by setting vertices to out-of-distribution values which maximally elicit a certain behavior. If the ablation constants $\hat{a}$ overparameterize the data, then performing OA could behave similarly to fine-tuning the model to perform the desired task.

Intuitively, however, OA strictly decreases the amount of computation available to the model, since we only add constants to model components and do not allow additional transformations of internal representation that are not already present in the downstream computation. To verify our stance, we compare the loss recovered by circuits discovered by UGS to random circuits to verify that OA indeed distinguishes subtask-performing mechanisms and does not provide enough degrees of freedom to elicit subtask behavior from unrelated model components.

In Table 3, we compare the $\Delta(\mathcal{M}, \tilde{E})$ achieved by random circuits $\tilde{E}$ to those achieved by circuits optimized with UGS for various ablation types. We construct $\tilde{E}$ by sampling each $\mathbb{1}(e_k \in \tilde{E})$ independently with some probability $p$, and prune dangling edges as detailed in Appendix F.7. We

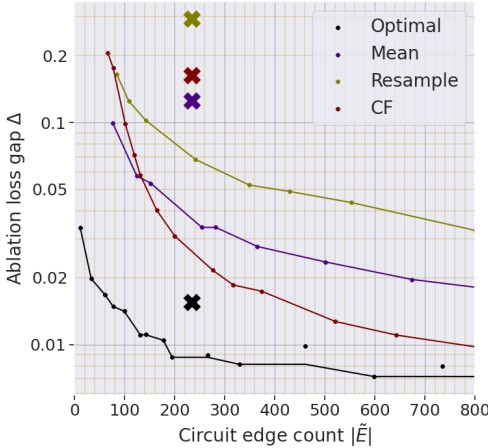

Figure 11: Comparison of different ablation methods for circuit discovery for Greater-Than.

Table 3: Optimized circuits compared to random circuits for various ablation types

| | | Mean | Resample | Optimal | Counterfactual |
|---|---|---|---|---|---|
| **IOI** | Random circuit loss | 4.529 | 6.527 | 2.723 | 4.264 |
| | UGS circuit loss | 0.264 | 1.779 | 0.176 | 0.191 |
| | Std | 0.200 | 0.085 | 0.024 | 0.049 |
| | Z-score | -21.28 | -55.67 | **-100.57** | -82.44 |
| **Greater-Than** | Random circuit loss | 1.010 | 2.109 | 0.900 | 1.785 |
| | UGS circuit loss | 0.033 | 0.056 | 0.029 | 0.021 |
| | Std | 0.020 | 0.039 | 0.011 | 0.027 |
| | Z-score | -49.29 | -52.89 | **-80.81** | -64.76 |

accept $\tilde{E}$ if $|\tilde{E}|$ is within an acceptable range, and we select $p$ to maximize the probability that $|\tilde{E}|$ falls within this range. We set our range of $|\tilde{E}|$ to be $[400, 500]$ for IOI and $[200, 300]$ for Greater-Than.

Recall that to evaluate circuits with OA, we perform gradient descent on $\hat{a}$ to approximate the optimal constants. Since repeating this process is expensive, we truncate training after just 200 training batches, far short of the 10,000 batches used for a full training run, for both the random circuits and optimized circuits. However, we test using a smaller sample size that the loss for the random circuits does not tend to decrease much with further training; in fact, for the optimized circuit, the loss typically drops by more than 50% after the first batch, which does not occur for the random circuits.

While random circuits achieve lower loss under OA than mean and resample ablation, the $\Delta_{\mathrm{opt}}$ measurements for random circuits do not approach the low figures achieved by optimized circuits. Furthermore, the standard deviation of $\Delta$ for random circuits is lower on average for OA than for mean or resample ablation, and surprisingly, the OA losses for optimized circuits have the most significant Z-score for both IOI and Greater-Than, though there is not necessarily a difference between Z-scores of such large magnitude. These results demonstrate that OA is likely highlighting specialized circuit components that already exist in the model rather than fabricating non-existent mechanisms.

## G    Causal tracing

### G.1    Transformer graph representation

Consider running a causal tracing experiment on vertex $\mathcal{A}$. We represent the model with four vertices: $\mathrm{Subj}(X)$ representing the subject tokens of the input, Non-Subj$(X)$ representing the remaining input tokens, the component of concern $\mathcal{A}(X) = \mathcal{A}(\mathrm{Subj}(X), \mathrm{Non\text{-}Subj}(X))$, and the model output

$\mathrm{Out}(X) = \mathrm{Out}(\mathrm{Subj}(X), \mathrm{Non\text{-}Subj}(X), \mathcal{A}(X))$. In particular, if $\mathcal{A} = \mathrm{MLP}^{(i)}$, then we compute $\mathrm{Out}(X)$ as a function of these three arguments by computing Equation (7), which takes $\mathcal{A}(X)$ and $\mathrm{MResid}^{(i)}(X)$ as input, by taking the latter term $\mathrm{MResid}^{(i)}(X)$ as a function of $\mathrm{Non\text{-}Subj}(X)$ and $\mathrm{Subj}(X)$, and then computing $\mathrm{Out}(X)$ as a function of $\mathrm{Resid}^{(i)}(X)$. A similar construction is used for attention layers.

Note that the AIE compares the performance of the model with the vertex Subj ablated (the denominator in Equation (5)) to the performance of the model with *only* the edge (Subj, Out) ablated (the numerator in Equation (5)).

## G.2 Relation of AIE to ablation loss gap

For consistency with Meng et al. (2022), we use a carefully selected loss function in the definition of $\Delta$ to represent proximity to the model's original predictions rather than the typical KL-divergence loss. In particular, we choose

$$\mathcal{L}_{\mathrm{AIE}}(P, Q) := \min\left(0, \max Q - [P]_{\arg\max Q}\right), \tag{24}$$

where $P$ and $Q$ represent probability distributions over the model vocabulary. Note that since the dataset is filtered so that $Y = \arg\max \mathcal{M}(X)$, replacing $\mathcal{L}$ with $\mathcal{L}_{\mathrm{AIE}}$ in $\Delta$, we get

$$\begin{aligned} \Delta &= \mathbb{E}_{X \sim \mathcal{D}} \mathcal{L}_{\mathrm{AIE}}(\mathcal{M}_{\mathcal{A}}(\xi(X), A(X)), \mathcal{M}(X)) \\ &= \mathbb{E}_{(X,Y) \sim \mathcal{D}} \max(0, [\mathcal{M}(X)]_Y - [\mathcal{M}_{\mathcal{A}}(\xi(X), A(X))]_Y) \end{aligned} \tag{25}$$

which is the numerator in Equation (5).

## G.3 Additional results

### Causal tracing intervention at all subject tokens, window size 1

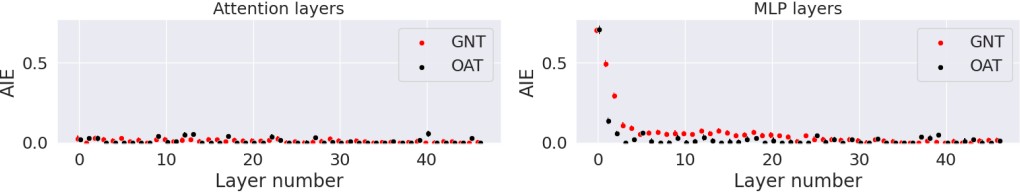

### Causal tracing intervention at last subject token, window size 1

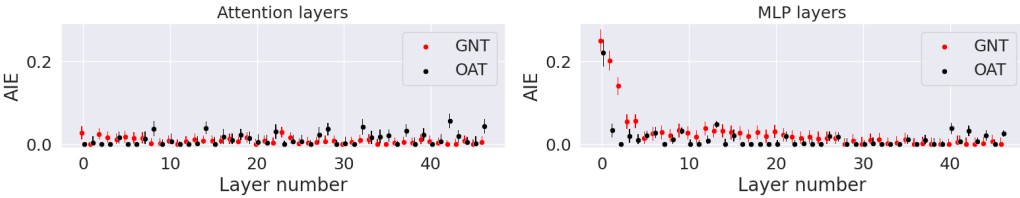

### Causal tracing intervention at last token, window size 1

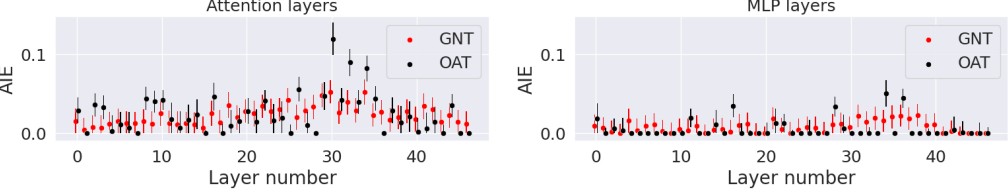

Figure 12: Causal tracing probabilities for different token positions with window size 1 (patching a single component). Error bars indicate the sample estimate plus/minus two standard errors.

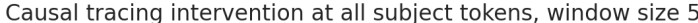

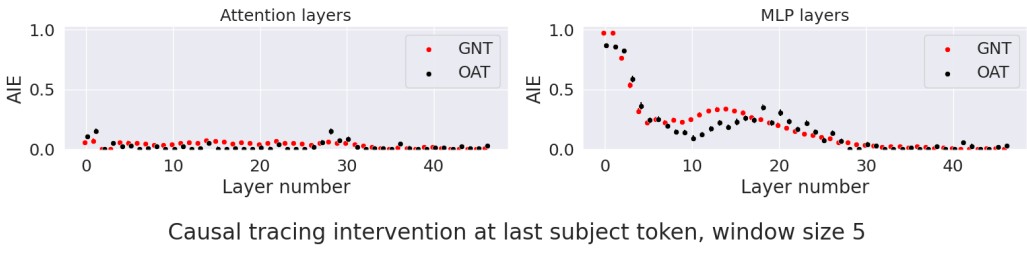

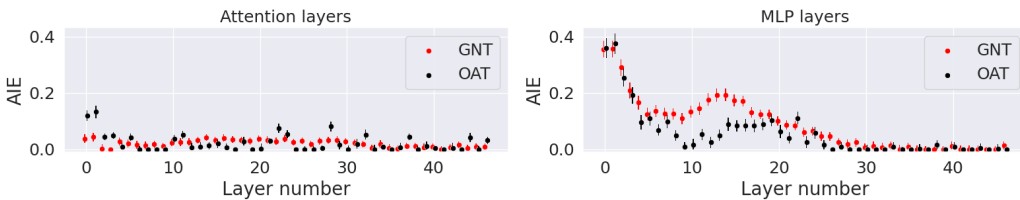

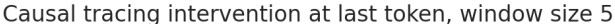

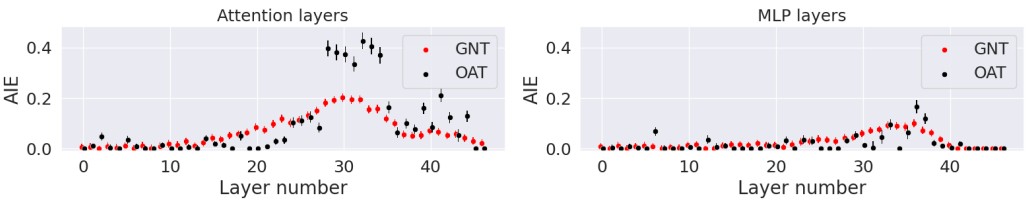

Figure 13: Causal tracing probabilities for different token positions with a sliding window of size 5. Error bars indicate the sample estimate plus/minus two standard errors.

We show results for additional window sizes and token positions. In particular, we show results for intervening at all subject token positions, only the last subject token position, and only the last token position, for window sizes 1 (see Figure 12), 5 (see Figure 13), and 9 (see Figure 14).

In addition to providing a more precise localization of components' informational contributions, the results provide some evidence against one of the claims of Meng et al. (2022), the idea that the last subject token position is a uniquely important "early site" for processing information at MLP layers 10-20. For sliding windows of size 5, GNT shows that intervening on MLPs at only the last subject token position achieves over half of the AIE of performing the same intervention at all subject token positions (33.8% vs 19.3%, shown in Figure 13). However, the OAT results indicate that intervening at all subject tokens is much more effective (35.2% vs 11.1%), indicating that early subject token positions may be more important than previously thought.

### G.4 Construction of standard errors

For input-label pairs $(X, Y) \sim \mathcal{D}$, let $W = \min(0, [\mathcal{M}(X)]_Y - [\mathcal{M}_\mathcal{A}(\xi(X), A(X))]_Y)$ and $Z = [\mathcal{M}(X)]_Y - [\mathcal{M}(\xi(X))]_Y$, and let $\widehat{W}_n$ and $\widehat{Z}_n$ be their respective sample means with $n$ samples. Recall from Equation (5) that we want to estimate from samples the quantity $\text{AIE}(\mathcal{A}) = \min\left(0, 1 - \frac{\mathbb{E}W}{\mathbb{E}Z}\right) =: \min\left(0, 1 - \frac{\mu_W}{\mu_Z}\right)$. By the central limit theorem,

$$\sqrt{n}\left(\begin{bmatrix} \widehat{W}_n \\ \widehat{Z}_n \end{bmatrix} - \begin{bmatrix} \mu_W \\ \mu_Z \end{bmatrix}\right) \xrightarrow{d} \mathcal{N}(0, \Sigma) := \mathcal{N}\left(0, \begin{bmatrix} \sigma_W^2 & \sigma_{WZ} \\ \sigma_{WZ} & \sigma_Z^2 \end{bmatrix}\right) \tag{26}$$

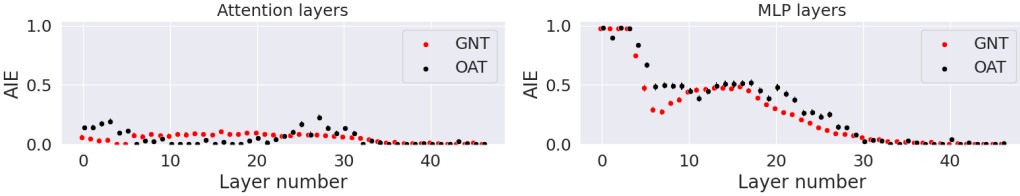

Causal tracing intervention at all subject tokens, window size 9

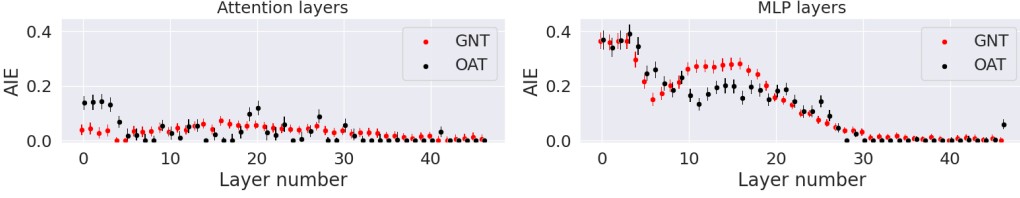

Causal tracing intervention at last subject token, window size 9

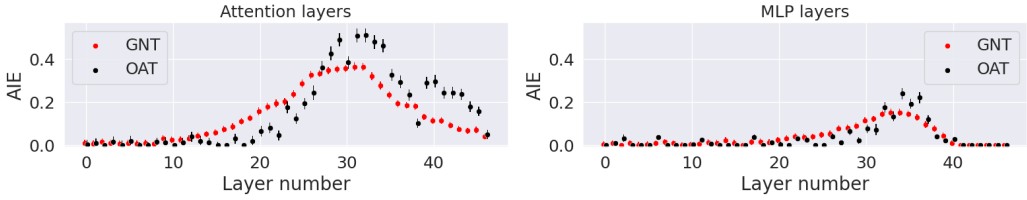

Causal tracing intervention at last token, window size 9

Figure 14: Causal tracing probabilities for different token positions with a sliding window of size 9. Error bars indicate the sample estimate plus/minus two standard errors.

By the multivariate delta method, for $h\left(\begin{bmatrix} w \\ z \end{bmatrix}\right) = \frac{w}{z}$ and $v := \nabla h\left(\begin{bmatrix} \mu_W \\ \mu_Z \end{bmatrix}\right) = \begin{bmatrix} \frac{1}{\mu_Z} \\ -\frac{\mu_W}{\mu_Z^2} \end{bmatrix}$

$$\sqrt{n}\left(\frac{\widehat{W}_n}{\widehat{Z}_n} - \frac{\mu_W}{\mu_Z}\right) = \sqrt{n}\left(h\left(\begin{bmatrix} \widehat{W}_n \\ \widehat{Z}_n \end{bmatrix}\right) - h\left(\begin{bmatrix} \mu_W \\ \mu_Z \end{bmatrix}\right)\right) \xrightarrow{d} \mathcal{N}(0, v^T \Sigma v) \qquad (27)$$

so the asymptotic variance is 
$$\frac{\mu_W^2}{\mu_Z^2}\left(\frac{\sigma_W^2}{\mu_W^2} + \frac{\sigma_Z^2}{\mu_Z^2} - 2\frac{\sigma_{WZ}}{\mu_W \mu_Z}\right)$$
which we estimate via samples to obtain our standard errors.

# H  OCA lens

## H.1  Transformer graph representation

We represent the model with $\mathrm{Resid}^{(i)}$, $\mathrm{MResid}^{(i)}$, $\mathrm{Attn}^{(i)}$, and $\mathrm{MLP}^{(i)}$ vertices for each layer $i$ and a vertex $\mathrm{Out}(x)$ representing the model output, where the relationships between the vertices are defined by the equations given in Appendix C.3. Applying OCA lens at layer $i$ entails ablating vertices $\mathrm{Attn}^{(i+1)}$ through $\mathrm{Attn}^{(N)}$ (where $N$ is the number of layers in the model).

## H.2  Additional prediction accuracy results

Figure 15 shows results on prediction loss for GPT-2-small, GPT-2-medium, and GPT-2-large. For all models, we use a learning rate of 0.01 for tuned lens and 0.002 for OCA lens.

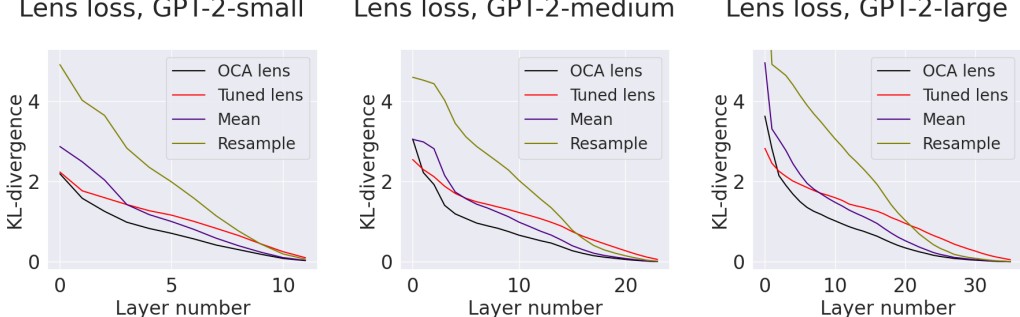

Figure 15: Comparison of prediction loss between tuned lens and ablation-based alternatives.

## H.3 Additional causal faithfulness results

The following figures show the causal faithfulness metrics with several kinds of perturbations. Let $\mu = \mathbb{E}[\ell_i(X)]$ and $\Sigma = \mathrm{Var}(\ell_i(X))$.

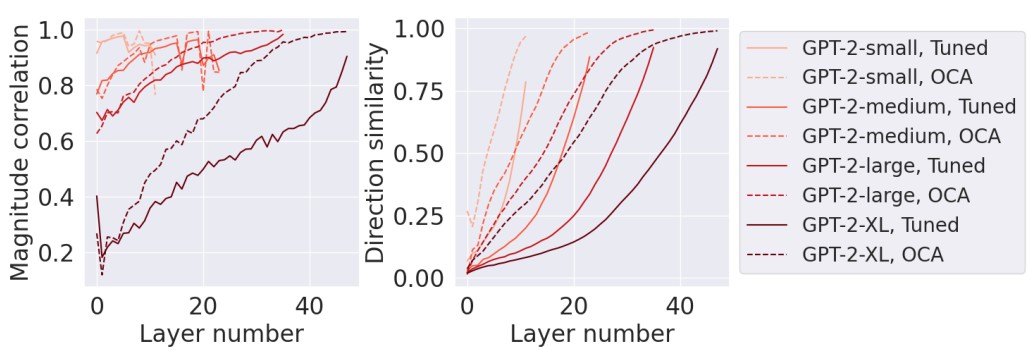

Figure 16: Causal faithfulness comparison under random perturbations.

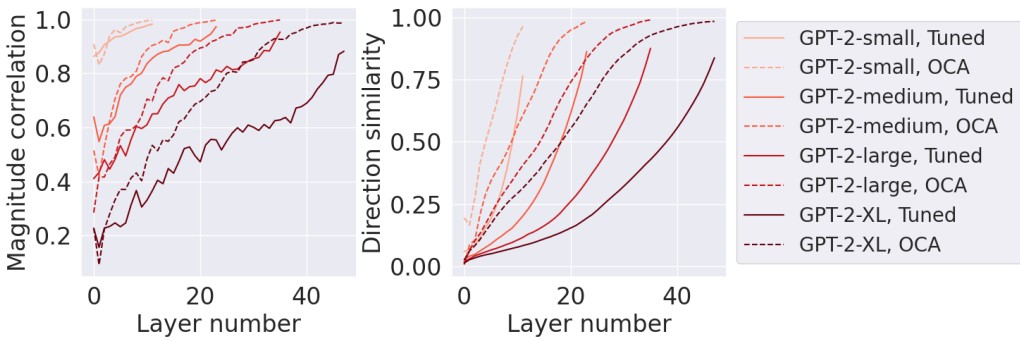

Figure 17: Causal faithfulness comparison under basis-aligned perturbations.

- Random perturbation: We sample $Z \sim \mathcal{N}(0, \Sigma)$, and let $V = Z/||Z||$. We let $Z' \sim \mathcal{N}(0,1)$, $Z' \perp\!\!\!\perp Z$. We define $\xi(a) = a + c \cdot Z' \cdot V$. We define the constant $c$ such that $\mathbb{E}[\mathcal{L}(\mathcal{M}(X;\xi), \mathcal{M}(X))] \approx 0.2$. Results shown in Figure 16.

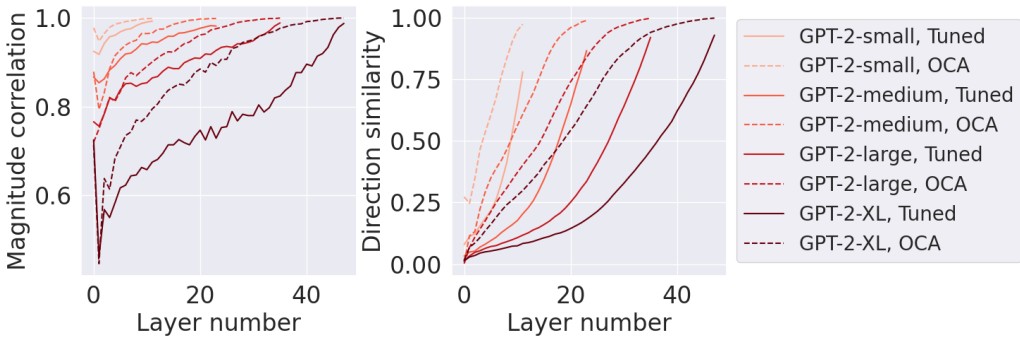

Figure 18: Causal faithfulness comparison under random projections.

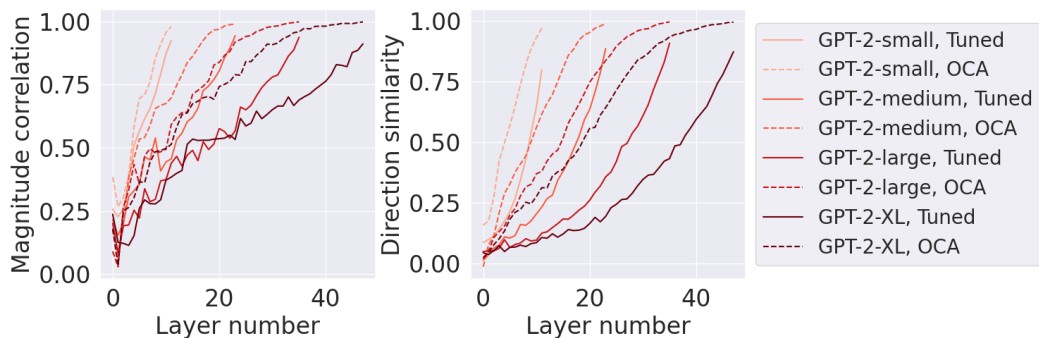

Figure 19: Causal faithfulness comparison under basis-aligned resample ablation.

- Basis-aligned perturbation: Same as random perturbation, except we choose a basis of $\overline{d_{\text{model}}}$ vectors as described in Section 5, and let $Z$ be a uniformly sampled basis element. Results shown in Figure 17.
- Random projection: We sample $Z \sim \mathcal{N}(0, \Sigma)$, and let $V = Z/||Z||$. We define $\xi(x) = \overline{\mu + p(a - \mu)}$, where $p$ represents the projection to the orthogonal complement of $V$. Results shown in Figure 18
- Basis-aligned projection: Described in the main text. Results shown in Figure 3.
- Basis-aligned resample ablation: We choose a basis as described in Section 5. We consider the subspace spanned by 100 basis elements with the largest singular vectors, and define $\xi(a)$ by performing resample ablation on the projection of $a$ to this subspace. Results shown in Figure 19.

We find that the improvement in causal faithfulness is consistent across all perturbation types studied.

### H.4 Elicitation results on factual datasets

We show additional results for elicitations on the text classification datasets. Figure 20 shows comprehensive results for each of the individual datasets. For our experiments, we use 10 demonstrations and sample from datasets without replacement to generate the demonstration examples. Note that we exclude SST2-AB, a toy dataset constructed by Halawi et al. (2024) that replaces SST2 sentiment labels with letters "A" and "B," since it is only created to show that elicitation accuracy does not improve when the expected answer is unrelated to the question (since the label is encoded in a non-intuitive manner, information from later layers is required to relate internal knowledge to the correct label). Figure 21 summarizes the elicitation accuracy boost between OCA lens and tuned lens across the datasets.

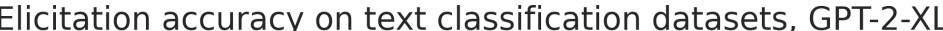

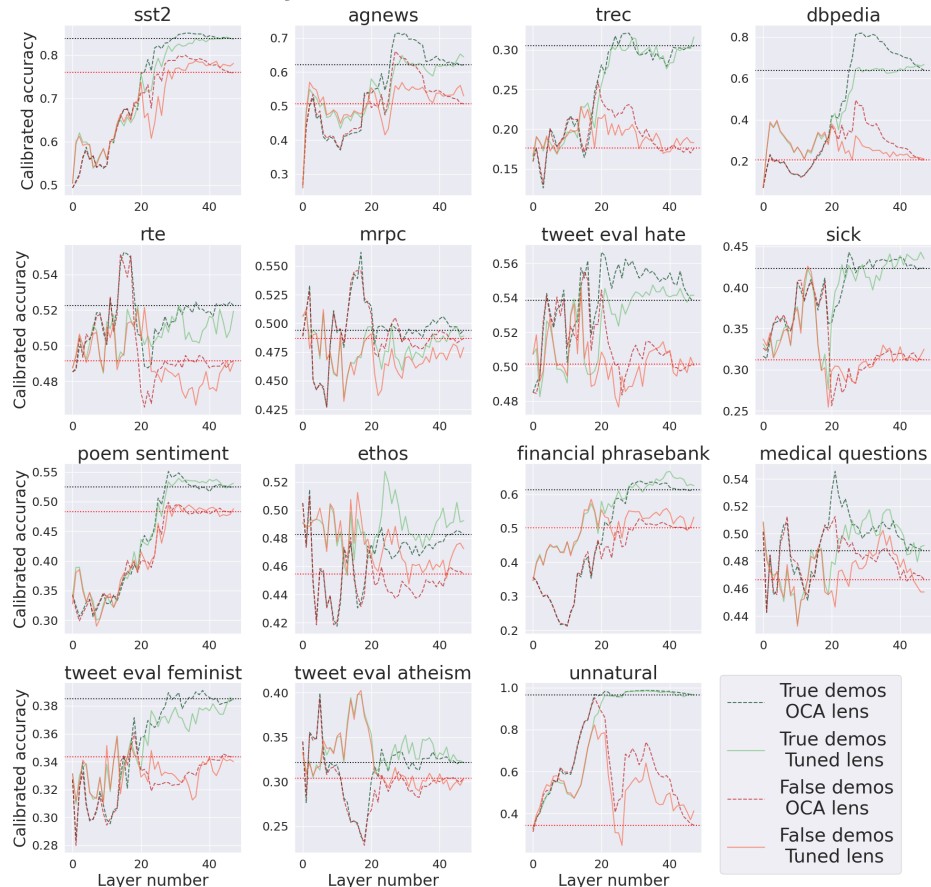

Figure 20: Comparison of calibrated accuracy of elicited completions on 15 datasets from Halawi et al. (2024). Dotted lines indicate the accuracy of the model's output predictions for true demonstrations (black) and false demonstrations (red).

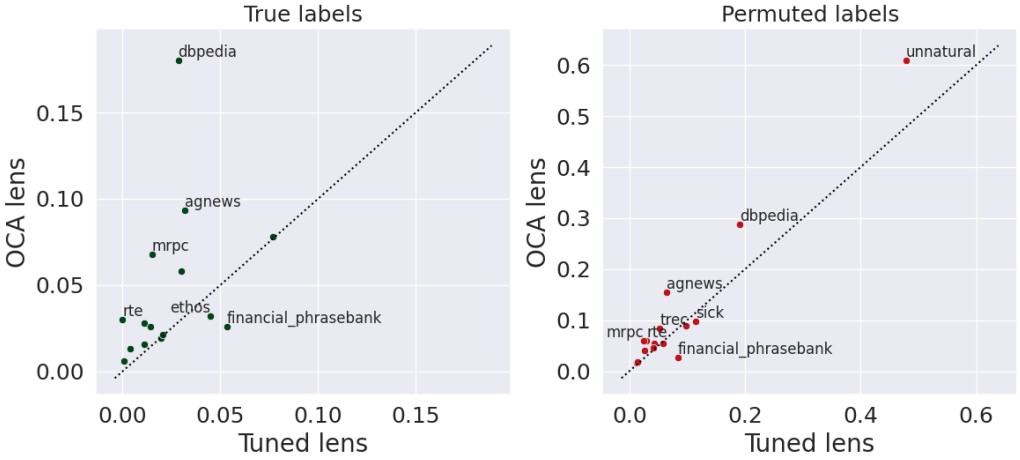

Figure 21: Comparison of elicitation accuracy boost between OCA lens and tuned lens.

# I Reproducibility

All code can be found at https://github.com/maxtli/optimalablation.

All experiments were run on a single Nvidia A100 GPU with 80GB VRAM. The cost of UGS is comparable to ACDC (about 1-2 hours to train). Training OCA lens until convergence also takes about 3-5 hours, which is similar to the amount of time to train tuned lens.

## J   Impact statement

We believe that OA can lead to a more granular level of understanding for models' internal mechanisms. A better understanding of interpretability can help to reduce risk from dangerous AIs, but more work is required to scale interpretability techniques to larger models. Interpretability can also help us to understand how to build better inductive biases into models, paving the way for future developments in architecture. On the other hand, advanced interpretability can also be repurposed for nefarious applications, like eliciting dangerous knowledge from models' latent space. However, we believe that better interpretability will also provide better clarity on how to mitigate these risks.

