# OpenReview forum: "Optimal ablation for interpretability"
_NeurIPS.cc/2024/Conference — NeurIPS 2024 spotlight_

### Official Review · Reviewer_Nr7M · 2024-07-11

**Soundness:** 3
**Presentation:** 3
**Contribution:** 4
**Rating:** 8
**Confidence:** 3

**Summary:**

This paper presents an alternative approach to perform component ablations for neural network interpretability. Specifically, the authors consider a neural network as a causal graph and propose *optimal ablations* which simulate component removal by setting the value of a node in the computational graph to a constant value. In contrast to alternative task-agnostic ablation methods (e.g. zero and mean ablations), this constant is learned using gradient descent. The authors argue that alternative task-agnostic ablation methods could confuse downstream computations as these specific values might not have been observed in the training data and then demonstrate how optimal ablations can lead to improvements across a range of different application domains: 1. They evaluate single-component ablations on the indirect object identification (IOI) task and find that optimal ablations perform better than other subtask-agnostic methods (although still worse than subtask-specific approaches); 2. They perform circuit discovery on the IOI and Greater-Than tasks using a new method, which they term uniform gradient sampling (UGS), which outperforms existing circuit discovery approaches; 3. They study factual recall, or more specifically localize where factual associations are stored within a language model, and in contrast to previous studies observe that adjacent components often have very different patching losses which suggests that factual associations are likely stored in single layers, as opposed to being spread across multiple layers as previously hypothesized; 4. Finally, the authors apply optimal ablations to the domain of decoding latent predictions. Here, they propose the optimal constant attention (OCA) lens, which sets all following components to their optimal constant value, instead of skipping these by learning a linear map such as in the tuned lens. They find that OCA scores better on KL-divergence and in terms of faithfulness to the model computations.

**Strengths:**

This paper made a strong impression on me as it combines methodological contributions with extensive empirical studies. The authors take a simple but well-motivated idea and then demonstrate how it can lead to improvements across a range of different interpretability application domains. To this end, the authors propose a range of novel methods that leverage this idea, including UGS, and OCA. I believe that these results could have a significant impact on the (mechanistic) interpretability field and should be of interest to various other subfields.

**Weaknesses:**

- Some of the applications lack depth as a result of the number of different methods studied. For example, it would have been interesting to further investigate the observations on factual recall.
- Optimal ablations still appear to perform worse than subtask-specific ablation methods in single-component ablations. However, the authors make a reasonable argument that these are subjective and require human and require manual effort, making them hard to use in some settings.

**Questions:**

1. Resample ablations are almost on par with optimal ablations in the single-component ablation setting (see Table 1), but perform significantly worse than mean and optimal ablations when it comes to making latent predictions (see Figure 3). Do you have a hypothesis as to why this might be the case?

**Limitations:**

I believe the limitations are properly addressed in the paper.

---

> ### Author Rebuttal · Authors · 2024-08-07
>
> Thank you for the positive and helpful feedback, and we’re glad you appreciated our work!
>
> > Some of the applications lack depth as a result of the number of different methods studied. For example, it would have been interesting to further investigate the observations on factual recall.
>
> ## Comment on Figure 2
>
> **We agree that it would be great to show further results on factual recall. We’ve extended our analysis to include localization results for different token positions and sliding window sizes as considered by Meng et al. (see PDF attached to global reply), which we will add to the appendix. We also made a few modifications, like increasing our dataset size for factual prompts from 500 to 4,000 and changing the y-axis to reflect a percent (%) difference between original and corrupted probability recovered rather than a difference in percentage points, and added appropriate standard errors to the figures, which is why the new Figure 2 looks different.**
>
> There are more directions we could pursue, like human mechanistic interpretability analysis to investigate how the recovered probability occurs, but in general, we acknowledge that we have limited space, and want to show as many different applications of OA as possible. We hope we’ve done enough to show that OA is a promising and versatile method; depth will hopefully come from further studies.
>
> > Optimal ablations still appear to perform worse than subtask-specific ablation methods in single-component ablations. However, the authors make a reasonable argument that these are subjective and require human and require manual effort, making them hard to use in some settings.
>
> Note that the only place where OA performs worse than CF is the predictive power of single-component OA loss for inclusion in the manual circuit (the second line of Table 1 in the original submission). However, one element we should point out is that **the manual circuit was discovered by using counterfactual ablations to investigate the model,** which provides a strong inductive bias in the human studies toward selecting components that create high loss when ablated with CF. Thus, it’s not surprising that single-component CF has better predictive power for this particular circuit.
>
> Instead, the point of this analysis is to confirm that OA produces measurements that are related to CF. To clarify, our stance is that, among *previous* methods, CF likely best reflects effect 1 (which we loosely consider “ground truth importance”) from the rewritten part of section 2.2 (see global reply), and we will clarify this point in the paper. Thus, it would make sense that, if OA were now the closest method to “ground truth importance,” then OA would also be the method closest to CF among existing methods, and we present the circuit prediction result in Table 1 to confirm this occurrence. However, after further reflection, we think this result may only confuse the reader, since we do not propose to use single-component ablation loss for circuit discovery, and propose to remove it from the paper (while keeping the correlation result).
>
> > Resample ablations are almost on par with optimal ablations in the single-component ablation setting (see Table 1), but perform significantly worse than mean and optimal ablations when it comes to making latent predictions (see Figure 3). Do you have a hypothesis as to why this might be the case?
>
> Yes – one general theme is that replacing activations from one sequence with those from another sequence produces the least loss of coherence when the sequences have a similar format or share many tokens in common. The single-component ablation experiment uses IOI, a dataset with 13 similar-looking prompt templates, so we are resampling between sequences that mostly look the same. But in the lens experiments, we are trying to perform latent prediction on the entire OpenWebText dataset, and we are resampling activations between sequences that may have completely different structure and content, hence the significantly worsened performance of resample ablation. This result furthers the notion that OA is more universally applicable than resample ablation and, in particular, can be used for more diverse datasets.

---

> > ### Comment · Reviewer_Nr7M · 2024-08-11
> >
> > Thank you for the additional experiments on factual recall and for clarifying some of the observations in the paper. I have read the rebuttal, as well as the other reviews. I believe that the ideas in the paper are well motivated and that it makes a strong contribution to the field. Thus, I strongly suggest to accept the paper for the conference and increase my score accordingly.

---

### Official Review · Reviewer_nbx7 · 2024-07-13

**Soundness:** 3
**Presentation:** 3
**Contribution:** 4
**Rating:** 8
**Confidence:** 4

**Summary:**

This paper explores a new approach to replacing features in the forward pass of a neural network, for purposes of interpretability. The approach is, rather than zeroing out features or replacing them with some pre-specified constant or random variable, to optimize a replacement constant to minimize the loss of the model over the subtask of interest. So when ablating an attention head or MLP layer, the ablated component’s outputs are replaced with a constant that is optimized to maximize model performance over some data of interest. This is done to be “minimally disruptive to model inference”, e.g. avoid disrupting model inference in the way that OOD replacement values are known to disrupt model inference, as has long been lamented by past work in interpretability. This new feature replacement method is demonstrated in three case studies, focused on circuit analysis, factual association localization, and something to do with the tuned lens that I could not quite understand. Results suggest that this method reveals clearer/sharper/stronger phenomena in circuit analysis and factual association localization, a promising result for interpretability research.

**Strengths:**

- Very important: The paper’s main strength is that it does something new and interesting in what has been a very saturated space for years. For a long time — since at least 2007 by my count — people have wondered how to remove features (or the outputs of some feature extractor / neural model) from a model, in order to do something like estimate that feature’s effect on the model. The debate rages on, now re-enacted in the era of mechanistic interpretability. While I’m still not sure we have a general theory for this, this paper makes a reasonable proposal on how to do this in a task and model agnostic way, with some promising early results for downstream applications. More specifically, avoiding disrupting model computation “too much” has stood out as a goal of feature replacement methods, and this paper proposes a new method for doing so and provides some interesting case studies showing that this may produce more stronger results in different kinds of interpretability analyses.
- Important: The paper is ambitious in a way rarely seen in ML papers, with a new method being applied across three case studies. The paper covers a lot of ground quickly.
- Important: The paper is well-written and the notation is clear.
- Important: I am sympathetic to the view that subtask-specific interchange interventions can be difficult to construct (a view argued for in the paper, that motivates the proposed method). At the very least, they take some manual effort to construct and are subjective. A drop-in automated replacement could be very useful. For what it’s worth, an argument I would add to the paper is that task-specific counterfactuals can sometimes be difficult to interpret as well. Does changing a name from Alice to Carol have *only* the effect of removing task-relevant information? We can imagine some hidden effects appearing with counterfactuals that we try to construct to remove only one variable, especially for more complex tasks.
- Important: The results on circuit analysis and localization are pretty promising, in my view. (I have a very hard time interpreting the results regarding the tuned lens — I don’t understand that experiment’s setup or its broader purpose.)
- Important: Experiments appear to be conducted with great care. Many small design choices seem good and clever, like the ReLU in Eq5 preventing rewarding the optimization process for finding a value that leads to over-restoration of model predictions during the causal intervention.
- Important: Appendix D shows that the optimization process is unlikely to adversarially induce good task performance in the model, and therefore unlikely to produce interpretability artifacts, because the causal intervention does not seem expressive enough to so.

**Weaknesses:**

- Very important: In my opinion, the approach presented cannot be the right one in general, preventing the proposal from providing a generic solution that circumvents the need to design subtask-specific interchange interventions / counterfactuals. There are two reasons for this, one a criticism of the argument in the paper and the other a counterexample. First, the argument in the paper, while being mathematically clear, is not philosophically rigorous. It is said that we want to “simulate the removal of” model components. What does that mean? I could imagine removing a layer from a Transformer by not running it — by virtue of the residual connection between layers, this is equivalent to zero-ablating the outputs of that layer. Yet the authors take issue with zero ablation, so removing a component must mean something else. Then, to remove a component, it is said that we should be “minimally disruptive to model inference”. What does that mean? Evidentally we are disrupting the model inference. What does it mean to not disrupt it too much? There are some other specific claims that are difficult to interpret precisely, such as avoiding an intervention that “disrupts this flow of information” or using an intervention that provides information that is “inconsistent with information the model derives from other vertices.” To be clear, I basically do like these kinds of arguments in ML papers, because we need arguments for what is right and wrong to do stated in plain English, but I am not sure this one is adequate for proving the point that optimal ablation is the right way to go about things.

    Second, a counterexample can build on an intuition hinted at in the paper that “there may not exist a constant value that perfectly conveys a lack of information.” The problem is that it is not clear that optimal ablation is the answer to an intelligible question. For example, suppose we have a neural model that classifies people into “give loan” and “reject loan” categories, and that we know that a neuron in this model represents a person’s annual income in USD. So, we know what 0 means, and we know what 100,000 means (it’s their income). Now let’s say there’s a component that takes in this value, and we want to know this value’s “importance” to that component (or later model outputs), so we replace this value with the optimal ablation constant c. What value of c would be correct for this? The question doesn’t really make sense, and it’s because it doesn’t make sense to ask how “important” a component is. Importance is not well-defined concept in causation. What would make sense is to ask what setting the income value to 100,000, when it was 60,000, for a particular datapoint $x$ does to the output of that downstream component. I would posit that the important thing about counterfactual patching / interchange intervention is not that it “selects uninformative counterfactual inputs“ but rather that it selects *known* counterfactual inputs, i.e. inputs we know the meaning of. This allows us to say precsely what the causal intervention means, as well as what its effect was. This is the argument in https://proceedings.neurips.cc/paper_files/paper/2021/file/4f5c422f4d49a5a807eda27434231040-Paper.pdf, as far as I can tell.

- Important: while results are plenty promising on the first two case studies, optimal ablation seems to hardly improve over marginal resampling based on Table 1, and the improvement is extremely marginal looking at the rank correlation in Fig. 6 where a linear correlation seems inappropriate for the log-log relationships. So why then, does optimal ablation perform far better than counterfactual patching in Fig. 1, when counterfactual patching was just treated as something of a ground-truth in Sec. 2.  This apparent over-performance in circuit analysis seems like it could be an interpretability artifact, in spite of Appendix D suggesting overfitting is unlikely.
- Important: I really could not tell what the third case study was trying to show, or how it tried to show it. See next point.
- Of some importance: The paper suffers from trying to do so much in nine pages. I think it would be difficult for someone without a deep background in interpretability to read this paper and get as much out of it as the authors would hope. Unless the authors feel strongly about the third case study, I would totally remove that and focus on providing more detail to earlier sections of the paper.
- Of some importance: It is hard give any credit to the paper for the introduction of the uniform gradient sampling (UGS) method because it is introduced without nearly enough detail, and without basically any motivation, in the main body of the paper. And it is evaluated only in one case study. I can’t tell if the UGS direction is big enough to split off into another paper, but it doesn’t feel like it belongs in this one.
- Of some importance: The organization of the paper is a little unorthodox, without a related work or conclusion section. While the introduction does a good job situating the paper in the literature, I would recommend discussing some earlier work on how this entire debate has played out with feature attribution methods before the intensified focus on “mechanistic” interpretability (i.e. methods focused on data space ablations and not only hidden feature ablations), including (1) https://arxiv.org/pdf/1806.10758 (2) https://arxiv.org/pdf/1910.13413 (3) https://arxiv.org/pdf/2106.00786.

**Questions:**

- See questions in the above.
- A note on the references: I think it’s extremely non-standard for the references section to be full of references that do not appear in the paper. I would say that this should be fixed.
- In the intro, I think causal tracing is described as a noising ablation, when it is a denoising ablation.
- (Goldowsky-Dill et al., 2023) recommends →  Goldowsky-Dill et al. (2023) recommend
- “constant to which to ablate” → constant for ablating?
- "confluence of conflicting information may cause model representations to spiral out-of-distribution” — are the hidden states during a forward pass in circuit analysis with optimal ablation in-distribution?
- The sentence beginning with “To assess the magnitude component” is not at all clear.
- The paper is not in correct submission format, as it is missing line numbers.

**Limitations:**

The limitations could discuss some higher level points around feature replacement and what it could be useful, vs. what the paper empirically shows it may be useful for. It could also include some of the caveats about the proposed method from Sec. 2.

---

> ### Author Rebuttal · Authors · 2024-08-07
>
> Thank you for the positive and helpful feedback, and we appreciate your optimism about our work!
>
> >First, the argument in the paper, while being mathematically clear, is not philosophically rigorous
>
> We agree and have rewritten this section (see global reply). The term “total ablation” clarifies what we consider ablation: replacing the value of component A with a random variable that is independent from the input X, so the new value of A cannot provide any information about X. We think the revised section 2.2 crystallizes why it’s desirable to reduce ∆.
>
> >it is not clear that optimal ablation is the answer to an intelligible question. For example, suppose we have a neural model that classifies people into “give loan” and “reject loan” categories, and that we know that a neuron in this model represents a person’s annual income in USD
>
> Interesting example! In this case, we think OA does, in fact, answer an intelligible question: “How much worse does the model perform if it had to treat everyone as having the same income?” which is arguably extremely similar to “How much worse would the model perform if it had no information to distinguish anyone’s income?,” i.e. "How important is information about income to the model's performance?"
>
> > it doesn’t make sense to ask how “important” a component is. Importance is not well-defined concept
>
> We agree it’s not clear “importance” is uniquely defined, but we think it’s still helpful to discuss! If we know a component represents income, there are specific interventions we can assess (e.g. input-counterfactual pairs). But if we aren’t sure what it represents, as is typically the case in interpretability work, we may make mistakes like (as you mentioned) changing its value from Alice to Carol without changing a different component that stores the “word starts with A” feature to a “word starts with C” feature, making the model lose coherence (whereas OA could have changed the component from Alice to a value that maintains consistency with whatever other components reflect). Even if we know what each component represents, it’s often too complicated to assess causal interventions between every pair of inputs, and it’s helpful to have a simple conception of importance that somehow aggregates causal effects to guide our search for relevant components a model uses to perform an algorithm. Our claim is that the average causal effect of setting an activation for every input to the same constant adequately summarizes importance (similar claims are made to justify other ablation methods).
>
> > while results are plenty promising on the first two case studies, optimal ablation seems to hardly improve over marginal resampling based on Table 1
>
> We were also confused, and when we thought about this experiment more carefully, we realized we implemented the ablation methods inconsistently and were not doing an apples-to-apples comparison. For each ablation method, we can either perform some prescribed intervention at the sequence level, or iterate an intervention over each token position in the sequence (note that Prop 2.3 only holds if we are consistent in this choice). For resample ablation, we were performing sequence-to-sequence replacement by resampling between corresponding token activations in a different input sequence, but for mean/optimal ablation, we were performing token-level replacement by setting activations to the same value at all sequence positions.
>
> We updated Table 1 to adopt the first approach for every method (i.e. conditioning on sequence position), which shows a much larger improvement of OA over resampling.
>
> | | Zero | Mean | Resample | CF-Mean | Optimal | CF |
> |-|-|-|-|-|-|-|
> Log-log correlation with CF | 0.626 | 0.831 | 0.826 | 0.847 | 0.908 | 1 |
> Mean ∆ | 0.0584 | 0.0405 | 0.0559 | 0.0412 | 0.0035 | 0.0296 |
> Median ratio of ∆(opt) to ∆ | 11.1% | 33.0% | 17.7% | 31.7% | 100% | 88.9% |
>
> For completeness, we’ll also show in the appendix that if we implement all ablation methods in the tokenwise fashion (e.g. resampling all tokens individually), OA still shows much higher correlation with CF.
>
> > why then, does optimal ablation perform far better than counterfactual patching in Fig. 1, when counterfactual patching was just treated as something of a ground-truth in Sec. 2.
>
> Just to clarify, our stance is that, among *previous* methods, CF may best reflect effect 1 (which we loosely consider “ground truth importance”) from the rewritten part of section 2.2, and we will clarify this point in the paper. Thus, it would make sense that, if OA were now the closest method to “ground truth importance,” then OA would also be the method closest to CF among existing methods. We present Table 1 to confirm this, and do not mean to present CF as a ground truth.
>
> Even for single components, OA achieves lower ∆ on average than CF (0.0035 vs 0.0296), in line with what we see for circuits. The disparity may be larger for circuits due to ablating many components at once. Rushing and Nanda (2024) showed that models can self-repair if a few components contribute weird values; however, ablating many components with previous methods may contribute much more to effect 2 (“spoofing” the model with inconsistent values).
>
> > I really could not tell what the third case study was trying to show
>
> See global reply
>
> > the uniform gradient sampling (UGS) method…is introduced without nearly enough detail
>
> See global reply
>
> > I would recommend discussing some earlier work on how this entire debate has played out with feature attribution methods
>
> Good suggestion and thanks for the references! We will add discussion.
>
> > it’s extremely non-standard for the references section to be full of references that do not appear in the paper
>
> We will correct this and the phrasing corrections mentioned in the questions section.
>
> > The limitations could discuss some higher level points [and] could also include caveats about the proposed method from Sec. 2.
>
> We agree and will add these points.

---

> > ### Comment · Reviewer_nbx7 · 2024-08-10
> > **Reply to rebuttal**
> >
> > Thanks to the authors for their reply above! Some comments below:
> >
> > > We agree and have rewritten this section (see global reply). The term “total ablation” clarifies what we consider ablation: replacing the value of component A with a random variable that is independent from the input X, so the new value of A cannot provide any information about X. We think the revised section 2.2 crystallizes why it’s desirable to reduce ∆.
> >
> > Thanks! I think this is a step in the right direction. I still do not totally agree, because I think there is something missing from the list. The list includes deletion and spoofing, but it could also include “insertion”. Similar to our other income example, changing someone’s income from 100k to 40k might be like deletion (I don’t think it is like spoofing, unless this new feature vector is totally OOD / logically impossible), but I think it is more like “inserting a new value.”
> >
> > For what it’s worth, some of my original criticisms still stand, like the idea of “inconsistency” with other information that is relied on in the “spoofing” definition. But I think the new 2.2 is clearer, and I don’t really expect this paper to present an unassailable definition of inconsistency anyway.
> >
> > >Interesting example! In this case, we think OA does, in fact, answer an intelligible question: “How much worse does the model perform if it had to treat everyone as having the same income?” which is arguably extremely similar to “How much worse would the model perform if it had no information to distinguish anyone’s income?,” i.e. "How important is information about income to the model's performance?”
> >
> > Interesting! I agree with your first claim here, and thanks for pointing it out. We do know the meaning of this operation, which is intervening on everyone’s income for it to be the same. I also understand that you treat the 2nd and 3rd question as the same. This is reasonable, and people used to do it years ago when estimating feature importance for decision trees. They would shuffle the column of a dataset, and check the accuracy of the model. That is, they would do marginal resampling. But the thing is, many ablation methods imply that the model has no information to distinguish anyone’s income, including marginal resampling, zero ablation, constant ablation and random ablation. So what is special about optimal ablation that it is the *right way* to make sure the model has no information about anyone’s income?
> >
> > My other question continues to be, why are we trying to estimate “importance” when it is ill-defined? But I see that comes up next.
> >
> > > We agree it’s not clear “importance” is uniquely defined, but we think it’s still helpful to discuss! If we know a component represents…
> >
> > Ultimately, I agree the whole argument here. Counterfactuals aren’t perfect, and I think we do need a method that “somehow aggregates causal effects to guide our search for relevant components.” This would be a nice big caveat to add to the paper when the word “importance” gets used, since it has haunted interpretability papers for years.
> >
> > >We updated Table 1 to adopt the first approach for every method (i.e. conditioning on sequence position), which shows a much larger improvement of OA over resampling.
> >
> > Ok great!
> >
> > >…We present Table 1 to confirm this, and do not mean to present CF as a ground truth
> >
> > Thanks makes sense.
> >
> > > Even for single components, OA achieves lower ∆ on average than CF (0.0035 vs 0.0296), in line with what we see for circuits. The disparity may be larger for circuits due to ablating many components at once.
> >
> > Thanks, this also makes sense. It would probably be worth adjusting some of the language in the paper to not present CF so much as a ground truth, but more as the closest thing we had to a ground-truth (to the extent the intro does this). Then it would help to present Table 1 and the single-components results on somewhat equal footing, to show that OA could even be better than CF in some ways, while also being easier/automatic. Right now the Table 1 result reads more like a sanity check/validation before moving into “real” results.
> >
> > ---
> >
> > Based on this discussion, I plan on keeping my score at 8. I leave it to the authors to clean up some of the motivation/argument for the method, and to figure out how to present some of the extra UGS / OCA results in a better way, but the core the paper is very good.

---

> > > ### Author Response · Authors · 2024-08-12
> > >
> > > Thank you for the insights and discussion! To touch on a few points:
> > >
> > > >The list includes deletion and spoofing, but it could also include “insertion”. Similar to our other income example, changing someone’s income from 100k to 40k might be like deletion (I don’t think it is like spoofing, unless this new feature vector is totally OOD / logically impossible), but I think it is more like “inserting a new value.”
> > >
> > > This is an excellent point! We thought we had accounted for this option in what we wrote, but upon rereading it we hadn’t, so we will definitely incorporate this category into this part!
> > >
> > > >But the thing is, many ablation methods imply that the model has no information to distinguish anyone’s income, including marginal resampling, zero ablation, constant ablation and random ablation. So what is special about optimal ablation that it is the right way to make sure the model has no information about anyone’s income?
> > >
> > > To be a bit more precise than what we wrote in the rebuttal, the question that optimal ablation answers is, “What is the best performance the model could have achieved on the task with no information to distinguish anyone’s income?” We think this is a good way to frame OA, and arguably component importance in general, and will make this question explicit in the paper.
> > >
> > > We think that “best performance,” rather than just “performance,” is the relevant question to ask when considering a component’s importance. There are many ways to remove information about income (such as assuming that everyone’s income is zero) that achieve worse performance than the optimal-ablated model, but this underperformance cannot be attributed to *removing information about income* (since OA also totally ablates the information in the component), and thus must be attributed to some arbitrary choice made about how to remove this information. This question has a clear parallel to other questions we might ask in ML. If we’re trying to assess, for example, “how much does the assumption of linearity constrain our ability to predict Y from X,” we want to compare the performance of a nonlinear model to the *best* linear model rather than a random linear model. Another way to see the importance of the word “best” (and this descriptor is what what makes OA distinct from other total ablation methods) in the definition of component importance is that we should expect an *unimportant* component to have a value of 0. But for this to be true, we must make sure that when we ablate an unimportant component, we don’t also do something that messes up the model in some other way, since if we do, then we’ll erroneously end up with a non-zero value for that component’s importance.
> > >
> > > We’ll be sure to incorporate these and the other presentational changes you suggested into the paper. Thank you again for the helpful remarks!

---

### Official Review · Reviewer_WFFz · 2024-07-14

**Soundness:** 2
**Presentation:** 2
**Contribution:** 2
**Rating:** 4
**Confidence:** 3

**Summary:**

- Introduces a notion of "optimal component ablation" for activation patching methods in mechanistic interpretability. Specifically, they propose an ablation method that involves setting a component’s value to the constant value that minimizes ablation loss over a chosen distribution.

- Shows that optimal ablation can improve algorithmic circuit discovery—the identification of sparse subnetworks that recover
low loss on interpretable subtasks.

**Strengths:**

- Simple approach to compute 'optimal' ablations that simulate the removal of a given component/vertex. This fares better than over zero, mean, and resample ablation in terms of ablation loss.
- The OCA lens is a clever idea to evaluate layerwise representations and an alternative to linear probing / tuned lens.

**Weaknesses:**

- Writing and paper structure can be significantly better (motivation, problem setup, describing the applications in more cohesive manner, results etc).
- Justification for why ablation loss is the "right" way to measure "optimality" could be better. Also, given that the proposed method explicitly optimizes for ablation loss minimization, it is unclear why making this the primary evaluation metric in the experiments is insightful. Are there other downstream metrics for localization that also improve as a result of minimizing ablation loss? For example, does it find smaller circuits that induce the same effect on model behavior? Is the second-order effect on unrelated tasks less (as intended) due to optimal ablations? These evaluations would make the overall story significantly stronger.
- One of the main contributions (uniform gradient sampling method) is not described well. I would like to see a more step-by-step description of the proposed method. Most of the important details are deferred to the appendix
- These applications are niche and specific to mechanistic interpretability subroutines. To me, factual recall (the second application) seems like a special case of the first one, as the goal is again to localize a given task down to a subset of model components. I would be interested to see optimal ablations improve model editing (e.g., https://arxiv.org/abs/2404.11534 show that accurate localization via zero ablations can be directly used for model editing).
- Novelty concerns. I am not sure if the contributions in this paper are strong enough right now, especially given related work in this area, e.g., https://arxiv.org/abs/2309.16042 also focus on systematically evaluating activation patching
- I like the OCA lens idea, but it's a bit unfair to compare it to tuned lens because OCA lens is still using additional computation via learned ablated layers. I think this experiment requires fair-er baselines that are more of an apples-to-apples comparison (e.g., smaller models with fewer learned layers?)

**Questions:**

See strengths and weaknesses

**Limitations:**

See strengths and weaknesses

---

> ### Author Rebuttal · Authors · 2024-08-07
>
> Thank you for the constructive feedback.
>
> >Writing and paper structure can be significantly better
>
> We’ve revised the writing so that the structure makes more sense, improving flow and clarifying differences between sections. In particular, we’ve substantially rewritten section 2 to better contextualize our motivation and problem setup, and edited later sections to clarify the motivation and description for each application.
>
> >Justification for why ablation loss is the "right" way to measure "optimality" could be better.
>
> We agree and have rewritten this section. See global reply.
>
> >it is unclear why making [ablation loss] the primary evaluation metric in the experiments is insightful
>
> The ablation loss gap ∆ is the primary metric in previous circuit discovery work. While it is true that OA is guaranteed to have lower ∆ than zero, mean, and resample ablation, studying ∆ still produces several surprising results.
>
> 1. The amount that OA reduces ∆ (67-89% for components, 85-97% for circuits) is **very large**. In light of our motivation for minimizing ∆ (see global reply), this suggests most of ∆ for these methods is accounted for by “spoofing,” or inserting inconsistent information into the model, not removing important information.
>
> 2. OA produces lower ∆ than CF, which is not guaranteed, since CF is input-dependent and removes less information than OA.
>
> 3. Manual circuits achieve close to Pareto-optimal ∆ when evaluated with OA, but for other ablation methods, there are similar-size circuits that get much lower ∆ than the manual circuit. This is likely because there are components that do not serve relevant functions according to the manual analysis but need to be included in the circuit to avoid spoofing the circuit components with conflicting information when ablated. If you believe that manual circuits from previous work capture the important mechanisms, then this is evidence that OA is a better evaluation metric for circuits.
>
> We agree with the general idea of having more evaluations. We added a result for OCA lens that shows that it elicits more truthful predictions than tuned lens (see global reply).
>
> >does it find smaller circuits that induce the same effect on model behavior?
>
> We believe we show the answer is yes by looking at the Pareto frontier; for the same effect on model behavior, we can achieve smaller circuits. We will make this conclusion more clear in the paper.
>
> >Is the second-order effect on unrelated tasks less (as intended) due to optimal ablations?
>
> While this is an interesting question, we’re not sure we should expect circuits to generalize to unrelated tasks. We are not trying to find a small general-purpose circuit; we are deliberately trying to isolate a circuit that performs one particular task, and we might even prefer the circuit does not perform unrelated tasks because we’d have wasted circuit complexity.
>
> >uniform gradient sampling method is not described well
>
> See global reply
>
> >These applications are niche and specific to mechanistic interpretability subroutines.
>
> Each of our selected applications is the subject of extensive discussion in interpretability. We acknowledge that in the original submission, we only cited a few relevant works. We’ve added far more citations for each of the applications and previous ablation methods.
>
> Furthermore, as other reviewers have noted, our work presents a new approach to the longstanding question of feature importance and has broad implications beyond mechanistic interpretability. We focus on this area specifically for the sake of cohesion and because it is a growing subfield in which the use of ablation is particularly common and for which our method results in concrete empirical improvements.
>
> >factual recall (the second application) seems like a special case of the first one
>
> We agree that these applications are mathematically similar, but they are answering substantively different scientific questions about a model, and they are separate lines of inquiry in previous work. Circuit discovery asks how a model performs an algorithmic task, and seeks to find components that perform steps of the algorithm. Factual recall asks where the model has memorized factual associations. We will clarify this point in the paper.
>
> >I would be interested to see optimal ablations improve model editing
>
> Thanks for pointing us to this paper, and we agree it would be an interesting result. Our results on localization suggest that our method could do well here. But we think it would require significant extra thought – we’d have to make choices about how to edit the model, compare to various baselines, etc. Given our space limitations, we defer this to future work.
>
> >Novelty concerns. I am not sure if the contributions in this paper are strong enough right now, especially given related work in this area, e.g., https://arxiv.org/abs/2309.16042 also focus on systematically evaluating activation patching
>
> As noted by other reviewers, our approach is fundamentally quite different from existing approaches, which manifests in dramatic differences in several important metrics. To be clear, unlike the linked paper, this paper is not about evaluating previous methods. Our evaluation is about showing our method outperforms others, including all methods that are evaluated in that paper.
>
> >I like the OCA lens idea, but it's a bit unfair to compare it to tuned lens because OCA lens is still using additional computation via learned ablated layers
>
> See global reply for more context about why it is a fair comparison – both methods elicit predictions from the last token position without using activations at previous token positions. Indeed, OCA lens is actually a more constrained elicitation model than tuned lens: it has a factor of $d_\text{model}/n_\text{layers}$ fewer learnable parameters, since we train a $d_\text{model}$-parameter constant at each layer for OCA lens, compared to a $d_\text{model}^2$-parameter weight matrix for tuned lens.

---

### Official Review · Reviewer_mH4h · 2024-07-14

**Soundness:** 4
**Presentation:** 4
**Contribution:** 4
**Rating:** 9
**Confidence:** 3

**Summary:**

Different intervention techniques aim to "ablate" parts of the representation of the model to infer their causal function, e.g. by adding gaussian noise. the paper suggests to derive a notion of "optimal" ablation. Particularly, instead of zeroing out or replacing the ablated part with its mean, it is proposed to optimize with GD the constant value that minimizes the loss, I.e, the "perfect" ablation would be replacing the element with the "best" constant value. It is shown that the proposed techniques improves circuit discovery, identifying a subnetwork that reconstructs the original performance of the model in some task.

**Strengths:**

The contribution is well defined and elegant. The proposed method is experimentally validated on several interpretability-related tasks and it improves over commonly used ablation method.

**Weaknesses:**

I don't see major weaknesses in this work.

**Questions:**

none.

**Limitations:**

see above.

---

> ### Author Rebuttal · Authors · 2024-08-07
>
> Thank you for your generous feedback, and for recognizing the value of our work!

---

### Author Rebuttal · Authors · 2024-08-07

We thank all reviewers for their constructive feedback. We use this space to address questions shared between multiple reviewers.

**Reviewers note our motivation for minimizing ∆ could be clearer.** We agree and propose the following changes. We add the following definition to section 2.1:

>For any model component $A$, a **total ablation method** satisfies $M^{\setminus A}(X) = M_A(X,Z)$ for some random $Z$, where $Z\perp \mkern-18mu \perp X$.

This definition aligns with an intuitive notion of entirely removing $A$: applying a total ablation method prevents $A$ from providing any information that distinguishes between inputs $X$. Total ablation methods include zero, mean, resample, and optimal ablation.

We also rewrite part of section 2.2:

>To see why lower ∆ is desirable, consider that intuitively, for ablation methods that involve intervening on M by replacing A(x) with a different value a, there are two potential contributors to ∆.

>1. Information deletion. The original value A(x) could carry informational content, specific to each x, that serves some mechanistic function in downstream computation and helps the model arrive at its prediction M(x). Replacing A(x) with some other value $a$ could delete this information about x, hindering the model’s ability to compute the original M(x).

>2. Information spoofing. The replacement value $a$ could insert information about the input that is inconsistent with information about x derived from retained components, “spoofing” the downstream computation. This confluence of conflicting information may cause later activations that combine A(x) with other information to become incoherent, leading to high ∆ because these abnormal activations were not observed during training and thus not necessarily regulated to lead to reasonable predictions.

>Measures of component importance seek to isolate the contribution of effect 1 from that of effect 2. Like other total ablation methods, optimal ablation captures a maximal information deletion effect since $a^*$ does not depend on x. However, compared to other methods, OA minimizes the contribution of information spoofing to ∆ by setting ablated components to constants $a^*$ that are maximally consistent with information from other components, e.g. by conveying a lack of information about the input or by hedging against a wide range of possible x rather than strongly associating with a particular input other than the original x. Optimal ablation does not entirely eliminate information spoofing, since it may be the case that every possible value of A conveys at least weak information about the input. However, the excess ablation gap ∆ − ∆(opt) for ∆ measured with ablation methods that replace A(x) with a (random) value A is *entirely* caused by information spoofing, since replacing A(x) with the constant $a^*$ achieves lower loss without giving away any more information about x. In practice, ∆ − ∆(opt) for prior ablation methods is typically very large compared to ∆(opt). In Appendix C.3, we show that on average, ∆(opt) accounts for only 11.1% of ∆(zero), 33.0% of ∆(mean), and 17.7% of ∆(resample) for attention heads and MLP blocks on a prototypical language modeling subtask. Section 3 extends this finding to circuits: circuits manually selected in prior work to capture important mechanisms on language subtasks incur 85-97% lower loss with OA than with other ablation methods. This disparity indicates that effect 2 dominates these other ∆ measurements, making them poor estimators for effect 1 compared to OA.

**Reviewers note that we do not spend much time explaining UGS.** While UGS may be of interest to some, it is an auxiliary contribution. As some pointed out, we packed a lot of content into this paper, which does not leave us room to delve separately into UGS in the main text. We choose to de-prioritize it, but wish to acknowledge that it is novel and could be useful to others, so we prefer to describe it briefly in the main text and defer additional context to the appendix for interested readers.

**Reviewers found the presentation of OCA lens confusing.** We rewrite some of section 5 to better contextualize the motivation for OCA lens and why the comparison to other methods is fair. We add references to logit attribution, a popular precursor to tuned lens. Latent prediction methods, e.g. logit attribution and tuned lens, ascribe semantic meaning to the activation at the last token position (LTP) at an early layer $i$ by making next-token predictions using only that activation.

>Tuned lens allows researchers to study *when* important information is transferred to LTP: if replacing $\ell_N(X)$ with $\hat\ell_N(X)$ achieves low loss, then $\ell_i(X)$ already contains crucial context for computing $M(X)$, and key information was transferred prior to layer $i$. Similar to tuned lens, OCA lens reveals whether the LTP activation at layer $i$ contains sufficient context to compute $M(X)$ by eliminating information transfer from previous token positions to LTP after layer $i$.

Tuned lens and OCA lens share the information-theoretic limitation that they are functions of *only* the LTP activation at layer $i$ and do not depend on activations at other token positions. Tuned lens is a linear map, while OCA lens is a function that involves applying later MLP layers and adding constants, and has far fewer learnable parameters: $O(Nd_{model}) < O(d_{model}^2)$. A key insight is that ablation is a more parameter-efficient way to elicit latent information than training a simple auxiliary model.

**Updates to results.** We update Figure 2 by augmenting our dataset and slightly modifying the metric. (See “Comment on Figure 2” in our reply to review Nr7M for more.) Additional figures show more added results for factual recall, varying token position and window size. We also add a new evaluation for OCA lens demonstrating that elicited responses on news and wiki datasets are more truthful than those for tuned lens.

---

### Decision · Program_Chairs · 2024-09-25

**Decision:**

Accept (spotlight)

**Comment:**

Pretty interesting paper on how to intervene on model internals in a principled way. Standard approaches are to zero out or add noise to model components in order to probe for their role in performing certain tasks or subtasks. Instead, the authors instead propose to find the particular constant value that minimizes the loss. They then apply this idea to some problems in mechanistic interpretability, such as finding locations in the model weights for memorized facts, a useful task for model editing. Most reviewers agree that the paper is innovative and interesting. The authors’ rebuttal also addresses a lot of the questions and should lead to a solid paper.